# SECOND ORDER BOUNDS FOR CONTEXTUAL BANDITS WITH FUNCTION APPROXIMATION

Aldo Pacchiano

Boston University and Broad Institute of MIT and Harvard

## ABSTRACT

Many works have developed no-regret algorithms for contextual bandits with function approximation, where the mean rewards over context-action pairs belong to a function class $\mathcal{F}$. Although there are many approaches to this problem, algorithms based on the principle of optimism, such as optimistic least squares have gained in importance. The regret of optimistic least squares scales as $\widetilde{\mathcal{O}}\left(\sqrt{d_{\text{eluder}}(\mathcal{F})\log(\mathcal{F})T}\right)$ where $d_{\text{eluder}}(\mathcal{F})$ is a statistical measure of the complexity of the function class $\mathcal{F}$ known as eluder dimension. Unfortunately, even if the variance of the measurement noise of the rewards at time $t$ equals $\sigma_t^2$ and these are close to zero, the optimistic least squares algorithm's regret scales with $\sqrt{T}$. In this work we are the first to develop algorithms that satisfy regret bounds for contextual bandits with function approximation of the form $\widetilde{\mathcal{O}}\left(\sigma\sqrt{\log(\mathcal{F})d_{\text{eluder}}(\mathcal{F})T} + d_{\text{eluder}}(\mathcal{F}) \cdot \log(|\mathcal{F}|)\right)$ when the variances are unknown and satisfy $\sigma_t^2 = \sigma$ for all $t$ and $\widetilde{\mathcal{O}}\left(d_{\text{eluder}}(\mathcal{F})\sqrt{\log(\mathcal{F})\sum_{t=1}^{T}\sigma_t^2} + d_{\text{eluder}}(\mathcal{F}) \cdot \log(|\mathcal{F}|)\right)$ when the variances change at every time-step. These bounds generalize existing techniques for deriving second order bounds in contextual linear problems.

## 1 INTRODUCTION

Modern decision-making algorithms have achieved impressive success in many important problem domains, including robotics Kober et al. (2013); Lillicrap et al. (2015), games Mnih et al. (2015); Silver et al. (2016), dialogue systems Li et al. (2016), and online personalization Agarwal et al. (2016); Tewari & Murphy (2017). Problems in these domains are characterized by the interactive nature of the data collection process. For example, to train a robotic agent to perform a desired behavior in an unseen environment, the agent is required to interact with the environment in a way that empowers it to learn about the world, while at the same time learning how to best achieve its objectives. Many models of sequential interaction have been proposed in the literature to capture scenarios such as this. Perhaps the most basic one is the multi-armed bandit model Thompson (1933); Lai & Robbins (1985); Auer et al. (2002a); Lattimore & Szepesvári (2020), where it is assumed a learner has access to $K \in \mathbb{N}$ arms (actions), such that when playing any of these results in a random reward. Typically, the learner's objective is to select actions, and observe rewards in order to learn which arm produces the highest mean reward value. Algorithms for the multi-armed bandit model can be used to solve problems such as selecting a treatment that in expectation over the population achieves the best expected success.

Deploying an algorithm designed for the multi-armed bandit setting may be suboptimal for applications where personalized policies are desirable, for example, when we would like to design a treatment regime that maximizes the expected success rate conditioned on an individual's characteristics. This situation arises in many different scenarios, from medical trials Villar et al. (2015); Aziz et al. (2021), to education Erraqabi et al. (2016), and recommendation systems Li et al. (2010) . In many of these decision-making scenarios, it is often advantageous to consider contextual information when making decisions. This recognition has sparked a growing interest in studying adaptive learning algorithms

in the setting of contextual bandits Langford & Zhang (2007); Li et al. (2010); Agrawal & Goyal (2013) and reinforcement learning (RL) Sutton (1992).

In the contextual bandit model, a learner interacts with the world in a sequential manner. At the start of round $t \in \mathbb{N}$ the learner receives a context $x_t \in \mathcal{X}$, for example in the form of user or patient features. The learner then selects an action to play $a_t \in \mathcal{A}$, representing for example a medical treatment, and then observes a reward $r_t \in \mathbb{R}$ that depends on the context $x_t$, the action $a_t$ and may be random. For example $r_t$ may be the random binary outcome of a medical treatment $a_t$ on a specific patient $x_t$. The study of contextual bandit scenarios has produced a rich literature. Many aspects of the contextual bandit model have been explored, such as regret bounds under adversarial rewards Auer et al. (2002b); Lattimore & Szepesvári (2020); Neu & Olkhovskaya (2020), learning with offline data Dudík et al. (2012), the development of statistical complexity measures that characterize learnability in this model Russo & Van Roy (2013); Foster et al. (2021) and others.

The focus of many works, including this one is to flesh out the consequences of different modeling assumptions governing the relationship between the context, the action and the reward For example by developing algorithms for scenarios where the reward is a linear function of a linear function of an embedding of the context and action pair Auer (2002); Rusmevichientong & Tsitsiklis (2010); Chu et al. (2011); Abbasi-Yadkori et al. (2011). This has lead to algorithms such as OFUL that can be used to derive bounds for contextual bandit problems with linear rewards Abbasi-Yadkori et al. (2011). Other works have considered scenarios that go beyond the linear case, where it is assumed the reward function over context action pair $x_t, a_t$ is realized by an unknown function $f_\star$ belonging to a known function class $\mathcal{F}$ (which can be more complex than linear). Various adaptive learning procedures compatible with generic function approximation have been proposed for contextual bandit problems. Among these, we highlight two significant methods relevant to our discussion; the Optimistic Least Squares algorithm introduced by Russo & Van Roy (2013) and the SquareCB algorithm introduced by Foster & Rakhlin (2020). Both of these methods offer guarantees for cumulative regret. Specifically, the cumulative regret of Optimistic Least Squares scales with factors $\mathcal{O}(\sqrt{d_{\mathrm{eluder}}(\mathcal{F}) \log(|\mathcal{F}|)})$, while the cumulative regret of Square CB scales as $\mathcal{O}\left(\sqrt{|\mathcal{A}| \log(|\mathcal{F}|)}\right)$, where $\mathcal{A}$ is the set of actions. The eluder dimension[1] ($d_{\mathrm{eluder}}$) is a statistical complexity measure introduced by Russo & Van Roy (2013), that enables deriving guarantees for adaptive learning algorithms based on the principle of optimism in contextual bandits and reinforcement learning Li et al. (2022); Jin et al. (2021); Osband & Van Roy (2014); Chan et al. (2021).

The design of algorithms that can handle rich function approximation scenarios represents a great leap towards making the assumptions governing contextual bandit models more realistic and the algorithms more practical. The concerns addressed by this line of research are focused on the nature of the mean reward function. Nonetheless, they have left open the study of the dependence on the noise $\xi_t = r_t - f_\star(x_t, a_t)$. Intuitively, as the conditional variance of $\xi_t$ decreases, the value of $r_t$ contains more information about the reward function $f_\star$. Algorithms that leverage the scale of the variance to achieve sharp regret bounds are said to satisfy a variance aware or second order bound.

Different works have considered this research direction and developed variance-dependent bounds for linear and contextual linear bandits Kirschner & Krause (2018); Zhou et al. (2021); Kim et al. (2022); Zhao et al. (2023); Xu et al. (2024). In summary, the sharpest bounds for contextual linear bandits are achieved by the SAVE Algorithm in Zhao et al. (2023) and scale (up to logarithmic factors) as $\mathcal{O}\left(d\sqrt{\sum_{t=1}^{T} \sigma_t^2}\right)$ where $\sigma_t^2$ is the conditional variance of $\xi_t$, the time $t$ measurement noise.

In the context of function approximation, second order bounds for contextual bandits have been developed in Zhao et al. (2022) under the assumption that the value of the conditional variances $\sigma_t$ is observed. This restrictive assumption has been lifted in more recent work Wang et al. (2024b;a) under a stronger distributional realizability assumption. In Wang et al. (2024b;a), the authors assume realizability of the noise distribution, that is, the existence of a function class that fits not only the mean rewards as a function of context-action pairs, but also the measurement noise. This is a somewhat restrictive assumption since it effectively reduces the set of problems that can be solved to parametric scenarios where the distributional class of the noise is known; something that in practical settings typically means simple scenarios such as gaussian or bernoulli noise.

---

[1] We formally introduce this quantity in Section 2. Here we use a simpler notation to avoid confusion.

A recent work Jia et al. (2024), published while our paper was under review, removes the assumptions made in Wang et al. (2024b;a). Here a summary of their results. When the variances $\sigma_t$ are revealed with the contexts, they show that for some function classes, any algorithm must incur a regret of $\Omega\left(\sqrt{\min(|\mathcal{A}|, d_{\text{eluder}})\Lambda} + \min(d_{\text{eluder}}, \sqrt{|\mathcal{A}|T})\right)$, where $\Lambda = \sum_{t=1}^{T} \sigma^2$. They also propose an algorithm with an upper bound of $\mathcal{O}\left(\sqrt{|\mathcal{A}|\Lambda \log(|\mathcal{F}|)} + d_{\text{eluder}} \log(|\mathcal{F}|)\right)$. In this setting, our techniques from Section 3 yield a refined bound of $\mathcal{O}\left(\sqrt{d_{\text{eluder}}\Lambda \log(|\mathcal{F}|)} + d_{\text{eluder}} \log(|\mathcal{F}|)\right)$ (see Theorem 3.5 for the special case where all variances are equal). For the setting where variances are not revealed with $x_t$ and may depend on the action $a_t$, Jia et al. (2024) derives the lower bound $\Omega\left(\min\left(\sqrt{d_{\text{eluder}}\Lambda} + d_{\text{eluder}}, \sqrt{|\mathcal{A}|T}\right)\right)$. For the unknown fixed-variance case, our bounds match their lower bound (see Theorem 4.6). They also present an algorithm achieving an upper bound of $\mathcal{O}\left(d_{\text{eluder}}\sqrt{\Lambda \log(|\mathcal{F}|)} + d_{\text{eluder}} \log(|\mathcal{F}|)\right)$, which matches our result in Theorem 4.10.

**Contributions.** In this work we present second order bounds for contextual bandit problems under a mean reward realizability assumption. The techniques we develop are inspired by previous works on variance aware linear bandits such as Zhao et al. (2023), and rely on an uncertainty filtered multi-scale least squares procedure. We are able to make the connection to general function approximation by refining existing techniques to prove eluder dimension regret bounds such as those presented in Russo & Van Roy (2013); Chan et al. (2021); Pacchiano et al. (2024). These techniques should be easily extended to the setting of reinforcement learning and beyond, thus unlocking an important area of research. The sharpest bounds we develop in this work (satisfied by the same algorithm) have the form $\mathcal{O}\left(d_{\text{eluder}}\sqrt{\log(|\mathcal{F}|)\sum_{t=1}^{T}\sigma_t^2} + d_{\text{eluder}} \cdot \log(|\mathcal{F}|)\right)$ when we allow different conditional variances during all time-steps, and $\mathcal{O}\left(\sigma\sqrt{d_{\text{eluder}}\log(|\mathcal{F}|)T} + d_{\text{eluder}} \cdot \log(|\mathcal{F}|)\right)$ when $\sigma_t = \sigma$ for all $t$. Although it is likely our bounds are not the sharpest in the case of different variances, since eluder dimension bounds as in Russo & Van Roy (2013) suggest the dominating term in the optimal bound should scale as $\mathcal{O}\left(\sqrt{d_{\text{eluder}}\log(|\mathcal{F}|)\sum_{t=1}^{T}\sigma_t^2}\right)$, we believe a sharper analysis based on our ideas might be sufficient to prove such a result.

## 2 PROBLEM DEFINITION

In this section we consider the scenario of contextual bandits, where at time $t$ the learner receives a context $x_t \in \mathcal{X}$ belonging to a context set $\mathcal{X}$, decides to take an action $a_t \in \mathcal{A}$ and observes a reward of the form $r_t$ such that $\mathbb{E}_t[r_t] = f_\star(x_t, a_t)$ where it is assumed that $f_\star \in \mathcal{F}$ for $\mathcal{F}$ a known class of functions with domain $\mathcal{X} \times \mathcal{A}$ (see Assumption 2.1). Throughout this section we use the notation $r_t = f_\star(x_t, a_t) + \xi_t$ so that the conditional expectation of $\xi_t$ satisfies $\mathbb{E}_t[\xi_t] = 0$. Throughout this work we will use the notation $\sigma_t^2 = \text{Var}_t(\xi_t)$ to denote the time $t$ conditional variance of the noise. We'll assume the random variables $r_t$ are bounded by a known parameter $B > 0$ with probability one.

The objetive of this work is to design algorithms with sublinear regret. Regret is a measure of performance defined in the realizable contextual scenario studied in this work as the cumulative difference between the best expected reward the learner may have achieved at each of the contexts it interacted with and the expected reward of the actions played.

$$\text{Regret}(T) = \sum_{t=1}^{T} \max_{a \in \mathcal{A}} f_\star(x_t, a) - f_\star(x_t, a_t)$$

The objective is to design algorithms with regret scaling sublinearly with the time horizon $T$.

**Assumption 2.1** (Realizability). *There exists a (known) function class $\mathcal{F} : \mathcal{X} \times \mathcal{A} \to \mathbb{R}$ such that $\mathbb{E}_t[r_t] = f_\star(x_t, a_t)$ for all $t \in \mathbb{N}$.*

**Assumption 2.2** (Boundedness). *There exists a (known) constant $B > 0$ such that $|r_\ell|, |\xi_\ell| \le B$ and $\max_{x \in \mathcal{X}, a \in \mathcal{A}} |f(x, a)| \le B$ and $\max_{x \in \mathcal{X}, a \in \mathcal{A}} |f(x, a) - f'(x, a)| \le B$ for all $f, f' \in \mathcal{F}$ and all $\ell \in \mathbb{N}$.*

The sample complexity analysis of our algorithms will rely on a combinatorial notion of statistical complexity of a scalar function class known as Eluder Dimension Russo & Van Roy (2013). We reproduce the necessary definitions here for completeness.

**Definition 2.1.** ($\epsilon-$*dependence*) *Let $\mathcal{G}$ be a scalar function class with domain $\mathcal{Z}$ and $\epsilon > 0$. An element $z \in \mathcal{Z}$ is $\epsilon-$dependent on $\{z_1, \cdots, z_n\} \subseteq \mathcal{Z}$ w.r.t. $\mathcal{G}$ if any pair of functions $g, g' \in \mathcal{G}$ satisfying $\sqrt{\sum_{i=1}^{n}(g(z_i) - g'(z_i))^2} \leqslant \epsilon$ also satisfies $g(z) - g'(z) \leqslant \epsilon$. Furthermore, $z \in \mathcal{Z}$ is $\epsilon-$independent of $\{z_1, \cdots, z_n\}$ w.r.t. $\mathcal{G}$ if it is not $\epsilon-$dependent on $\{z_1, \cdots, z_n\}$.*

**Definition 2.2.** ($\epsilon$-*eluder*) *The $\epsilon-$non monotone eluder dimension $\overline{d_{\text{eluder}}}(\mathcal{G}, \epsilon)$ of $\mathcal{G}$ is the length of the longest sequence of elements in $\mathcal{Z}$ such that every element is $\epsilon-$independent of its predecessors. Moreover, we define the $\epsilon-$eluder dimension $d_{\text{eluder}}(\mathcal{G}, \epsilon)$ as $d_{\text{eluder}}(\mathcal{G}, \epsilon) = \max_{\epsilon' \geqslant \epsilon} \overline{d_{\text{eluder}}}(\mathcal{G}, \epsilon)$.*

In order to introduce our methods we require some notation. The uncertainty radius function is a mapping $\omega : \mathcal{X} \times \mathcal{X} \times \mathcal{P}(\mathcal{F}) \to \mathbb{R}$ is defined as,

$$\omega(x, a, \mathcal{G}) = \max_{f, f' \in \mathcal{G}} f(x, a) - f'(x, a)$$

for $x \in \mathcal{X}, a \in \mathcal{A}, \mathcal{G} \subseteq \mathcal{F}$. The quantity $\omega(x, a, \mathcal{G})$ equals the maximum fluctuations in value for the function class $\mathcal{G}$ when evaluated in context $x \in \mathcal{X}$ and action $a \in \mathcal{A}$. Throughout this work we will use the notation $\Sigma(A, B, \cdots, C)$ to denote the sigma algebra generated by the random variables $A, B, \cdots, C$.

In this work we design the first algorithm for contextual bandits with function approximation that satisfies a variance dependent regret bound. In this work we extend the optimistic least squares algorithm for contextual bandits with function approximation Russo & Van Roy (2013). Our main result (simplified) states that,

**Theorem 2.1** (Simplified). *Let $\delta \in (0, 1)$. There exists an algorithm that achieves a regret rate of,*

$$\text{Regret}(T) \leqslant \widetilde{\mathcal{O}} \left( d_{\text{eluder}} \left( \mathcal{F}, \frac{B}{T} \right) \sqrt{\left( \sum_{t=1}^{T} \sigma_t^2 \right) \log\left(|\mathcal{F}|/\delta\right)} + B d_{\text{eluder}} \left( \mathcal{F}, \frac{B}{T} \right) \log(|\mathcal{F}|/\delta) \right)$$

*for all $T \in \mathbb{N}$ with probability at least $1 - \delta$. Where $\widetilde{\mathcal{O}}(\cdot)$ hides logarithmic dependencies.*

## 3 SECOND ORDER OPTIMISTIC LEAST SQUARES WITH KNOWN VARIANCE

In this section we introduce an algorithm that satisfies second order regret bounds. Algorithm 1 takes as input a variance upper bound $\sigma^2$ such that $\sigma_t^2 \leqslant \sigma^2$, and achieves a regret bound of order

$$\text{Regret}(T) \leqslant \mathcal{O} \left( \sigma \sqrt{d_{\text{eluder}}(\mathcal{F}, B/T) T \log(T|\mathcal{F}|/\delta)} + d_{\text{eluder}} \cdot \log(T|\mathcal{F}|/\delta) \right).$$

This is a warm-up example that will be sharpened in section 4.2 to the case where the variance is unknown where we can achieve regret bounds of the same order. This algorithm is based on an uncertainty filtered least squares procedure that satisfies sharper bounds than the unfiltered ordinary least squares guarantees. For a complete discussion of estimation bounds for least squares, and their use in the optimistic least squares algorithm from Russo & Van Roy (2013) see Appendix B. Since $\sigma^2 \leqslant B$ this bound could be much smaller than the regret bound for Optimistic Least Squares (Algorithm 3) described in Theorem B.4 that scale as $\mathcal{O}(\sigma \sqrt{d_{\text{eluder}}(\mathcal{F}, B/T) T \log(T|\mathcal{F}|/\delta)})$. In this section we work under the following assumption that we relax in section 4.2,

**Assumption 3.1** (Known Variance Upper Bound). *There exists a (known) constant $\sigma > 0$ such that $\sigma_t \leqslant \sigma^2$ for all $t \in \mathbb{N}$.*

Given a data stream $\{(x_\ell, a_\ell, r_\ell)\}_{\ell \in \mathbb{N}}$ where $r_\ell = f_\star(x_\ell, a_\ell) + \xi_\ell$ for $f_\star \in \mathcal{F}$ such that $\xi_\ell$ is conditionally zero mean, a sequence of subsets of $\mathcal{G}_t \subseteq \cdots \mathcal{G}_2 \subseteq \mathcal{G}_1 = \mathcal{F}$ such that $\mathcal{G}_t$ is a function of $\{(x_\ell, a_\ell, r_\ell)\}_{\ell=1}^{t-1}$, and $f_\star \in \mathcal{G}_t$ for all $t \in \mathbb{N}$. Given $\tau > 0$ we define an uncertainty filtered least squares objective that takes a filtering parameter $\tau > 0$ and defines a least squares regression function computed only over datapoints whose uncertainty radius is smaller than $\tau$,

$$f_t^\tau = \underset{f \in \mathcal{G}_{t-1}}{\text{argmin}} \sum_{\ell=1}^{t-1} (f(x_\ell, a_\ell) - r_\ell)^2 \mathbf{1}(\omega(x_\ell, a_\ell, \mathcal{G}_\ell) \leqslant \tau) \tag{1}$$

The uncertainty filtering procedure will allow us to prove a least squares guarantee with dependence on the variance and also on a vanishing low order term that scales with $\tau B$. We'll use the notation

$$\beta_t(\tau, \tilde{\delta}, \tilde{\sigma}^2) = (4\min(\tau B, B^2) + 16\tilde{\sigma}^2)\log(t|\mathcal{F}|/\tilde{\delta})$$

to denote the confidence radius function, in this case a function of $\tau, \tilde{\delta}$ and $\tilde{\sigma}^2$. Algorithm 1 shows the pseudo-code for our Second Order Optimistic Algorithm.

---

**Algorithm 1** Second Order Optimistic Least Squares

1: **Input:** function class $\mathcal{F}$, variance upper bound $\sigma^2$.
2: Set the initial confidence set $\mathcal{G}_0 = \mathcal{F}$.
3: **for** $t = 1, 2, \cdots$ **do**
4:     Compute regression function for each threshold level $\tau_i = \frac{B}{2^i}$ for $i \in \{0\} \cup [q_t]$ where $q_t = \lceil \log(t) \rceil$

$$f_t^{\tau_i} = \operatorname*{argmin}_{f \in \mathcal{G}_{t-1}} \sum_{\ell=1}^{t-1} (f(x_\ell, a_\ell) - r_\ell)^2 \mathbf{1}(\omega(x_\ell, a_\ell, \mathcal{G}_\ell) \leq \tau_i)$$

5:     Compute threshold confidence sets for all $i \in \{0\} \cup [q_t]$,

$$\mathcal{G}_t(\tau_i) =$$

$$\left\{ f \in \mathcal{F} : \sum_{\ell=1}^{t-1} \left( f_t^\tau(x_\ell, a_\ell) - f(x_\ell, a_\ell) \right)^2 \mathbf{1}(\omega(x_\ell, a_\ell, \mathcal{G}_\ell) \leq \tau_i) \leq \beta_t \left( \tau_i, \delta_i, \sigma^2 \right) \right\} \cap \mathcal{G}_{t-1}(\tau_i) \tag{2}$$

6:     where $\delta_i = \frac{\delta}{2(i+1)^2}$.
7:     Compute $\mathcal{G}_t = \mathcal{G}_{t-1} \cap (\cap_{i=0}^q \mathcal{G}_t(\tau_i))$
8:     Receive context $x_t$.
9:     Compute $U_t(x_t, a) = \max_{f \in \mathcal{G}_t} f(x_t, a)$ for all $a \in \mathcal{A}$.
10:     play $a_t = \operatorname{argmax}_{a \in \mathcal{A}} U_t(x_t, a)$ and receive $r_t = f_\star(x_t, a_t) + \xi_t$.
11: **end for**

---

Notice that by definition in Algorithm 1 the confidence sets satisfy $\mathcal{G}_\ell \subseteq \mathcal{G}_{\ell'}$ for all $\ell \geq \ell'$. In order to state our results we'll define a sequence of events $\{\mathcal{E}_\ell\}_{\ell=1}^\infty$ such that $\mathcal{E}_\ell$ corresponds to the event that $f_\star \in \mathcal{G}_{\ell-1}$ and therefore $f_\star \in \mathcal{G}_{\ell'}$ for all $\ell' \leq \ell - 1$. The following proposition characterizes the error of the filtered least squares estimator $f_t^\tau$ when $\mathcal{E}_t$ holds.

**Proposition 3.1.** *[Variance Dependent Least Squares] Let $t \in \mathbb{N}, \tau \geq 0$ and $\tilde{\delta} > 0$. If $\sigma_\ell^2 \leq \tilde{\sigma}^2$ for all $\ell \leq t - 1$ and $\mathcal{E}_t$ holds then*

$$\mathbb{P}\left( \sum_{\ell=1}^{t-1} \left( f_t^\tau(x_\ell, a_\ell) - f_\star(x_\ell, a_\ell) \right)^2 \mathbf{1}(\omega(x_\ell, a_\ell, \mathcal{G}_\ell) \leq \tau) \leq \beta_t(\tau, \tilde{\delta}, \tilde{\sigma}^2), \ \mathcal{E}_t \right) \geq \mathbb{P}(\mathcal{E}_t) - \tilde{\delta}. \tag{3}$$

The proof of Proposition 3.1 can be found in Appendix C. It follows the structure of the least squares result from Proposition 3.1. For a given $\tau > 0$, estimator $f_t^\tau$ achieves a sharper bound than the ordinary least squares estimator because the low order term in the portion of the analysis that requires the use of Freedman's inequality (see Lemma A.1) that has a magnitude scaling with the error of $f_t^\tau$ on historial points can be upper bounded by $\tau$ instead of scaling with $B$. This results in a second order term scaling with $\min(\tau B, B^2)$ instead of $B^2$ as is reflected by the definition of $\beta_t(\tau, \tilde{\delta}, \tilde{\sigma}^2)$.

In contrast with the results of Lemma B.1 the confidence radius of the $\tau$-uncertainty filtered least squares estimator depends on a variance upper bound whereas the uncertainty radius in Lemma B.1 doesn't. Proposition 3.1 provides us with a variance aware least squares guarantee. If the uncertainty threshold $\tau$ is small, the historical least squares error captured by equation 3 scales with $\sigma^2 \log(t|\mathcal{F}|/\tilde{\delta})$ and does not depend on the scale of $B$. Algorithm 1 leverages these confidence sets to design a variance aware second order optimistic least squares algorithm. The basis of the regret analysis for Algorithm 1 is the validity of the confidence sets $\mathcal{G}_t$ and therefore the estimators $U_t(x_t, a_t)$ being optimistic.

**Lemma 3.2.** *The confidence intervals are valid so that $f_\star \in \mathcal{G}_t$ for all $t \in \mathbb{N}$ and optimism holds, $\max_{a \in \mathcal{A}} f_\star(x_t, a) \leqslant U_t(x_t, a_t)$ with probability at least $1 - \delta$ for all $t \in \mathbb{N}$.*

The proof of Proposition 3.2 can be found in Appendix C. From now on we denote by $\mathcal{E}$ the event described in Lemma 3.2 where all the confidence intervals are valid. In order to relate the regret to the eluder dimension of $\mathcal{F}$, we develop a sharpened version of Lemma B.3 to bound the sum of the uncertainty widths over the context-action pairs played by Algorithm 1. Lemma B.3's guarantees are insufficient to yield the desired result because this result is unable to leverage any dependence on the scale of the widths in the definition of the confidence sets. This is sufficient to show a regret bound as it is evident by following the same logic as in the analysis of the optimistic least squares (Theorem B.4). In order to prove this result, we need to first bound the number of context-action pairs with large uncertainty radius.

**Lemma 3.3.** *If Algorithm 1 is run with input variance upper bound $\sigma > 0$, $\mathcal{E}$ is satisfied and $\{\mathcal{G}_t\}_{t=1}^\infty$ is the sequence of confidence sets produced by Algorithm 1 then for all $T \in \mathbb{N}$ and $\tau \geqslant \tau_{q_T}$ ,*

$$\sum_{t=1}^T \mathbf{1}(\omega(x_t, a_t, \mathcal{G}_t) > \tau) \leqslant 3 \cdot d_{\mathrm{eluder}}(\mathcal{F}, \tau) \left( \frac{64B \log(T|\mathcal{F}|/\delta)}{\tau} + \frac{64\sigma^2 \log(T|\mathcal{F}|/\delta)}{\tau^2} + 1 \right)$$

Lemma 3.3 can be used to show the following sharpened version of Lemma B.3.

**Lemma 3.4.** *If $\mathcal{E}$ holds, then for all $T \in \mathbb{N}$ the uncertainty widths of context-action pairs from Algorithm 1 satisfy,*

$$\sum_{t=1}^T \omega(x_t, a_t, \mathcal{G}_t) \leqslant \mathcal{O} \left( \sigma \sqrt{d_{\mathrm{eluder}}(\mathcal{F}, B/T) \log(T|\mathcal{F}|/\delta) T} + B d_{\mathrm{eluder}}(\mathcal{F}, B/T) \log(T) \log(T|\mathcal{F}|/\delta) \right).$$

The proof of this result is based on an integration argument that leverages the inequality in Lemma 3.3.

Algorithm 1 satisfies the following regret bound,

**Theorem 3.5.** *If $\delta \in (0, 1)$ is the input to Algorithm 1 satisfies,*

$$\mathrm{Regret}(T) \leqslant \mathcal{O} \left( \sigma \sqrt{d_{\mathrm{eluder}}(\mathcal{F}, B/T) \log(T|\mathcal{F}|/\delta) T} + B \cdot d_{\mathrm{eluder}}(\mathcal{F}, B/T) \log(T) \log(T|\mathcal{F}|/\delta) \right).$$

*for all $T \in \mathbb{N}$ with probability at least $1 - \delta$.*

The proof of this Theorem can be found in Appendix D.

## 4 CONTEXTUAL BANDITS WITH UNKNOWN VARIANCE

In the case where the variance is not known our contextual bandit algorithms work by estimating the cumulative variance up to constant multiplicative accuracy and use this estimator to build confidence sets as in Algorithms 3 and 1. In section 4.1 we describe how to successfully estimate the cumulative variance in contextual bandit problems, in section 4.2 we show how to adapt a version of Algorithm 1 to the case of a single unknown variance and finally in section 4.3 we introduce Algorithm 2 that satisfies a regret guarantee whose dominating term scales with $d_{\mathrm{eluder}} \sqrt{\log(|\mathcal{F}|) \sum_{t=1}^T \sigma_t^2}$, and the low order term with $d_{\mathrm{eluder}} \cdot \log(|\mathcal{F}|)$.

### 4.1 VARIANCE ESTIMATION IN CONTEXTUAL BANDIT PROBLEMS

In this section we discuss methods for estimating the variance in contextual bandit problems. Our estimator is the cumulative least squares error of a sequence of (biased) estimators. Given context-action pairs and reward information $\{(x_\ell, a_\ell, r_\ell)\}_{\ell=1}^{t-1}$ and a filtering process $\mathbf{b}_t = \{b_\ell\}_{\ell=1}^{t-1}$ of bernoulli random variables $b_\ell \in \{0, 1\}$ such that $b_\ell$ is $\Sigma(x_1, a_1, b_1, r_1, \cdots, x_{\ell-1}, a_{\ell-1}, b_{\ell-1}, r_{\ell-1}, x_\ell, a_\ell)$-measurable. Let $f_t^{\mathbf{b}_t}$ be the "filtered" least squares estimator:

$$f_t^{\mathbf{b}_t} = \operatorname*{argmin}_{f \in \mathcal{F}} \sum_{\ell=1}^{t-1} b_\ell \cdot (f(x_\ell, a_\ell) - r_\ell)^2 .$$

A filtered least squares estimator satisfies a least squares bound similar to Lemma B.1,

**Lemma 4.1.** *Let $\tilde{\delta} \in (0,1)$, $t \in \mathbb{N}$, $\{x_\ell, a_\ell\}_{\ell=1}^{t-1}$ be a sequence of context-action pairs and and $\{r_t\}_{t=1}^{t-1}$ be a sequence of values satisfying $r_\ell = f_\star(x_\ell, a_\ell) + \xi_\ell$ where $f_\star \in \mathcal{F}$ and the $\xi_\ell$ are conditionally zero mean. Let $\{b_\ell\}_{\ell=1}^{t-1}$ be a filtering process of Bernoulli random variables $b_\ell \in \{0,1\}$ such that $b_\ell$ is $\Sigma(x_1, a_1, b_1, r_1, \cdots, x_{\ell-1}, a_{\ell-1}, b_{\ell-1}, r_{\ell-1}, x_\ell, a_\ell)$-measurable. Let $f_t^{\mathbf{b}_t} = \operatorname{argmin}_{f \in \mathcal{F}} \sum_{\ell=1}^{t-1} b_\ell \cdot (f(x_\ell, a_\ell) - r_\ell)^2$ be the "filtered" least squares estimator. If Assumption 2.2 holds then,*

$$\left| \sum_{\ell=1}^{t-1} \xi_\ell \cdot b_\ell \cdot \left( f_\star(x_\ell, a_\ell) - f_t^{\mathbf{b}_t}(x_\ell, a_\ell) \right) \right| \leqslant 6B^2 \log(2|\mathcal{F}|/\tilde{\delta}).$$

*and*

$$\sum_{\ell=1}^{t-1} b_\ell \cdot \left( f_t^{\mathbf{b}_t}(x_\ell, a_\ell) - f_\star(x_\ell, a_\ell) \right)^2 \leqslant 8B^2 \log(2|\mathcal{F}|/\tilde{\delta})$$

*with probability at least $1 - \tilde{\delta}$.*

Based on the definitions above we will consider the following cumulative variance estimator for a filtered context, action, reward process:

$$W_t^{\mathbf{b}_t} = \sum_{\ell=1}^{t-1} b_\ell \cdot (r_\ell - f_t^{\mathbf{b}_t}(x_\ell, a_\ell))^2. \tag{4}$$

We now prove this estimator achieves a small error.

**Lemma 4.2.** *Let $\tilde{\delta} \in (0,1)$ be a probability parameter. If Assumption 2.2 holds,*

$$\frac{2}{3} \cdot W_t^{\mathbf{b}_t} - 11B^2 \log(4|\mathcal{F}|/\tilde{\delta}) \leqslant \overline{W}_t^{\mathbf{b}_t} \leqslant 2W_t^{\mathbf{b}_t} + 48B^2 \log(4|\mathcal{F}|/\tilde{\delta})$$

*with probability at least $1 - \tilde{\delta}$ where $\overline{W}_t^{\mathbf{b}_t} = \sum_{\ell=1}^{t-1} b_\ell \cdot \sigma_\ell^2$.*

Using the union bound (by setting $\tilde{\delta} = \frac{\delta'}{2 \cdot t^2}$ in Lemma 4.1) we can write an anytime guarantee for the variance estimators $W_t^{\mathbf{b}_t}$).

**Corollary 4.3.** *Let $\delta' \in (0,1)$, $\{x_\ell, a_\ell, r_\ell\}_{\ell=1}^{\infty}$ be a sequence of context-action and rewards triplets such that $r_\ell = f_\star(x_\ell, a_\ell) + \xi_\ell$ where $f_\star \in \mathcal{F}$ and the $\xi_\ell$ are conditionally zero mean. Let $\{b_\ell\}_{\ell=1}^{t-1}$ be a filtering process of Bernoulli random variables $b_\ell \in \{0,1\}$ such that $b_\ell$ is $\Sigma(x_1, a_1, b_1, r_1, \cdots, x_{\ell-1}, a_{\ell-1}, b_{\ell-1}, r_{\ell-1}, x_\ell, a_\ell)$-measurable and $f_t^{\mathbf{b}_t} = \operatorname{argmin}_{f \in \mathcal{F}} \sum_{\ell=1}^{t-1} b_\ell \cdot (f(x_\ell, a_\ell) - r_\ell)^2$ be the "filtered" least squares estimator. If Assumption 2.2 holds there exists a universal constant $C > 0$ such that the cumulative variance estimator $W_t^{\mathbf{b}_t} = \sum_{\ell=1}^{t-1} b_\ell \cdot (r_\ell - f_t^{\mathbf{b}_t}(x_\ell, a_\ell))^2$ satisfies,*

$$\frac{2}{3} \cdot W_t^{\mathbf{b}_t} - C \cdot B^2 \log(t|\mathcal{F}|/\delta') \leqslant \overline{W}_t^{\mathbf{b}_t} \leqslant 2W_t^{\mathbf{b}_t} + C \cdot B^2 \log(t|\mathcal{F}|/\delta')$$

*with probability at least $1 - \delta'$ for all $t \in \mathbb{N}$.*

The proof of Corollary 4.3 can be found in Appendix E.1.

## 4.2 Unknown-Variance Guarantees for Algorithm 1

Although Algorithm 1 was formulated under the assumption of a known variance upper bound $\sigma$, in this section we show it is possible to combine the variance estimation procedure we propose here with Algorithm 1. A simple and immediate consequence of Corollary 4.3 is,

**Corollary 4.4.** *Let $\delta' \in (0,1)$. Under the assumptions of Corollary 4.3. If $\sigma_t = \sigma$ for all $t \in \mathbb{N}$ and we define $N_t^{\mathbf{b}_t} = \sum_{\ell=1}^{t-1} b_\ell$ then,*

$$\frac{2W_t^{\mathbf{b}_t}}{3N_t^{\mathbf{b}_t}} - \frac{C \cdot B^2 \log(t|\mathcal{F}|/\delta')}{N_t^{\mathbf{b}_t}} \leqslant \sigma^2 \leqslant \frac{2W_t^{\mathbf{b}_t}}{N_t^{\mathbf{b}_t}} + \frac{C \cdot B^2 \log(t|\mathcal{F}|/\delta')}{N_t^{\mathbf{b}_t}}$$

*with probability at least $1 - \delta'$ for all $t \in \mathbb{N}$. Where $C > 0$ is the same universal constant as in Corollary 4.3.*

Let $\{\mathbf{b}_t\}_{t\in\mathbb{N}}$ be the trivial filtering process defined by setting $b_\ell = 1$ for $\ell \leqslant t - 1$ so that $N_t^{\mathbf{b}_t} = t - 1$ and define the variance upper-bound estimator sequence $\widehat{\sigma}_1^2 = B^2$ and $\widehat{\sigma}_t^2 = \min\left(\widehat{\sigma}_{t-1}^2, \frac{2W_t^{\mathbf{b}_t}}{t-1} + \frac{C \cdot B^2 \log(t|\mathcal{F}|/\delta')}{t-1}\right)$ for all $t \geqslant 2$. Corollary 4.4 implies that

$$\sigma^2 \leqslant \widehat{\sigma}_t \leqslant \frac{2W_t^{\mathbf{b}_t}}{t-1} + \frac{C \cdot B^2 \log(t|\mathcal{F}|/\delta')}{t-1} \leqslant 3\sigma^2 + \frac{3C \cdot B^2 \log(t|\mathcal{F}|/\delta')}{t-1} \tag{5}$$

for all $t \in \mathbb{N}$ with probability at least $1 - \delta'$.

We'll analyze a version of Algorithm 1 where the confidence sets $\mathcal{G}_t(\tau_i)$ are computed using confidence radii equal to $\beta_t\left(\tau_i, \frac{\delta}{2(i+1)^2}, \widehat{\sigma}_t^2\right)$ in equation 2. With these parameter choices, we can show a result equivalent to Lemma 3.2,

**Corollary 4.5.** *Let $\delta \in (0, 1)$. If $\delta/2$ is the input to Algorithm 1 and $\widehat{\sigma}_t^2$ estimators are computed by setting $\delta' = \delta/2$, then the confidence intervals are valid so that $f_\star \in \mathcal{G}_t$ for all $t \in \mathbb{N}$ and optimism holds, $\max_{a\in\mathcal{A}} f_\star(x_t, a) \leqslant U_t(x_t, a_t)$ with probability at least $1 - \delta$ for all $t \in \mathbb{N}$.*

The proof of this result follows by a simple union bound between the result of Lemma 3.2 and the inequality $\sigma^2 \leqslant \widehat{\sigma}_t^2$. Let $\bar{\mathcal{E}}$ denote the event described in Corollary 4.5. This version of Algorithm 1 satisfies the following guarantees,

**Theorem 4.6.** *Let $\delta \in (0, 1)$. If $\delta/2$ is the input to Algorithm 1 and $\widehat{\sigma}_t^2$ estimators are computed by setting $\delta' = \delta/2$. If $\sigma_t = \sigma$ for all $t \in \mathbb{N}$ the regret of Algorithm 1 with modified confidence set sizes satisfies,*

$$\text{Regret}(T) \leqslant \mathcal{O}\left(\sigma\sqrt{d_{\text{eluder}}(\mathcal{F}, B/T)\log(T|\mathcal{F}|/\delta)T} + Bd_{\text{eluder}}(\mathcal{F}, B/T)\log^2(T)\log(T|\mathcal{F}|/\delta)\right).$$

*for all $T \in \mathbb{N}$ with probability at least $1 - \delta$.*

The proof of Theorem 4.6 can be found in Appendix E.2.

## 4.3 UNKNOWN-VARIANCE DEPENDENT LEAST SQUARES REGRESSION

We borrow the setting of Section 3 with a few modifications. Given a data stream $\{(x_\ell, a_\ell, r_\ell)\}_{\ell\in\mathbb{N}}$ where $r_\ell = f_\star(x_\ell, a_\ell) + \xi_\ell$ for $f_\star \in \mathcal{F}$ such that $\xi_\ell$ is conditionally zero mean, a sequence of subsets of $\mathcal{G}_t' \subseteq \cdots \mathcal{G}_2' \subseteq \mathcal{G}_1' = \mathcal{F}$ such that $\mathcal{G}_t'$ is a function of $\{(x_\ell, a_\ell, r_\ell)\}_{\ell=1}^{t-1}$, and $f_\star \in \mathcal{G}_t'$ for all $t \in \mathbb{N}$. Given $\tau > 0$ we define an uncertainty-filtered least squares objective,

$$f_t^{(\tau, 2\tau]} = \operatorname*{argmin}_{f\in\mathcal{G}_{t-1}} \sum_{\ell=1}^{t-1} (f(x_\ell, a_\ell) - r_\ell)^2 \mathbf{1}\left(\omega(x_\ell, a_\ell, \mathcal{G}_\ell') \in (\tau, 2\tau]\right). \tag{6}$$

in the following, for any $\tau$ we'll use the notation $b_\ell^\tau$ to denote the filtering random variables $b_\ell^\tau = \mathbf{1}(\omega(x_\ell, a_\ell, \mathcal{G}_\ell') \in (\tau, 2\tau])$. Similarly, we denote by $\mathbf{b}_t^\tau = (b_1^\tau, \cdots, b_{t-1}^\tau)$.

We develop a result equivalent to Lemma B.1 and Proposition 3.1 to characterize the confidence sets. Just like in Section 3 we write our result in terms of a sequence of events $\{\mathcal{E}_\ell'\}_{\ell=1}^\infty$ such that $\mathcal{E}_\ell'$ corresponds to the event that $f_\star \in \mathcal{G}_{\ell-1}'$ and therefore $f_\star \in \mathcal{G}_{\ell'}'$ for all $\ell' \leqslant \ell - 1$ so that $\mathcal{E}_\ell \subseteq \mathcal{E}_\ell'$ for all $\ell \geqslant \ell'$.

**Proposition 4.7.** *Let $\widetilde{\delta} \in (0, 1)$ and $\tau > 0$. Let $\{\widetilde{\mathcal{E}}_\ell'\}_{\ell=1}^\infty$ be a sequence of events such that $\widetilde{\mathcal{E}}_1' \supseteq \widetilde{\mathcal{E}}_2' \cdots$ and $\widetilde{\mathcal{E}}_t' \subseteq \mathcal{E}_t'$. Let $f_t^{(\tau, 2\tau]}$ be result of solving the uncertainty-filtered least-squares objective from equation 6. Additionally let $W_t^{\mathbf{b}_t^\tau}$ be the filtered estimator of the cumulative variances defined by equation 4 when setting $b_\ell^\tau = \mathbf{1}(\omega(x_\ell, a_\ell, \mathcal{G}_\ell') \in (\tau, 2\tau])$ and $\overline{W}_t^{\mathbf{b}_t^\tau} := \sum_{\ell=1}^{t-1} \sigma_\ell^2 \cdot \mathbf{1}(\omega(x_\ell, a_\ell, \mathcal{G}_\ell') \in (\tau, 2\tau])$. There exist universal constants $C, C' > 0$ such that the events $\mathcal{W}_t(\tau)$ defined for any $t$ as*

$$\sum_{\ell=1}^{t-1} \left(f_t^{(\tau, 2\tau]}(x_\ell, a_\ell) - f_\star(x_\ell, a_\ell)\right)^2 \mathbf{1}(\omega(x_\ell, a_\ell, \mathcal{G}_\ell) \in (\tau, 2\tau])$$

$$\leqslant C'\tau\sqrt{W_t^{\mathbf{b}_t^\tau} \log\left(t|\mathcal{F}|/\widetilde{\delta}\right)} + C'\tau B \log\left(t|\mathcal{F}|/\widetilde{\delta}\right)$$

---

**Algorithm 2** Unknown-Variance Second Order Optimistic Least Squares

1: **Input:** probability parameter $\delta \in (0, 1)$, function class $\mathcal{F}$.
2: Set the initial confidence set $\mathcal{G}'_0 = \mathcal{F}$.
3: **for** $t = 1, 2, \cdots$ **do**
4:     Compute regression function for each threshold level $\tau_i = \frac{B}{2^i}$ for $i \in [q_t]$ where $q_t = \lceil \log(t) \rceil$

$$f_t^{(\tau_i, \tau_{i-1}]} = \underset{f \in \mathcal{G}'_{t-1}}{\arg\min} \sum_{\ell=1}^{t-1} (f(x_\ell, a_\ell) - r_\ell)^2 \mathbf{1}\left(\omega(x_\ell, a_\ell, \mathcal{G}'_\ell) \in (\tau_i, \tau_{i-1}]\right)$$

5:     Estimate the sum of the filtered variances for all threshold levels $\tau_i$ for $i \in [q_t]$.

$$W_t^{\mathbf{b}_t^{\tau_i}} = \sum_{\ell=1}^{t-1} \mathbf{1}\left(\omega(x_\ell, a_\ell, \mathcal{G}'_\ell) \in (\tau_i, \tau_{i-1}]\right) \cdot (r_\ell - f_t^{(\tau_i, \tau_{i-1}]}(x_\ell, a_\ell))^2.$$

6:     Compute threshold confidence sets for all $i \in [q_t]$,

$$\mathcal{G}'_t(\tau_i) = \left\{ f \in \mathcal{F} : \sum_{\ell=1}^{t-1} (f_t^\tau(x_\ell, a_\ell) - f(x_\ell, a_\ell))^2 \mathbf{1}(\omega(x_\ell, a_\ell, \mathcal{G}_\ell) \leqslant \tau_i) \leqslant C' \tau_i \sqrt{W_t^{\mathbf{b}_t^{\tau_i}} \log\left(2i^2 t|\mathcal{F}|/\delta\right)} \right.$$

$$\left. + C' \tau_i B \log\left(2i^2 t|\mathcal{F}|/\delta\right) \right\} \cap \mathcal{G}'_{t-1}(\tau_i)$$

    where $C' > 0$ is the constant from Proposition 4.7.
7:     Compute $\mathcal{G}'_t = \mathcal{G}'_{t-1} \cap (\cap_{i=1}^q \mathcal{G}'_t(\tau_i))$
8:     Receive context $x_t$.
9:     Compute $U_t(x_t, a) = \max_{f \in \mathcal{G}_t} f(x_t, a)$ for all $a \in \mathcal{A}$.
10:    play $a_t = \arg\max_{a \in \mathcal{A}} U_t(x_t, a)$ and receive $r_t = f_\star(x_t, a_t) + \xi_t$.
11: **end for**

---

*and*

$$C' \tau \sqrt{W_t^{\mathbf{b}_t^\tau} \log\left(t|\mathcal{F}|/\tilde{\delta}\right)} + C' \tau B \log\left(t|\mathcal{F}|/\tilde{\delta}\right) \leqslant C'' \tau \sqrt{\overline{W}_t^{\mathbf{b}_t^\tau} \log\left(t|\mathcal{F}|/\tilde{\delta}\right)} + C'' \tau B \log\left(t|\mathcal{F}|/\tilde{\delta}\right)$$

*satisfy the bound* $\mathbb{P}(\widetilde{\mathcal{E}}'_t \cap (\mathcal{W}_t(\tau))^c) \leqslant \frac{\tilde{\delta}}{2t^2}$.

The proof of Proposition 4.7 can be found in Appendix E.3. Similar to Corollary B.2 and Proposition 4.7 we derive the following anytime guarantee for the confidence sets, and show that optimism holds

**Lemma 4.8.** *Let* $\delta \in (0, 1)$. *When the confidence sets* $\mathcal{G}'_t \subseteq \mathcal{F}$ *are defined as in Algorithm 2, then* $f_\star \in \mathcal{G}'_t$ *so that* $\max_{a \in \mathcal{A}} f_\star(x_t, a) \leqslant U_t(x_t, a_t)$ *(optimism holds), and for all* $i \in [q_t]$,

$$\mathcal{G}'_t(\tau_i) \subseteq \left\{ f \in \mathcal{F} \text{ s.t. } \sum_{\ell=1}^{t-1} \left( f_t^{(\tau_i, 2\tau_i]}(x_\ell, a_\ell) - f(x_\ell, a_\ell) \right)^2 \mathbf{1}(\omega(x_\ell, a_\ell, \mathcal{G}_\ell) \in (\tau_i, 2\tau_i]) \leqslant \right.$$

$$\left. C'' \tau_i \sqrt{\overline{W}_t^{\mathbf{b}_t^{\tau_i}} \log\left(2(i+1)^2 t|\mathcal{F}|/\delta\right)} + C'' \tau_i B \log\left(2i^2 t|\mathcal{F}|/\delta\right) \right\}.$$

(7)

*with probability at least* $1 - \delta$ *for all* $t \in \mathbb{N}$. *Where* $C'' > 0$ *is the same universal constant as in Proposition 4.7.*

The proof of Lemma 4.8 can be found in Appendix E.3. Define $\mathcal{E}'$ as the event outlined in Lemma 4.8 such that $f_\star \in \mathcal{G}'_t$, optimism holds, and inequality 7 holds for all $i \in [q_t]$ and all $t \in \mathbb{N}$. This event satisfies $\mathbb{P}(\mathcal{E}') \geqslant 1 - \delta$. The proof of the regret guarantees of Algorithm 2 will follow a similar template as in the previous sections; establishing optimism and then bounding the sum of the uncertainty widths over the context-action pairs played by the algorithm. In order to execute this proof strategy we need a way to relate the sum of the uncertainty widths to the definition of

the confidence sets and by doing so with the true cumulative sum of variances. We do this via the following Lemma.

**Lemma 4.9.** *If $\{\mathcal{G}'_t\}_{t=1}^\infty$ is the sequence of confidence sets produced by Algorithm 1, there exists a universal constant $\widetilde{C} > 0$ such that when $\mathcal{E}'$ is satisfied,*

$$\sum_{t=1}^T \mathbf{1}(\omega(x_t, a_t, \mathcal{G}'_t) \in (\tau_i, 2\tau_i]) \leqslant$$

$$\frac{\widetilde{C} \cdot d_{\mathrm{eluder}}(\mathcal{F}, \tau_i)}{\tau_i} \sqrt{\overline{W}_T^{\mathbf{b}_T^{\tau_i}} \log{(iT|\mathcal{F}|/\delta)}} + \frac{\widetilde{C} \cdot B \cdot d_{\mathrm{eluder}}(\mathcal{F}, \tau_i)}{\tau_i} \log{(iT|\mathcal{F}|/\delta)} + \widetilde{C} \cdot d_{\mathrm{eluder}}(\mathcal{F}, \tau_i)$$

*for all $T \in \mathbb{N}$ and $i \in [q_T]$.*

The proof of Lemma 4.9 can be found in Appendix F. Finally, we can combine the result above with an optimism argument to prove the following regret bound for Algorithm 2.

**Theorem 4.10.** *Let $T \in \mathbb{N}$, $\delta \in (0,1)$ and $q = \log(T)$. The regret of Algorithm 2 satisfies,*

$$\mathrm{Regret}(T)$$

$$\leqslant \mathcal{O}\left( d_{\mathrm{eluder}}\left(\mathcal{F}, \frac{B}{T}\right) \sqrt{\left(\sum_{t=1}^T \sigma_t^2\right) \log{(T)} \log{(T|\mathcal{F}|/\delta)}} + B d_{\mathrm{eluder}}\left(\mathcal{F}, \frac{B}{T}\right) \log(T) \log(T|\mathcal{F}|/\delta) \right)$$

*simultaneously for all $T \in \mathbb{N}$ with probability at least $1 - \delta$.*

*Proof Sketch.* The proof of Theorem 4.10 relies on observation that when $\mathcal{E}'$ holds, optimism implies

$$\mathrm{Regret}(T) \leqslant \sum_{t=1}^T \omega(x_t, a_t, \mathcal{G}'_t).$$

The sum of widths can be upper bounded as,

$$\sum_{t=1}^T \omega(x_t, a_t, \mathcal{G}'_t) \leqslant \frac{T \cdot B}{2^q} + 2 \sum_{i=1}^q \tau_i \cdot \left( \sum_{t=1}^T \mathbf{1}\left(\omega(x_t, a_t, \mathcal{G}'_t) \in (\tau_i, 2\tau_i]\right)\right).$$

Finally Lemma 4.9 can be used to finish the proof. $\qquad\square$

## 5  CONCLUSION

In this work we have introduced second order bounds for contextual bandits with function approximation. These bounds improve on existing results such as Wang et al. (2024b) because they only require a realizability assumption on the mean reward values of each context-action pair. We introduce two types of algorithm, one that achieves what we believe is sharp dependence on the complexity of the underlying reward class measured by the eluder dimension when all the measurement noise variances are the same and unknown, and a second one that in the case of changing noise variances achieves a bound that scales with the square root of the sum of these variances but scales linearly in the eluder dimension. In a future version of this writeup we will strive to update our results to achieve a sharper dependence on the eluder dimension scaling with its square root. We hope the techniques we have developed in this manuscript can be easily used to develop second order algorithms with function approximation in other related learning models such as reinforcement learning. These techniques distill, simplify and present in a didactic manner many of the ideas developed for the variance aware literature in linear contextual bandit problems in works such as Kirschner & Krause (2018); Zhou et al. (2021); Kim et al. (2022); Zhao et al. (2023); Xu et al. (2024) and transports them to the setting of function approximation. Although we did not cover this in our work, an interesting avenue of future research remains to understand when can we design second order bounds for algorithms based on the inverse gap weighting technique that forms the basis of the SquareCB algorithm from Foster & Rakhlin (2020).

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

# CONTENTS

## A    SUPPORTING RESULTS

Our results relies on the following variant of Bernstein inequality for martingales, or Freedman's inequality Freedman (1975), as stated in e.g., Agarwal et al. (2014); Beygelzimer et al. (2011).

**Lemma A.1** (Simplified Freedman's inequality). *Let $Z_1, ..., Z_T$ be a bounded martingale difference sequence with $|Z_\ell| \leqslant R$. For any $\delta' \in (0, 1)$, and $\eta \in (0, 1/R)$, with probability at least $1 - \delta'$,*

$$\sum_{\ell=1}^{T} Z_\ell \leqslant \eta \sum_{\ell=1}^{T} \mathbb{E}_\ell[Z_\ell^2] + \frac{\log(1/\delta')}{\eta}. \tag{8}$$

*where $\mathbb{E}_\ell[\cdot]$ is the conditional expectation[2] induced by conditioning on $Z_1, \cdots, Z_{\ell-1}$.*

**Lemma A.2** (Anytime Freedman). *Let $\{Z_t\}_{t=1}^{\infty}$ be a bounded martingale difference sequence with $|Z_t| \leqslant R$ for all $t \in \mathbb{N}$. For any $\delta' \in (0, 1)$, and $\eta \in (0, 1/R)$, there exists a universal constant $C > 0$ such that for all $t \in \mathbb{N}$ simultaneously with probability at least $1 - \delta'$,*

$$\sum_{\ell=1}^{t} Z_\ell \leqslant \eta \sum_{\ell=1}^{t} \mathbb{E}_\ell[Z_\ell^2] + \frac{C \log(t/\delta')}{\eta}. \tag{9}$$

*where $\mathbb{E}_\ell[\cdot]$ is the conditional expectation induced by conditioning on $Z_1, \cdots, Z_{\ell-1}$.*

---

[2]We will use this notation to denote conditional expectations throughout this work.

*Proof.* This result follows from Lemma A.1. Fix a time-index $t$ and define $\delta_t = \frac{\delta'}{12t^2}$. Lemma A.1 implies that with probability at least $1 - \delta_t$,

$$\sum_{\ell=1}^{t} Z_\ell \leqslant \eta \sum_{\ell=1}^{t} \mathbb{E}_\ell\left[Z_\ell^2\right] + \frac{\log(1/\delta_t)}{\eta}.$$

A union bound implies that with probability at least $1 - \sum_{\ell=1}^{t} \delta_t \geqslant 1 - \delta'$,

$$\sum_{\ell=1}^{t} Z_\ell \leqslant \eta \sum_{\ell=1}^{t} \mathbb{E}_\ell\left[Z_\ell^2\right] + \frac{\log(12t^2/\delta')}{\eta}$$

$$\overset{(i)}{\leqslant} \eta \sum_{\ell=1}^{t} \mathbb{E}_\ell\left[Z_\ell^2\right] + \frac{C\log(t/\delta')}{\eta}.$$

holds for all $t \in \mathbb{N}$. Inequality $(i)$ holds because $\log(12t^2/\delta') = \mathcal{O}\left(\log(t\delta')\right)$.

$\square$

**Lemma A.3** (Uniform empirical Bernstein bound). *In the terminology of Howard et al. (2021), let $S_t = \sum_{i=1}^{t} Y_i$ be a sub-$\psi_P$ process with parameter $c > 0$ and variance process $W_t$. Then with probability at least $1 - \widetilde{\delta}$ for all $t \in \mathbb{N}$*

$$S_t \leqslant 1.44\sqrt{\max(W_t, m)\left(1.4\ln\ln\left(2\left(\max\left(\frac{W_t}{m}, 1\right)\right)\right) + \ln\frac{5.2}{\widetilde{\delta}}\right)}$$
$$+ 0.41c\left(1.4\ln\ln\left(2\left(\max\left(\frac{W_t}{m}, 1\right)\right)\right) + \ln\frac{5.2}{\widetilde{\delta}}\right)$$

*where $m > 0$ is arbitrary but fixed.*

As a corollary of Lemma A.3 we can show the following,

**Lemma A.4** (Freedman). *Suppose $\{X_t\}_{t=1}^{\infty}$ is an adapted process with $|X_t| \leqslant b$. Let $V_t = \sum_{\ell=1}^{t} \mathrm{Var}_\ell$ where $\mathrm{Var}_\ell = \mathbb{E}_\ell[X_\ell^2] - \mathbb{E}_\ell^2[X_\ell]$. For any $\widetilde{\delta} \in (0, 1)$, with probability at least $1 - \widetilde{\delta}$,*

$$\sum_{\ell=1}^{t} X_\ell - \mathbb{E}_\ell[X_\ell] \leqslant 4\sqrt{V_t \ln\frac{12\ln 2t}{\widetilde{\delta}}} + 6b\ln\frac{12\ln 2t}{\widetilde{\delta}}.$$

*for all $t \in \mathbb{N}$ simultaneously.*

*Proof.* We are ready to use Lemma A.3 (with $c = b$). Let $S_t = \sum_{\ell=1}^{t} X_t$ and $W_t = \sum_{\ell=1}^{t} \mathrm{Var}_\ell(X_\ell)$. Let's set $m = b^2$. It follows that with probability $1 - \widetilde{\delta}$ for all $t \in \mathbb{N}$

$$S_t \leqslant 1.44\sqrt{\max(W_t, b^2)\left(1.4\ln\ln\left(2\left(\max\left(\frac{W_t}{b^2}, 1\right)\right)\right) + \ln\frac{5.2}{\widetilde{\delta}}\right)}$$
$$+ 0.41b\left(1.4\ln\ln\left(2\left(\max\left(\frac{W_t}{b}, 1\right)\right)\right) + \ln\frac{5.2}{\widetilde{\delta}}\right)$$
$$\leqslant 2\sqrt{\max(W_t, b^2)\left(2\ln\ln\left(2\left(\max\left(\frac{W_t}{b^2}, 1\right)\right)\right) + \ln\frac{6}{\widetilde{\delta}}\right)}$$
$$+ b\left(2\ln\ln\left(2\left(\max\left(\frac{W_t}{b^2}, 1\right)\right)\right) + \ln\frac{6}{\widetilde{\delta}}\right)$$
$$= 2\max(\sqrt{W_t}, b)A_t + bA_t^2$$
$$\leqslant 2\sqrt{W_t}A_t + 2bA_t + bA_t^2$$
$$\overset{(i)}{\leqslant} 2\sqrt{W_t}A_t + 3bA_t^2,$$

where $A_t = \sqrt{2\ln\ln\left(2\left(\max\left(\frac{W_t}{b^2}, 1\right)\right)\right) + \ln\frac{6}{\delta}}$. Inequality $(i)$ follows because $A_t \geqslant 1$. By identifying $V_t = W_t$ we conclude that for any $\widetilde{\delta} \in (0, 1)$ and $t \in \mathbb{N}$

$$\mathbb{P}\left(\sum_{\ell=1}^{t} X_\ell > 2\sqrt{V_t}A_t + 3bA_t^2\right) \leqslant \widetilde{\delta}$$

Where $A_t = \sqrt{2\ln\ln\left(2\left(\max\left(\frac{V_t}{b^2}, 1\right)\right)\right) + \ln\frac{6}{\delta}}$. Since $V_t \leqslant tb^2$ with probability 1,

$$\frac{V_t}{b^2} \leqslant t,$$

And therefore $2\ln\ln\left(2\max(\frac{V_t}{b^2}, 1)\right) \leqslant 2\ln\ln 2t$ implying,

$$A_t \leqslant \sqrt{2\ln\frac{12\ln t}{\widetilde{\delta}}}$$

Thus

$$\mathbb{P}\left(\sum_{\ell=1}^{t} X_\ell > 4\sqrt{V_t\ln\frac{12\ln 2t}{\widetilde{\delta}}}A + 6b\ln\frac{12\ln 2t}{\widetilde{\delta}}\right) \leqslant \widetilde{\delta}$$

Since $V_t \leqslant S_t$ the result follows. $\qquad\square$

**Proposition A.5.** *Let $\delta' \in (0, 1)$, $\beta \in (0, 1]$ and $\{Z_\ell\}_{\ell=1}^{\infty}$ be an adapted sequence satisfying $0 \leqslant Z_\ell \leqslant \tilde{B}$ for all $\ell \in \mathbb{N}$. It follows that,*

$$(1 - \beta)\sum_{\ell=1}^{t}\mathbb{E}_\ell[Z_\ell] - \frac{2\tilde{B}\log(1/\delta')}{\beta} \leqslant \sum_{\ell=1}^{t} Z_\ell \leqslant (1 + \beta)\sum_{\ell=1}^{t}\mathbb{E}_\ell[Z_\ell] + \frac{2\tilde{B}\log(1/\delta')}{\beta}$$

*with probability at least $1 - 2\delta'$.*

*Proof.* Consider the martingale difference sequence $X_t = Z_t - \mathbb{E}_t[Z_t]$. Notice that $|X_t| \leqslant \tilde{B}$. Using the inequality of Lemma A.1 we obtain for all $\eta \in (0, 1/B^2)$.

$$\sum_{\ell=1}^{t} X_\ell \leqslant \eta\sum_{\ell=1}^{t}\mathbb{E}_\ell[X_\ell^2] + \frac{\log(1/\delta')}{\eta}$$

$$\stackrel{(i)}{\leqslant} 2\eta B^2\sum_{\ell=1}^{t}\mathbb{E}_\ell[Z_\ell] + \frac{\log(1/\delta')}{\eta}$$

with probability at least $1 - \delta'$. Inequality $(i)$ holds because $\mathbb{E}_t[X_t^2] \leqslant B^2\mathbb{E}[|X_t|] \leqslant 2B^2\mathbb{E}_t[Z_t]$ for all $t \in \mathbb{N}$. Setting $\eta = \frac{\beta}{2B^2}$ and substituting $\sum_{\ell=1}^{t} X_\ell = \sum_{\ell=1}^{t} Z_\ell - \mathbb{E}_\ell[Z_\ell]$,

$$\sum_{\ell=1}^{t} Z_\ell \leqslant (1 + \beta)\sum_{\ell=1}^{t}\mathbb{E}_\ell[Z_\ell] + \frac{2B^2\log(1/\delta')}{\beta} \tag{10}$$

with probability at least $1 - \delta'$. Now consider the martingale difference sequence $X_t' = \mathbb{E}[Z_t] - Z_t$ and notice that $|X_t'| \leqslant B^2$. Using the inequality of Lemma A.1 we obtain for all $\eta \in (0, 1/B^2)$,

$$\sum_{\ell=1}^{t} X_\ell' \leqslant \eta\sum_{\ell=1}^{t}\mathbb{E}_\ell[(X_\ell')^2] + \frac{\log(1/\delta')}{\eta}$$

$$\leqslant 2\eta B^2\sum_{\ell=1}^{t}\mathbb{E}_\ell[Z_\ell] + \frac{\log(1/\delta')}{\eta}$$

Setting $\eta = \frac{\beta}{2B^2}$ and substituting $\sum_{\ell=1}^{t} X_\ell' = \sum_{\ell=1}^{t}\mathbb{E}[Z_\ell] - Z_\ell$ we have,

$$(1 - \beta) \sum_{\ell=1}^{t} \mathbb{E}[Z_\ell] \leqslant \sum_{\ell=1}^{t} Z_\ell + \frac{2B^2 \log(1/\delta')}{\beta} \tag{11}$$

with probability at least $1 - \delta'$. Combining Equations 10 and 11 and using a union bound yields the desired result.

$\square$

## B  OPTIMISTIC LEAST SQUARES

The algorithms we propose in this work are based on the optimism principle. This simple yet powerful algorithmic idea is the basis of a celebrated algorithm for contextual bandits with function approximation known as Optimistic Least Squares. Algorithm 3 presents the pseudo-code of the Optimistic Least Squares algorithm.

---

**Algorithm 3** Optimistic Least Squares

1: **Input:** Function class $\mathcal{F}$, confidence radius functions $\{\beta_t : [0, 1] \to \mathbb{R}\}_{t=1}^{\infty}$.
2: **for** $t = 1, 2, \cdots$ **do**
3:    Compute least squares regression

$$f_t = \underset{f \in \mathcal{F}}{\operatorname{argmin}} \sum_{\ell=1}^{t-1} (f(x_\ell, a_\ell) - r_\ell)^2 . \tag{12}$$

4:    Compute confidence set[3],

$$\mathcal{G}_t = \left\{ f \in \mathcal{F} : \sum_{\ell=1}^{t-1} (f_t(x_\ell, a_\ell) - f(x_\ell, a_\ell))^2 \leqslant \beta_t(\delta) \right\}$$

5:    Receive context $x_t$.
6:    Compute $U_t(x_t, a) = \max_{f \in \mathcal{G}_t} f(x_t, a)$ for all $a \in \mathcal{A}$.
7:    play $a_t = \operatorname{argmax}_{a \in \mathcal{A}} U_t(x_t, a)$ and receive $r_t = f_\star(x_t, a_t) + \xi_t$.
8: **end for**

---

To derive a bound for the optimistic least squares algorithm, we require guarantees for the confidence sets. This is captured by the following Lemma.

**Lemma B.1.** *[LS guarantee] Let $\delta \in (0, 1)$, $\{x_t, a_t\}_{t=1}^{\infty}$ be a sequence of context-action pairs and and $\{r_t\}_{t=1}^{\infty}$ be a sequence of reward values satisfying $r_t = f_\star(x_t, a_t) + \xi_t$ where $f_\star \in \mathcal{F}$ and $\xi_t$ is conditionally zero mean. Let $f_t = \arg\min_{f \in \mathcal{F}} \sum_{\ell=1}^{t-1} (f(x_\ell, a_\ell) - r_\ell)^2$ be the least squares fit. If Assumption 2.2 holds then there is a constant $C > 0$ such that,*

$$\sum_{\ell=1}^{t-1} (f_t(x_\ell, a_\ell) - f_\star(x_\ell, a_\ell))^2 \leqslant \beta_t(\delta)$$

*with probability at least $1 - \delta$ for all $t \in \mathbb{N}$ where $\beta_t(\delta) = 4CB^2 \log(t \cdot |\mathcal{F}|/\delta)$.*

The proof of Lemma B.1 can be found in Section C. This result allows provides us with the tools to justify the choice of confidence sets in Algorithm 3. A simple corollary is,

**Corollary B.2.** *Let $\delta \in (0, 1)$ and $\beta_t(\delta) = 4CB^2 \log(t \cdot |\mathcal{F}|/\delta)$ as defined in Lemma B.1. The confidence sets $\mathcal{G}_t$ satisfy $f_\star \in \mathcal{G}_t$ for all $t \in \mathbb{N}$ simultaneously with probability at least $1 - \delta$. Moreover, this property holds, $U_t(x, a_t) \geqslant \max_{a \in \mathcal{A}} f_\star(x, a)$ for all $t \in \mathbb{N}$.*

The proof of this result can be found in Appendix C. In order to relate the scale of these confidence sets with the algorithm's regret we need to tie these values to the statistical capacity of the function class $\mathcal{F}$. This can be captured by its eluder dimension (see Definition 2.2). This is done via Lemma B.3, a standard result that is crucial in showing an upper bound for the optimistic least squares algorithm

regret. This result is a version of Lemma 3 from (Chan et al., 2021) presented as Lemma 4.3 in (Pacchiano et al., 2024) which we reproduce here for readability.

**Lemma B.3.** *Let $\mathcal{F}$ be a function class satisfying Assumption 2.2 and with $\epsilon$-eluder dimension $\bar{d}_{\mathrm{eluder}}(\mathcal{F}, \epsilon)$. For all $T \in \mathbb{N}$ and any dataset sequence $\{\bar{D}_t\}_{t=1}^{\infty}$ for $\bar{D}_1 = \varnothing$ and $\bar{D}_t = \{(\bar{x}_\ell, \bar{a}_\ell)\}_{\ell=1}^{t-1}$ of context-action pairs, the following inequality on the sum of the uncertainty radii holds,*

$$\sum_{t=1}^{T} \omega(\bar{x}_t, \bar{a}_t, \bar{D}_t) \leqslant \mathcal{O}\left( \min\left( BT, \sqrt{\beta_t(\delta) \cdot \bar{d}_{\mathrm{eluder}}(\mathcal{F}, B/T) \cdot T} + B\bar{d}_{\mathrm{eluder}}(\mathcal{F}, B/T) \right) \right)$$

Lemmas B.1 and B.3 can be used to prove Algorithm 3 satisfies the following regret guarantee,

**Theorem B.4.** *The regret of the Optimistic Least Squares (Algorithm 3) with input values $\delta \in (0, 1)$ and $\beta_t(\delta) = 4CB^2 \log(t|\mathcal{F}|/\delta)$ satisfies,*

$$\mathrm{Regret}(T) \leqslant \mathcal{O}\left( B\sqrt{d_{\mathrm{eluder}}(\mathcal{F}, B/T) \cdot T \cdot \log(T|\mathcal{F}|/\delta)} + Bd_{\mathrm{eluder}}(\mathcal{F}, B/T) \right)$$

*with probability at least $1 - \delta$ for all $T \in \mathbb{N}$ simultaneously.*

The proof of this result can be found in Appendix C. Thus the dominating term of the regret bound (the term growing at a $\sqrt{T}$ rate) for optimistic least squares scales with the square root of the uncertainty radius, in this case given by the function $\beta_T(\delta) = 4CB^2 \log(T|\mathcal{F}|/\delta)$ defined in Lemma B.1. Unfortunately, this introduces an unavoidable dependence on $B$. Thus, the dominating term of our regret bound has a scale controlled by $B$ instead of the variances $\{\sigma_\ell^2\}_{\ell=1}^T$. This dependence comes up because the proof of Lemma B.1 relies on Freedman's inequality (Lemma A.1 in Appendix A) that exhibits an unavoidable dependence on the scale of the random variables in the low order terms. In the following section we show a way to bypass this issue by introducing a multi-bucket regression approach that has a vanishing dependence on the low order terms.

## C    PROOFS OF SECTION B

**Lemma B.1.** *[LS guarantee] Let $\delta \in (0, 1)$, $\{x_t, a_t\}_{t=1}^{\infty}$ be a sequence of context-action pairs and and $\{r_t\}_{t=1}^{\infty}$ be a sequence of reward values satisfying $r_t = f_\star(x_t, a_t) + \xi_t$ where $f_\star \in \mathcal{F}$ and $\xi_t$ is conditionally zero mean. Let $f_t = \arg\min_{f \in \mathcal{F}} \sum_{\ell=1}^{t-1} (f(x_\ell, a_\ell) - r_\ell)^2$ be the least squares fit. If Assumption 2.2 holds then there is a constant $C > 0$ such that,*

$$\sum_{\ell=1}^{t-1} (f_t(x_\ell, a_\ell) - f_\star(x_\ell, a_\ell))^2 \leqslant \beta_t(\delta)$$

*with probability at least $1 - \delta$ for all $t \in \mathbb{N}$ where $\beta_t(\delta) = 4CB^2 \log(t \cdot |\mathcal{F}|/\delta)$.*

*Proof.* Substituting $r_\ell = f_\star(x_\ell, a_\ell) + \xi_\ell$ into the definition of $f_t$ we obtain the following inequalities,

$$\sum_{\ell=1}^{t-1} (f_t(x_\ell, a_\ell) - r_\ell)^2 \leqslant \sum_{\ell=1}^{t-1} (f_\star(x_\ell, a_\ell) - r_\ell)^2 = \sum_{\ell=1}^{t-1} \xi_\ell^2$$

substituting again the definition of $r_\ell$ on the left hand side of the inequality above and rearranging terms yields,

$$\sum_{\ell=1}^{t-1} (f_t(x_\ell, a_\ell) - f_\star(x_\ell, a_\ell))^2 \leqslant 2 \sum_{\ell=1}^{t-1} \xi_\ell \cdot (f_\star(x_\ell, a_\ell) - f_t(x_\ell, a_\ell)) \tag{13}$$

We now focus on bounding the RHS of equation 13. For any $f \in \mathcal{F}$ let $Z_\ell^f = \xi_\ell \cdot (f_\star(x_\ell, a_\ell) - f(x_\ell, a_\ell))$. The sequence $Z_t$ forms a martingale difference sequence such that $\mathbb{E}_\ell\left[ \left(Z_\ell^f\right)^2 \right] = \sigma_\ell^2 \cdot (f(x_\ell, a_\ell) - f_\star(x_\ell, a_\ell))^2$ and Assumption 2.2 implies $|Z_\ell^f| \leqslant B^2$ for all $\ell \in \mathbb{N}$.

We can use Freedman inequality (see for example Lemma A.2 in Appendix A) to bound this term and show that with probability at least $1 - \delta'$ for all $t \in \mathbb{N}$,

$$\sum_{\ell=1}^{t-1} \xi_\ell \cdot (f(x_\ell, a_\ell) - f_\star(x_\ell, a_\ell)) \leqslant \eta \cdot \left( \sum_{\ell=1}^{t-1} \sigma_\ell^2 \cdot (f(x_\ell, a_\ell) - f_\star(x_\ell, a_\ell))^2 \right) + \frac{C \log(t/\delta')}{\eta}$$

$$\overset{(i)}{\leqslant} \frac{1}{4} \sum_{\ell=1}^{t-1} (f(x_\ell, a_\ell) - f_\star(x_\ell, a_\ell))^2 + 4CB^2 \log(t/\delta').$$

Where inequality $(i)$ follows from setting $\eta = \frac{1}{4B^2}$ and noting that $\sigma_\ell \leqslant B^2$ for all $\ell$. Finally, setting $\delta' = \frac{\delta}{|\mathcal{F}|}$ and considering a union bound over all $f \in \mathcal{F}$ we conclude that,

$$\sum_{\ell=1}^{t-1} \xi_\ell \cdot (f_\star(x_\ell, a_\ell) - f_t(x_\ell, a_\ell)) \leqslant \frac{1}{4} \sum_{\ell=1}^{t-1} (f_t(x_\ell, a_\ell) - f_\star(x_\ell, a_\ell))^2 + 4CB^2 \log(t \cdot |\mathcal{F}|/\delta).$$

Plugging this inequality into equation 13 and rearranging terms yields,

$$\sum_{\ell=1}^{t-1} (f_t(x_\ell, a_\ell) - f_\star(x_\ell, a_\ell))^2 \leqslant 4CB^2 \log(t \cdot |\mathcal{F}|/\delta).$$

$\square$

**Corollary B.2.** *Let $\delta \in (0, 1)$ and $\beta_t(\delta) = 4CB^2 \log(t \cdot |\mathcal{F}|/\delta)$ as defined in Lemma B.1. The confidence sets $\mathcal{G}_t$ satisfy $f_\star \in \mathcal{G}_t$ for all $t \in \mathbb{N}$ simultaneously with probability at least $1 - \delta$. Moreover, this property holds, $U_t(x, a_t) \geqslant \max_{a \in \mathcal{A}} f_\star(x_t, a)$ for all $t \in \mathbb{N}$.*

*Proof.* Lemma B.1 implies that $f_\star \in \mathcal{G}_t$ for all $t \in \mathbb{N}$ simultaneously with probability at least $1 - \delta$. When this occurs, the following sequence of inequalities is satisfied,

$$f_\star(x_t, a) \leqslant \max_{f \in \mathcal{G}_t} f(x_t, a) = U_t(x_t, a) \leqslant U_t(x_t, a_t).$$

for all $a \in \mathcal{A}$. Thus it holds that $\max_{a \in \mathcal{A}} f_\star(x_t, a) \leqslant U_t(x_t, a_t)$. $\square$

**Theorem B.4.** *The regret of the Optimistic Least Squares (Algorithm 3) with input values $\delta \in (0, 1)$ and $\beta_t(\delta) = 4CB^2 \log(t|\mathcal{F}|/\delta)$ satisfies,*

$$\text{Regret}(T) \leqslant \mathcal{O}\left( B\sqrt{d_{\text{eluder}}(\mathcal{F}, B/T) \cdot T \cdot \log(T|\mathcal{F}|/\delta)} + Bd_{\text{eluder}}(\mathcal{F}, B/T) \right)$$

*with probability at least $1 - \delta$ for all $T \in \mathbb{N}$ simultaneously.*

*Proof.* Lemma 3 implies the event $\mathcal{E}$ where $f_\star \in \mathcal{G}_t$ for all $t \in \mathbb{N}$ occurs with probability at least $1 - \delta$. The analysis of the regret of Algorithm 3 follows the typical analysis for optimistic algorithms,

$$\text{Regret}(T) = \sum_{t=1}^{T} \max_{a \in \mathcal{A}} f_\star(x_t, a) - f_\star(x_t, a_t)$$

$$\leqslant \sum_{t=1}^{T} U_t(x_t, a_t) - f_\star(x_t, a_t)$$

$$= \sum_{t=1}^{T} f_t(x_t, a_t) - f_\star(x_t, a_t)$$

$$\overset{(i)}{\leqslant} \sum_{t=1}^{T} \max_{f, f' \in \mathcal{G}_t} f(x_t, a_t) - f'(x_t, a_t)$$

$$= \sum_{t=1}^{T} \omega(x_t, a_t, \mathcal{G}_t)$$

$$\overset{(ii)}{\leqslant} \mathcal{O}\left( Bd_{\text{eluder}}(\mathcal{F}, B/T) + \sqrt{\beta_T(\delta) \cdot d_{\text{eluder}}(\mathcal{F}, B/T) \cdot T} \right)$$

where $f_t$ is the function that achieves the argmax in the definition of $U_t$ over input context $x_t$. Inequality $(i)$ holds because when $\mathcal{E}$ holds, $f_\star \in \mathcal{G}_t$ and $f_t \in \mathcal{G}_t$ for all $t \in \mathbb{N}$. Inequality $(ii)$ is a variation of Lemma 3 in (Chan et al., 2021) (see a simplified version in Lemma B.3 from Appendix F). Substituting $\beta_T(\delta) = 4CB^2 \log(T|\mathcal{F}|/\delta)$ finalizes the result.

$\square$

## D   Proofs of Section 3

**Proposition 3.1.** *[Variance Dependent Least Squares] Let $t \in \mathbb{N}, \tau \geqslant 0$ and $\tilde{\delta} > 0$. If $\sigma_\ell^2 \leqslant \tilde{\sigma}^2$ for all $\ell \leqslant t - 1$ and $\mathcal{E}_t$ holds then*

$$\mathbb{P}\left(\sum_{\ell=1}^{t-1}(f_t^\tau(x_\ell, a_\ell) - f_\star(x_\ell, a_\ell))^2 \, \mathbf{1}(\omega(x_\ell, a_\ell, \mathcal{G}_\ell) \leqslant \tau) \leqslant \beta_t(\tau, \tilde{\delta}, \tilde{\sigma}^2), \ \mathcal{E}_t\right) \geqslant \mathbb{P}(\mathcal{E}_t) - \tilde{\delta}. \quad (3)$$

*Proof.* Given $f \in \mathcal{F}$ we consider a martingale difference sequence $Z_\ell^f$ for $\ell \in \mathbb{N}$ defined as,

$$Z_\ell^f = (f(x_\ell, a_\ell) - f_\star(x_\ell, a_\ell)) \cdot \mathbf{1}(\omega(x_\ell, a_\ell, \mathcal{G}_\ell) \leqslant \tau) \cdot \mathbf{1}(f \in \mathcal{G}_\ell) \cdot \xi_\ell$$

First let's see that

$$|Z_\ell^f| \leqslant \min(\tau B, B^2) \quad \forall \ell \in \mathbb{N}.$$

To see this we recognize two cases, first when $f \notin \mathcal{G}_\ell$ in which case $Z_\ell^\tau = 0$. When $f \in \mathcal{G}_\ell$, we also recognize two cases. When $\mathbf{1}(\omega(x_\ell, a_\ell, \mathcal{G}_\ell) \leqslant \tau) = 0$ the random variable $Z_\ell^f = 0$. When $f \in \mathcal{G}_\ell$, and $\omega(x_\ell, a_\ell, \mathcal{G}_\ell) \leqslant \tau$, it follows that $|f(x_\ell, a_\ell) - f_\star(x_\ell, a_\ell)| \leqslant \tau$. Finally since $|\xi_\ell| \leqslant B$ we conclude $|Z_\ell^f| \leqslant \min(\tau B, B^2)$.

The conditional variance of the martingale difference sequence $\{Z_\ell^f\}_\ell$ can be upper bounded as

$$\begin{aligned}
\mathrm{Var}_\ell(Z_\ell^f) &= \mathbb{E}_\ell[(Z_\ell^f)^2] \\
&= \sigma_\ell^2 (f(x_\ell, a_\ell) - f_\star(x_\ell, a_\ell))^2 \mathbf{1}(\omega(x_\ell, a_\ell, \mathcal{G}_\ell) \leqslant \tau) \cdot \mathbf{1}(f \in \mathcal{G}_\ell) \\
&\stackrel{(i)}{\leqslant} \tilde{\sigma}^2 (f(x_\ell, a_\ell) - f_\star(x_\ell, a_\ell))^2 \mathbf{1}(\omega(x_\ell, a_\ell, \mathcal{G}_\ell) \leqslant \tau) \cdot \mathbf{1}(f \in \mathcal{G}_\ell)
\end{aligned}$$

where inequality $(i)$ follows because of $\sigma_\ell^2 \leqslant \tilde{\sigma}^2$.

We now invoke Lemma A.1 applied to the martingale difference sequence $\{Z_\ell^f\}_{\ell=1}^{t-1}$. In this case $R = \tau B$ and we'll set $\eta = \frac{1}{\min(\tau B, B^2) + 4\tilde{\sigma}^2} \leqslant \frac{1}{R}$. Thus,

$$\begin{aligned}
\sum_{\ell=1}^{t-1} Z_\ell^f &\leqslant \frac{1}{\min(\tau B, B^2) + 4\sigma^2} \sum_{\ell=1}^{t-1} \tilde{\sigma}^2 (f(x_\ell, a_\ell) - f_\star(x_\ell, a_\ell))^2 \mathbf{1}(\omega(x_\ell, a_\ell, \mathcal{G}_\ell) \leqslant \tau) \cdot \mathbf{1}(f \in \mathcal{G}_\ell) + \\
&\quad (\min(\tau B, B^2) + 4\tilde{\sigma}^2) \log(|\mathcal{F}|/\tilde{\delta}) \\
&\leqslant \frac{1}{4} \sum_{\ell=1}^{t-1} (f(x_\ell, a_\ell) - f_\star(x_\ell, a_\ell))^2 \mathbf{1}(\omega(x_\ell, a_\ell, \mathcal{G}_\ell) \leqslant \tau) \cdot \mathbf{1}(f \in \mathcal{G}_\ell) + (\min(\tau B, B^2) + 4\tilde{\sigma}^2) \log(|\mathcal{F}|/\tilde{\delta})
\end{aligned}$$

$$(14)$$

with probability at least $1 - \frac{\tilde{\delta}}{|\mathcal{F}|}$. A union bound implies the same inequality holds for all $f \in \mathcal{F}$ simultaneously with probability at least $1 - \tilde{\delta}$. Let's call this event $\mathcal{B}$. We have just shown that $\mathbb{P}(\mathcal{B}) \geqslant 1 - \tilde{\delta}$. In particular when $\mathcal{B}$ holds, inequality 14 is also satisfied for $f = f_t^\tau$. When $\mathcal{E}_t$ holds $f_t^\tau \in \mathcal{G}_{t-1}$ then $\mathbf{1}(f_t^\tau \in \mathcal{G}_\ell) = 1$ for all[4] $\ell \leqslant t - 1$ and therefore,

$$\sum_{\ell=1}^{t-1} Z_\ell^{f_t^\tau} = \sum_{\ell=1}^{t-1} (f_t^\tau(x_\ell, a_\ell) - f_\star(x_\ell, a_\ell)) \cdot \xi_\ell \cdot \mathbf{1}(\omega(x_\ell, a_\ell, \mathcal{G}_\ell) \leqslant \tau). \quad (15)$$

---

[4]This is where the definition of $G_{t-1}$ as an intersection of all previous confidence sets is important. The intersection ensures that for any $\tau$ the minimizer of the filtered least squares is achieved at an $f_t^\tau$ for which the inidicator $\mathbf{1}(f \in \mathcal{G}_\ell) = 1$ is true for all $\ell \leqslant t - 1$.

We proceed by substituting the definition of $r_\ell = f_\star(x_\ell, a_\ell) + \xi_\ell$ in equation 1 and noting that when $\mathcal{E}_t$ holds $f_\star \in \mathcal{G}_{t-1}$, so that $f_t^\tau$, the minimizer of the uncertainty filtered least squares loss satisfies,

$$\sum_{\ell=1}^{t-1} \left(f_t^\tau(x_\ell, a_\ell) - f_\star(x_\ell, a_\ell) - \xi_\ell\right)^2 \mathbf{1}(\omega(x_\ell, a_\ell, \mathcal{G}_\ell) \leqslant \tau) \leqslant \sum_{\ell=1}^{t-1} \xi_\ell^2 \mathbf{1}(\omega(x_\ell, a_\ell, \mathcal{G}_\ell) \leqslant \tau)$$

expanding the left hand side of the inequality above and rearranging terms yields,

$$\sum_{\ell=1}^{t-1} \left(f_t^\tau(x_\ell, a_\ell) - f_\star(x_\ell, a_\ell)\right)^2 \mathbf{1}(\omega(x_\ell, a_\ell, \mathcal{G}_\ell) \leqslant \tau) \leqslant 2\sum_{\ell=1}^{t-1} \left(f_t^\tau(x_\ell, a_\ell) - f_\star(x_\ell, a_\ell)\right) \cdot \xi_\ell \cdot \mathbf{1}(\omega(x_\ell, a_\ell, \mathcal{G}_\ell) \leqslant \tau)$$

$$(16)$$

To bound the right hand side of the inequality above we plug inequality 14 and equality 15 into equation 16 we conclude that when $\mathcal{B} \cap \mathcal{E}_t$ holds,

$$\sum_{\ell=1}^{t-1} \left(f_t^\tau(x_\ell, a_\ell) - f_\star(x_\ell, a_\ell)\right)^2 \mathbf{1}(\omega(x_\ell, a_\ell, \mathcal{G}_\ell) \leqslant \tau) \leqslant 2\sum_{\ell=1}^{t-1} \left(f_t^\tau(x_\ell, a_\ell) - f_\star(x_\ell, a_\ell)\right) \cdot \xi_\ell \cdot \mathbf{1}(\omega(x_\ell, a_\ell, \mathcal{G}_\ell) \leqslant \tau) \leqslant$$

$$\frac{1}{2}\sum_{\ell=1}^{t-1} \left(f_t^\tau(x_\ell, a_\ell) - f_\star(x_\ell, a_\ell)\right)^2 \mathbf{1}(\omega(x_\ell, a_\ell, \mathcal{G}_\ell) \leqslant \tau) + (2\min(\tau B, B^2) + 8\widetilde{\sigma}^2)\log(|\mathcal{F}|/\widetilde{\delta})$$

rearranging terms we conclude that when $\mathcal{B} \cap \mathcal{E}_t$ is satisfied,

$$\sum_{\ell=1}^{t-1} \left(f_t^\tau(x_\ell, a_\ell) - f_\star(x_\ell, a_\ell)\right)^2 \mathbf{1}(\omega(x_\ell, a_\ell, \mathcal{G}_\ell) \leqslant \tau) \leqslant (4\min(\tau B, B^2) + 16\widetilde{\sigma}^2)\log(|\mathcal{F}|/\widetilde{\delta})$$

with probability at least $\mathbb{P}(\mathcal{B} \cap \mathcal{E}_t) \geqslant \mathbb{P}(\mathcal{E}_t) - \widetilde{\delta}$. This finalizes the result.

$\square$

**Proposition D.1** (Intersection Result). *Let $A, B, C$ be three sets such that,*

$$\mathbb{P}(A \cap B) \geqslant \mathbb{P}(A) - \delta_1, \quad \mathbb{P}(A \cap C) \geqslant \mathbb{P}(A) - \delta_2$$

*then $\mathbb{P}(A \cap B \cap C) \geqslant \mathbb{P}(A) - \delta_1 - \delta_2$.*

*Proof.* Notice that $\mathbb{P}(A \cap B) \geqslant \mathbb{P}(A) - \delta_1$ is equivalent to $\mathbb{P}(A \backslash B) \leqslant \delta_1$. Similarly $\mathbb{P}(A \cap C) \geqslant \mathbb{P}(A) - \delta_2$ is equivalent to $\mathbb{P}(A \backslash C) \leqslant \delta_2$. Therefore,

$$\mathbb{P}(A \backslash [B \cap C]) \leqslant \mathbb{P}(A \backslash B) + \mathbb{P}(A \backslash C) \leqslant \delta_1 + \delta_2.$$

Finally, this is equivalent to the statement $\mathbb{P}(A \cap B \cap C) \geqslant \mathbb{P}(A) - \delta_1 - \delta_2$.

$\square$

**Lemma 3.2.** *The confidence intervals are valid so that $f_\star \in \mathcal{G}_t$ for all $t \in \mathbb{N}$ and optimism holds, $\max_{a \in \mathcal{A}} f_\star(x_t, a) \leqslant U_t(x_t, a_t)$ with probability at least $1 - \delta$ for all $t \in \mathbb{N}$.*

*Proof.* Applying the results of Proposition 3.1 setting $\widetilde{\delta} = \frac{\delta}{4(i+1)^2 t^2}, \tau = \tau_i$ for $i \in \{1, \cdots, \lceil\log(t)\rceil\}$ and $\widetilde{\sigma} = \sigma$ we conclude that when $\mathcal{E}_t$ holds,

$$f_\star \in \left\{ f \in \mathcal{F} : \sum_{\ell=1}^{t-1} \left(f_t^\tau(x_\ell, a_\ell) - f(x_\ell, a_\ell)\right)^2 \mathbf{1}(\omega(x_\ell, a_\ell, \mathcal{G}_\ell) \leqslant \tau_i) \leqslant \beta_t\left(\tau_i, \frac{\delta}{4(i+1)^2 t^2}, \sigma^2\right) \right\}$$

with probability at least $\mathbb{P}(\mathcal{E}_t) - \frac{\delta}{2(i+1)^2 t^2}$. Recall that

$$\mathcal{G}_t(\tau_i) = \left\{ f \in \mathcal{F} : \sum_{\ell=1}^{t-1} \left(f_t^\tau(x_\ell, a_\ell) - f(x_\ell, a_\ell)\right)^2 \mathbf{1}(\omega(x_\ell, a_\ell, \mathcal{G}_\ell) \leqslant \tau_i) \leqslant \beta_t\left(\tau_i, \frac{\delta}{4(i+1)^2 t^2}, \sigma^2\right) \right\} \cap \mathcal{G}_{t-1}(\tau_i)$$

Thus, we conclude that $f_\star \in \mathcal{G}_t(\tau_i)$ with probability at least $\mathbb{P}(\mathcal{E}_t) - \frac{\delta}{4(i+1)^2 t^2}$. Proposition D.1 and a union bound over all $i \in \{0\} \cup [q_t]$ we conclude that $f_\star \in \mathcal{G}_t$ for all $t \in \mathbb{N}$ with probability at

least $\mathbb{P}(\mathcal{E}_t) - \frac{\delta}{2t^2}$. And therefore $\mathbb{P}(\mathcal{E}_{t+1}) \geqslant \mathbb{P}(\mathcal{E}_t) - \frac{\delta}{2t2}$. Finally, since $\mathbb{P}(\mathcal{E}_1) = 1$ we conclude that $\mathbb{P}(\cap_{t=1}^{\infty}\mathcal{E}_t) \geqslant 1 - \delta$.

Optimism is an immediate consequence of the previous result. When $f_\star \in \mathcal{G}_t$, it follows that $f_\star(x_t, a) \leqslant \max_{f \in \mathcal{G}_t, a' \in \mathcal{A}} f(x_t, a') = U_t(x_t, a_t)$ for any $a \in \mathcal{A}$. And therefore $\max_{a \in \mathcal{A}} f_\star(x_t, a) \leqslant U_t(x_t, a_t)$.

$\square$

**Theorem 3.5.** *If $\delta \in (0, 1)$ is the input to Algorithm 1 satisfies,*

$$\text{Regret}(T) \leqslant \mathcal{O}\left(\sigma\sqrt{d_{\text{eluder}}(\mathcal{F}, B/T)\log(T|\mathcal{F}|/\delta)T} + B \cdot d_{\text{eluder}}(\mathcal{F}, B/T)\log(T)\log(T|\mathcal{F}|/\delta)\right).$$

*for all $T \in \mathbb{N}$ with probability at least $1 - \delta$.*

*Proof.* The analysis of the regret of Algorithm 1 follows the typical analysis for optimistic algorithms. When $\mathcal{E}$ holds,

$$\text{Regret}(T) \overset{(i)}{\leqslant} \sum_{t=1}^{T} \omega(x_t, a_t, \mathcal{G}_t)$$

$$\overset{(ii)}{\leqslant} \mathcal{O}\left(\sigma\sqrt{d_{\text{eluder}}(\mathcal{F}, B/T)\log(T|\mathcal{F}|/\delta)T} + B \cdot d_{\text{eluder}}(\mathcal{F}, B/T)\log(T)\log(T|\mathcal{F}|/\delta)\right)$$

Where inequality $(i)$ is a consequence of conditioning on $\mathcal{E}$ and Lemma 3.2 where optimism holds and follows the same logic as in the proof of Theorem B.4. The last inequality $(ii)$ follows from Lemma 3.4. We finish the proof by noting that $\mathbb{P}(\mathcal{E}) \geqslant 1 - \delta$. $\square$

# E PROOFS OF SECTION 4

In this section we list the proofs of Section 4. These are split in two subsections. In Section E.1 we present the proofs of Section 4.1. In Section E.3 we present the proofs of Section 4.3.

## E.1 PROOFS OF SECTION 4.1

**Lemma 4.1.** *Let $\tilde{\delta} \in (0, 1)$, $t \in \mathbb{N}$, $\{x_\ell, a_\ell\}_{\ell=1}^{t-1}$ be a sequence of context-action pairs and and $\{r_t\}_{t=1}^{t-1}$ be a sequence of values satisfying $r_\ell = f_\star(x_\ell, a_\ell) + \xi_\ell$ where $f_\star \in \mathcal{F}$ and the $\xi_\ell$ are conditionally zero mean. Let $\{b_\ell\}_{\ell=1}^{t-1}$ be a filtering process of Bernoulli random variables $b_\ell \in \{0, 1\}$ such that $b_\ell$ is $\Sigma(x_1, a_1, b_1, r_1, \cdots, x_{\ell-1}, a_{\ell-1}, b_{\ell-1}, r_{\ell-1}, x_\ell, a_\ell)$-measurable. Let $f_t^{\mathbf{b}_t} = \text{argmin}_{f \in \mathcal{F}} \sum_{\ell=1}^{t-1} b_\ell \cdot (f(x_\ell, a_\ell) - r_\ell)^2$ be the "filtered" least squares estimator. If Assumption 2.2 holds then,*

$$\left|\sum_{\ell=1}^{t-1} \xi_\ell \cdot b_\ell \cdot \left(f_\star(x_\ell, a_\ell) - f_t^{\mathbf{b}_t}(x_\ell, a_\ell)\right)\right| \leqslant 6B^2 \log(2|\mathcal{F}|/\tilde{\delta}).$$

*and*

$$\sum_{\ell=1}^{t-1} b_\ell \cdot \left(f_t^{\mathbf{b}_t}(x_\ell, a_\ell) - f_\star(x_\ell, a_\ell)\right)^2 \leqslant 8B^2 \log(2|\mathcal{F}|/\tilde{\delta})$$

*with probability at least $1 - \tilde{\delta}$.*

*Proof.* Substituting $y_\ell = f_\star(x_\ell, a_\ell) + \xi_\ell$ into the definition of $f_t$ we obtain the following inequalities,

$$\sum_{\ell=1}^{t-1} b_\ell \cdot (f_t^{\mathbf{b}_t}(x_\ell, a_\ell) - y_\ell)^2 \leqslant \sum_{\ell=1}^{t-1} b_\ell \cdot (f_\star(x_\ell, a_\ell) - y_\ell)^2 = \sum_{\ell=1}^{t-1} b_\ell \cdot \xi_\ell^2$$

substituting again the definition of $y_\ell$ on the left hand side of the inequality above and rearranging terms yields,

$$\sum_{\ell=1}^{t-1} b_\ell \cdot (f_t^{\mathbf{b}_t}(x_\ell, a_\ell) - f_\star(x_\ell, a_\ell))^2 \leqslant 2\sum_{\ell=1}^{t-1} \xi_\ell \cdot b_\ell \cdot \left(f_\star(x_\ell, a_\ell) - f_t^{\mathbf{b}_t}(x_\ell, a_\ell)\right) \quad (17)$$

We now focus on bounding the RHS of equation 17. For any $f \in \mathcal{F}$ let $Z_\ell^f = \xi_\ell \cdot b_\ell \cdot (f_\star(x_\ell, a_\ell) - f(x_\ell, a_\ell))$. The sequence $Z_t$ forms a martingale difference sequence such that $\mathbb{E}_\ell\left[\left(Z_\ell^f\right)^2\right] = \sigma_\ell^2 \cdot b_\ell \cdot (f(x_\ell, a_\ell) - f_\star(x_\ell, a_\ell))^2$ and Assumption 2.2 implies $|Z_\ell^f| \leqslant B^2$ for all $\ell \in \mathbb{N}$.

We can use a two sided version of Freedman inequality (see for example Lemma A.1 in Appendix A) to bound this term and show that with probability at least $1 - \delta'$,

$$
\left|\sum_{\ell=1}^{t-1} \xi_\ell \cdot b_\ell \cdot (f(x_\ell, a_\ell) - f_\star(x_\ell, a_\ell))\right| \leqslant \eta \cdot \left(\sum_{\ell=1}^{t-1} \sigma_\ell^2 \cdot b_\ell \cdot (f(x_\ell, a_\ell) - f_\star(x_\ell, a_\ell))^2\right) + \frac{\log(2/\delta')}{\eta}
$$

$$
\overset{(i)}{\leqslant} \frac{1}{4}\sum_{\ell=1}^{t-1} b_\ell \cdot (f(x_\ell, a_\ell) - f_\star(x_\ell, a_\ell))^2 + 4B^2 \log(2/\delta').
$$

Where inequality $(i)$ follows from setting $\eta = \frac{1}{4B^2}$ and noting that $\sigma_\ell \leqslant B^2$ for all $\ell$. Finally, setting $\delta' = \frac{\tilde{\delta}}{|\mathcal{F}|}$ and considering a union bound over all $f \in \mathcal{F}$ we conclude that,

$$
\left|\sum_{\ell=1}^{t-1} \xi_\ell \cdot b_\ell \cdot \left(f_\star(x_\ell, a_\ell) - f_t^{\mathbf{b}_t}(x_\ell, a_\ell)\right)\right| \leqslant \frac{1}{4}\sum_{\ell=1}^{t-1} b_\ell \cdot \left(f_t^{\mathbf{b}_t}(x_\ell, a_\ell) - f_\star(x_\ell, a_\ell)\right)^2 + 4B^2 \log(2|\mathcal{F}|/\tilde{\delta}).
$$

$$(18)$$

Plugging this inequality into equation 13 and rearranging terms yields,

$$
\sum_{\ell=1}^{t-1} b_\ell \cdot \left(f_t^{\mathbf{b}_t}(x_\ell, a_\ell) - f_\star(x_\ell, a_\ell)\right)^2 \leqslant 8B^2 \log(2|\mathcal{F}|/\tilde{\delta}). \tag{19}
$$

Moreover, combining equations 18 and 19,

$$
\left|\sum_{\ell=1}^{t-1} \xi_\ell \cdot b_\ell \cdot \left(f_\star(x_\ell, a_\ell) - f_t^{\mathbf{b}_t}(x_\ell, a_\ell)\right)\right| \leqslant 6B^2 \log(2|\mathcal{F}|/\tilde{\delta}).
$$

$\square$

**Lemma 4.2.** *Let $\tilde{\delta} \in (0,1)$ be a probability parameter. If Assumption 2.2 holds,*

$$
\frac{2}{3} \cdot W_t^{\mathbf{b}_t} - 11B^2 \log(4|\mathcal{F}|/\tilde{\delta}) \leqslant \overline{W}_t^{\mathbf{b}_t} \leqslant 2W_t^{\mathbf{b}_t} + 48B^2 \log(4|\mathcal{F}|/\tilde{\delta})
$$

*with probability at least $1 - \tilde{\delta}$ where $\overline{W}_t^{\mathbf{b}_t} = \sum_{\ell=1}^{t-1} b_\ell \cdot \sigma_\ell^2$.*

*Proof.* This result follows the same template of the proof of Lemma B.1 which we reproduce here. Substituting $r_\ell = f_\star(x_\ell, a_\ell) + \xi_\ell$ in the definition of $W_t^{\mathbf{b}_t}$,

$$
W_t^{\mathbf{b}_t} = \sum_{\ell=1}^{t-1} b_\ell \cdot (f_\star(x_\ell, a_\ell) + \xi_t - f_t^{\mathbf{b}_t}(x_\ell, a_\ell))^2
$$

$$
= \sum_{\ell=1}^{t-1} b_\ell \cdot (f_\star(x_\ell, a_\ell) - f_t^{\mathbf{b}_t}(x_\ell, a_\ell))^2 + 2\sum_{\ell=1}^{t-1} b_\ell \cdot \xi_\ell \cdot (f_\star(x_\ell, a_\ell) - f_t^{\mathbf{b}_t}(x_\ell, a_\ell)) + \sum_{\ell=1}^{t-1} b_\ell \cdot \xi_\ell^2.
$$

Applying Lemma 4.1 we conclude that,

$$
W_t^{\mathbf{b}_t} - 12B^2 \log(4|\mathcal{F}|/\tilde{\delta}) \leqslant \sum_{\ell=1}^{t-1} b_\ell \cdot \xi_\ell^2 \leqslant W_t^{\mathbf{b}_t} + 20B^2 \log(4|\mathcal{F}|/\tilde{\delta}). \tag{20}
$$

with probability at least $1 - \tilde{\delta}/2$. Using Proposition A.5 setting $\beta = 1/2$ and $\tilde{B} = B^2$ and $Z_\ell = b_\ell \cdot \xi_\ell^2$,

$$
\frac{1}{2}\sum_{\ell=1}^{t-1} b_\ell \cdot \sigma_\ell^2 - 4B^2 \log(2/\tilde{\delta}) \leqslant \sum_{\ell=1}^{t-1} b_\ell \cdot \xi_\ell^2 \leqslant \frac{3}{2}\sum_{\ell=1}^{t-1} b_\ell \cdot \sigma_\ell^2 + 4B^2 \log(2/\tilde{\delta}) \tag{21}
$$

with probability at least $1 - \tilde{\delta}/2$. Combining equations 20 and 21 with a union bound we conclude,

$$\frac{2}{3} \cdot W_t^{\mathbf{b}_t} - 8B^2 \log(4|\mathcal{F}|/\tilde{\delta}) - \frac{8}{3} \cdot B^2 \log(2/\tilde{\delta}) \leqslant \sum_{\ell=1}^{t-1} b_\ell \cdot \sigma_\ell^2 \leqslant 2W_t^{\mathbf{b}_t} + 40B^2 \log(4|\mathcal{F}|/\tilde{\delta}) + 8B^2 \log(2/\tilde{\delta})$$

with probability at least $1 - \tilde{\delta}$. Thus,

$$\frac{2}{3} \cdot W_t^{\mathbf{b}_t} - 11B^2 \log(4|\mathcal{F}|/\tilde{\delta}) \leqslant \sum_{\ell=1}^{t-1} b_\ell \cdot \sigma_\ell^2 \leqslant 2W_t^{\mathbf{b}_t} + 48B^2 \log(4|\mathcal{F}|/\tilde{\delta})$$

with probability at least $1 - \tilde{\delta}$.

$\square$

**Corollary 4.3.** *Let $\delta' \in (0, 1)$, $\{x_\ell, a_\ell, r_\ell\}_{\ell=1}^{\infty}$ be a sequence of context-action and rewards triplets such that $r_\ell = f_\star(x_\ell, a_\ell) + \xi_\ell$ where $f_\star \in \mathcal{F}$ and the $\xi_\ell$ are conditionally zero mean. Let $\{b_\ell\}_{\ell=1}^{t-1}$ be a filtering process of Bernoulli random variables $b_\ell \in \{0, 1\}$ such that $b_\ell$ is $\Sigma(x_1, a_1, b_1, r_1, \cdots, x_{\ell-1}, a_{\ell-1}, b_{\ell-1}, r_{\ell-1}, x_\ell, a_\ell)$-measurable and $f_t^{\mathbf{b}_t} = \arg\min_{f \in \mathcal{F}} \sum_{\ell=1}^{t-1} b_\ell \cdot (f(x_\ell, a_\ell) - r_\ell)^2$ be the "filtered" least squares estimator. If Assumption 2.2 holds there exists a universal constant $C > 0$ such that the cumulative variance estimator $W_t^{\mathbf{b}_t} = \sum_{\ell=1}^{t-1} b_\ell \cdot (r_\ell - f_t^{\mathbf{b}_t}(x_\ell, a_\ell))^2$ satisfies,*

$$\frac{2}{3} \cdot W_t^{\mathbf{b}_t} - C \cdot B^2 \log(t|\mathcal{F}|/\delta') \leqslant \overline{W}_t^{\mathbf{b}_t} \leqslant 2W_t^{\mathbf{b}_t} + C \cdot B^2 \log(t|\mathcal{F}|/\delta')$$

*with probability at least $1 - \delta'$ for all $t \in \mathbb{N}$.*

*Proof.* Applying Lemma 4.1 with $\tilde{\delta} = \frac{\delta'}{2t^2}$ and applying a union bound over all $t \in \mathbb{N}$ yields the inequality,

$$\frac{2}{3} \cdot W_t^{\mathbf{b}_t} - 11B^2 \log(8t^2|\mathcal{F}|/\delta') \leqslant \overline{W}_t^{\mathbf{b}_t} \leqslant 2W_t^{\mathbf{b}_t} + 48B^2 \log(8t^2|\mathcal{F}|/\tilde{\delta})$$

finally, $48 \log(8t^2|\mathcal{F}|/\delta') = \Theta\left(t|\mathcal{F}|/\delta'\right)$ yields the desired result. $\square$

### E.2 PROOFS OF SECTION 4.2

**Theorem 4.6.** *Let $\delta \in (0, 1)$. If $\delta/2$ is the input to Algorithm 1 and $\hat{\sigma}_t^2$ estimators are computed by setting $\delta' = \delta/2$. If $\sigma_t = \sigma$ for all $t \in \mathbb{N}$ the regret of Algorithm 1 with modified confidence set sizes satisfies,*

$$\text{Regret}(T) \leqslant \mathcal{O}\left(\sigma\sqrt{d_{\text{eluder}}(\mathcal{F}, B/T) \log(T|\mathcal{F}|/\delta)T} + Bd_{\text{eluder}}(\mathcal{F}, B/T) \log^2(T) \log(T|\mathcal{F}|/\delta)\right).$$

*for all $T \in \mathbb{N}$ with probability at least $1 - \delta$.*

*Proof.* The analysis of the regret of Algorithm 1 follows the typical analysis for optimistic algorithms. When $\bar{\mathcal{E}}$ holds optimism implies,

$$\text{Regret}(T) \leqslant \sum_{t=1}^{T} \omega(x_t, a_t, \mathcal{G}_t)$$

In order to bound the right hand side of the inequality above, we split it in $\lceil \log(T) \rceil$ epochs $\mathcal{T}_j$ such that $\mathcal{T}_j = [2^{j-1} + 1, \cdots, \min(2^j, T)]$.

$$\sum_{t=1}^{T} \omega(x_t, a_t, \mathcal{G}_t) \leqslant \sum_{j=1}^{\lceil \log(T) \rceil} \sum_{t \in \mathcal{T}_j} \omega(x_t, a_t, \mathcal{G}_t)$$

We proceed to bound the sums $\sum_{t \in \mathcal{T}_j} \omega(x_t, a_t, \mathcal{G}_t)$ for all epochs $\mathcal{T}_i$. When $\bar{\mathcal{E}}$ holds, and $t \in \mathcal{T}_j$ for some $j \in \lceil \log(T) \rceil$, it follows that if $f, f' \in G_t$ then for each threshold level $i \in \{0\} \cup [q_t]$,

$$\sum_{\ell \in \mathcal{T}_j \cap [t-1]} \left(f(x_\ell, a_\ell) - f'(x_\ell, a_\ell)\right)^2 \mathbf{1}(\omega(x_\ell, a_\ell, \mathcal{G}_\ell) \leqslant \tau_i) \leqslant \sum_{\ell=1}^{t-1} \left(f(x_\ell, a_\ell) - f'(x_\ell, a_\ell)\right)^2 \mathbf{1}(\omega(x_\ell, a_\ell, \mathcal{G}_\ell) \leqslant \tau_i)$$

$$\leqslant \beta_t \left(\tau_i, \frac{\delta}{2(i+1)^2}, \widehat{\sigma}_t^2\right)$$

$$\leqslant \beta_t \left(\tau_i, \frac{\delta}{2(i+1)^2}, \widehat{\sigma}_{\min(\mathcal{T}_j)}^2\right)$$

$$= \left(4 \min(\tau_i B, B^2) + 16 \widehat{\sigma}_{\min(\mathcal{T}_j)}^2\right) \log(t|\mathcal{F}|/\tilde{\delta})$$

$$\leqslant \mathcal{O} \left(\min(\tau_i B, B^2) + \sigma^2 + \frac{B^2 \log(t|\mathcal{F}|/\delta) \cdot \log(t|\mathcal{F}|/\tilde{\delta})}{t}\right)$$

where the last inequality follows from noting that equation 5 implies $\widehat{\sigma}_{\min(\mathcal{T}_j)}^2 = \mathcal{O}\left(\sigma^2 + \frac{B^2 \log(t|\mathcal{F}|/\delta)}{t}\right)$.

Thus the same argument as in the proof of Lemma 3.3 implies that when $\bar{\mathcal{E}}$ holds, for $\tau \geqslant \tau_{\max(\mathcal{T}_j)}$,

$$\sum_{t \in \mathcal{T}_j} \mathbf{1}(\omega(x_t, a_t, \mathcal{G}_t) > \tau) \leqslant \mathcal{O} \left(d_{\text{eluder}}(\mathcal{F}, \tau) \left(\frac{B \log(T|\mathcal{F}|/\delta)}{\tau} + \frac{\sigma^2 \log(T|\mathcal{F}|/\delta)}{\tau^2} + \frac{B^2 \log^2(T|\mathcal{F}|/\delta)}{\tau^2 \max(\mathcal{T}_j)} + 1\right)\right).$$

in particular, for each $t \in \mathcal{T}_j$,

$$\sum_{\ell = \min(\mathcal{T}_j)}^{t} \mathbf{1}(\omega(x_\ell, a_\ell, \mathcal{G}_\ell) > \tau) \leqslant \mathcal{O} \left(d_{\text{eluder}}(\mathcal{F}, \tau) \left(\frac{B \log(T|\mathcal{F}|/\delta)}{\tau} + \frac{\sigma^2 \log(T|\mathcal{F}|/\delta)}{\tau^2} + \frac{B^2 \log^2(T|\mathcal{F}|/\delta)}{\tau^2(t - \min(\mathcal{T}_j) + 1)}\right)\right)$$

$$\tag{22}$$

For the remainder of the argument we will mimic the proof of Lemma 3.4. We'll use the notation $d = d_{\text{eluder}}(\mathcal{F}, \frac{B}{T})$ and $\omega_t = \omega(x_t, a_t, \mathcal{G}_t)$. We will first order the sequence $\{w_t\}_{t=1}^{T}$ in descending order, as $w_{i_1}, \cdots, w_{i_T}$. We have,

$$\sum_{t=\min(\mathcal{T}_j)}^{\max(\mathcal{T}_j)} \omega_t = \sum_{t=\min(\mathcal{T}_j)}^{\max(\mathcal{T}_j)} \omega_{i_t} = \sum_{t=\min(\mathcal{T}_j)}^{\max(\mathcal{T}_j)} w_{i_t} \mathbf{1}(w_{i_t} > \frac{B}{T}) + \sum_{t=\min(\mathcal{T}_j)}^{\max(\mathcal{T}_j)} w_{i_t} \mathbf{1}(w_{i_t} \leqslant \frac{B}{T})$$

$$\leqslant \frac{B|\mathcal{T}_j|}{T} + \sum_{t=\min(\mathcal{T}_j)}^{\max(\mathcal{T}_j)} \omega_{i_t} \mathbf{1}(w_{i_t} > \frac{B}{T})$$

Applying inequality 22 setting $\tau = \omega_{i_t} > \frac{B}{T}$ we have that,

$$t - \min(\mathcal{T}_j) + 1 \leqslant \sum_{\ell = \min(\mathcal{T}_j)}^{t} \mathbf{1}(\omega(x_\ell, a_\ell, \mathcal{G}_\ell) > \omega_{i_t})$$

$$\leqslant \mathcal{O} \left(d_{\text{eluder}}(\mathcal{F}, \tau) \left(\frac{B \log(T|\mathcal{F}|/\delta)}{\omega_{i_t}} + \frac{\sigma^2 \log(T|\mathcal{F}|/\delta)}{\omega_{i_t}^2} + \frac{B^2 \log^2(T|\mathcal{F}|/\delta)}{\omega_{i_t}^2 t - \min(\mathcal{T}_j) + 1}\right)\right)$$

Therefore,

$$t - \min(\mathcal{T}_j) + 1 \leqslant \mathcal{O} \left(d_{\text{eluder}}(\mathcal{F}, B/T) \log(T|\mathcal{F}|/\delta) \cdot \max\left(\frac{B}{\omega_{i_t}}, \frac{\sigma^2}{\omega_{i_t}^2}, \frac{B}{\sqrt{d_{\text{eluder}}(\mathcal{F}, B/T)} \omega_{i_t}}\right)\right)$$

$$\leqslant \mathcal{O} \left(d_{\text{eluder}}(\mathcal{F}, B/T) \log(T|\mathcal{F}|/\delta) \cdot \max\left(\frac{B}{\omega_{i_t}}, \frac{\sigma^2}{\omega_{i_t}^2}\right)\right).$$

Thus, for all $t \in \mathcal{T}_j$ it follows that,

$$\omega_{i_t} \leqslant \mathcal{O}\left(d_{\text{eluder}}(\mathcal{F}, B/T)\log(T|\mathcal{F}|/\delta) \cdot \frac{B}{t - \min(\mathcal{T}_j) + 1} + \sigma\sqrt{\frac{d_{\text{eluder}}(\mathcal{F}, B/T)\log(T|\mathcal{F}|/\delta)}{t - \min(\mathcal{T}_j) + 1}}\right)$$

Finally, we can use this formula to sum over all $t \in \mathcal{T}_j$ for any given $j$,

$$\sum_{t \in \mathcal{T}_j} \omega_t \leqslant \mathcal{O}\left(Bd_{\text{eluder}}(\mathcal{F}, B/T)\log(|\mathcal{T}_j|)\log(T|\mathcal{F}|/\delta) + \sigma\sqrt{d_{\text{eluder}}(\mathcal{F}, B/T)\log(T|\mathcal{F}|/\delta)|\mathcal{T}_j|}\right)$$

finally, summing over all $j \in \lceil \log(T) \rceil$ we conclude,

$$\sum_{t=1}^{T} \omega_t \leqslant \mathcal{O}\left(\sum_{j=1}^{\lceil \log(T) \rceil} Bd_{\text{eluder}}(\mathcal{F}, B/T)\log(|\mathcal{T}_j|)\log(T|\mathcal{F}|/\delta) + \sigma\sqrt{d_{\text{eluder}}(\mathcal{F}, B/T)\log(T|\mathcal{F}|/\delta)|\mathcal{T}_j|}\right)$$

$$\leqslant \mathcal{O}\left(Bd_{\text{eluder}}(\mathcal{F}, B/T)\log^2(T)\log(T|\mathcal{F}|/\delta) + \sigma\sqrt{d_{\text{eluder}}(\mathcal{F}, B/T)\log(T|\mathcal{F}|/\delta)T}\right)$$

$\square$

### E.3 Proofs of Section 4.3

**Proposition 4.7.** *Let $\tilde{\delta} \in (0,1)$ and $\tau > 0$. Let $\{\widetilde{\mathcal{E}}'_\ell\}_{\ell=1}^{\infty}$ be a sequence of events such that $\widetilde{\mathcal{E}}'_1 \supseteq \widetilde{\mathcal{E}}'_2 \cdots$ and $\widetilde{\mathcal{E}}'_t \subseteq \mathcal{E}'_t$. Let $f_t^{(\tau, 2\tau]}$ be result of solving the uncertainty-filtered least-squares objective from equation 6. Additionally let $W_t^{\mathbf{b}_t^\tau}$ be the filtered estimator of the cumulative variances defined by equation 4 when setting $b_\ell^\tau = \mathbf{1}\left(\omega(x_\ell, a_\ell, \mathcal{G}'_\ell) \in (\tau, 2\tau]\right)$ and $\overline{W}_t^{\mathbf{b}_t^\tau} := \sum_{\ell=1}^{t-1} \sigma_\ell^2 \cdot \mathbf{1}\left(\omega(x_\ell, a_\ell, \mathcal{G}'_\ell) \in (\tau, 2\tau]\right)$. There exist universal constants $C, C' > 0$ such that the events $\mathcal{W}_t(\tau)$ defined for any $t$ as*

$$\sum_{\ell=1}^{t-1}\left(f_t^{(\tau, 2\tau]}(x_\ell, a_\ell) - f_\star(x_\ell, a_\ell)\right)^2 \mathbf{1}(\omega(x_\ell, a_\ell, \mathcal{G}_\ell) \in (\tau, 2\tau])$$

$$\leqslant C'\tau\sqrt{W_t^{\mathbf{b}_t^\tau}\log\left(t|\mathcal{F}|/\tilde{\delta}\right)} + C'\tau B\log\left(t|\mathcal{F}|/\tilde{\delta}\right)$$

*and*

$$C'\tau\sqrt{W_t^{\mathbf{b}_t^\tau}\log\left(t|\mathcal{F}|/\tilde{\delta}\right)} + C'\tau B\log\left(t|\mathcal{F}|/\tilde{\delta}\right) \leqslant C''\tau\sqrt{\overline{W}_t^{\mathbf{b}_t^\tau}\log\left(t|\mathcal{F}|/\tilde{\delta}\right)} + C''\tau B\log\left(t|\mathcal{F}|/\tilde{\delta}\right)$$

*satisfy the bound $\mathbb{P}(\widetilde{\mathcal{E}}'_t \cap (\mathcal{W}_t(\tau))^c) \leqslant \frac{\tilde{\delta}}{2t^2}$.*

*Proof.* Given $f \in \mathcal{F}$ we consider a martingale difference sequence $Z_\ell^f$ for $\ell \in \mathbb{N}$ defined as,

$$Z_\ell^f = (f(x_\ell, a_\ell) - f_\star(x_\ell, a_\ell)) \cdot \mathbf{1}(\omega(x_\ell, a_\ell, \mathcal{G}'_\ell) \in (\tau, 2\tau]) \cdot \mathbf{1}(f \in \mathcal{G}'_\ell) \cdot \xi_\ell$$

First let's see that

$$|Z_\ell^f| \leqslant \min(2\tau B, B^2) \quad \forall \ell \in \mathbb{N}.$$

To see this we recognize two cases, first when $f \notin \mathcal{G}'_\ell$ in which case $Z_\ell^\tau = 0$. When $f \in \mathcal{G}'_\ell$, we also recognize two cases. When $\mathbf{1}(\omega(x_\ell, a_\ell, \mathcal{G}'_\ell) \in (\tau, 2\tau]) = 0$ the random variable $Z_\ell^f = 0$. When $f \in \mathcal{G}'_\ell$, and $\omega(x_\ell, a_\ell, \mathcal{G}'_\ell) \leqslant 2\tau$, it follows that $|f(x_\ell, a_\ell) - f_\star(x_\ell, a_\ell)| \leqslant 2\tau$. Finally since $|\xi_\ell| \leqslant B$ we conclude $|Z_\ell^f| \leqslant \min(2\tau B, B^2)$.

The conditional variance of the martingale difference sequence $\{Z_\ell^f\}_\ell$ is upper bounded as

$$\text{Var}_\ell(Z_\ell^f) = \mathbb{E}_\ell[(Z_\ell^f)^2]$$

$$= \sigma_\ell^2(f(x_\ell, a_\ell) - f_\star(x_\ell, a_\ell))^2 \cdot \mathbf{1}(\omega(x_\ell, a_\ell, \mathcal{G}'_\ell) \in (\tau, 2\tau]) \cdot \mathbf{1}(f \in \mathcal{G}'_\ell)$$

$$\overset{(i)}{\leqslant} 4\tau^2\sigma_\ell^2\mathbf{1}(\omega(x_\ell, a_\ell, \mathcal{G}'_\ell) \in (\tau, 2\tau]) \cdot \mathbf{1}(f \in \mathcal{G}'_\ell)$$

where inequality $(i)$ holds because $(f(x_\ell, a_\ell) - f_\star(x_\ell, a_\ell))^2 \cdot \mathbf{1}(\omega(x_\ell, a_\ell, \mathcal{G}'_\ell) \in (\tau, 2\tau]) \cdot \mathbf{1}(f \in \mathcal{G}'_\ell) \leq 4\tau^2$. Let $\tilde{\delta}_t = \frac{\tilde{\delta}}{4t^2}$. We now invoke Lemma A.4 applied to the martingale difference sequence $\{Z_\ell^f\}$. We conclude that,

$$\sum_{\ell=1}^{t'-1} Z_\ell^f \leq 8\tau \sqrt{\sum_{\ell=1}^{t'-1} \sigma_\ell^2 \cdot \mathbf{1}\left(\omega(x_\ell, a_\ell, \mathcal{G}'_\ell) \in [\tau, 2\tau)\right) \cdot \mathbf{1}(f \in \mathcal{G}'_\ell) \cdot \ln \frac{12|\mathcal{F}|\ln 2t'}{\tilde{\delta}_t}} + 12\tau B \ln \frac{12|\mathcal{F}|\ln 2t'}{\tilde{\delta}} \tag{23}$$

with probability at least $1 - \frac{\tilde{\delta}_t}{|\mathcal{F}|}$ for all $t' \in \mathbb{N}$. A union bound implies the same inequality holds for $t' = t$ and for all $f \in \mathcal{F}$ simultaneously with probability at least $1 - \tilde{\delta}_t$. Let's call this event $\mathcal{B}_t$ so that $\mathbb{P}(\mathcal{B}_t) \geq 1 - \tilde{\delta}$. We have just shown that $\mathbb{P}(\mathcal{B}_t) \geq 1 - \tilde{\delta}_t$. In particular when $\mathcal{B}_t$ holds, inequality 23 is also satisfied for $t' = t$ and $f = f_t^{(\tau, 2\tau]}$. When $\mathcal{E}'_t$ holds we have $f_t^{(\tau, 2\tau]} \in \mathcal{G}'_{t-1}$ then $\mathbf{1}(f_t^{(\tau, 2\tau]} \in \mathcal{G}'_\ell) = 1$ for all[5] $\ell \leq t - 1$ and therefore,

$$\sum_{\ell=1}^{t-1} Z_\ell^{f_t^{(\tau, 2\tau]}} = \sum_{\ell=1}^{t-1} \left(f_t^{(\tau, 2\tau]}(x_\ell, a_\ell) - f_\star(x_\ell, a_\ell)\right) \cdot \xi_\ell \cdot \mathbf{1}(\omega(x_\ell, a_\ell, \mathcal{G}'_\ell) \in (\tau, 2\tau]). \tag{24}$$

We proceed by substituting the definition of $r_\ell = f_\star(x_\ell, a_\ell) + \xi_\ell$ in equation 1 and noting that when $\widetilde{\mathcal{E}}'_t$ holds $f_\star \in \mathcal{G}'_{t-1}$, so that $f_t^{(\tau, 2\tau]}$, the minimizer of the uncertainty-filtered empirical least squares loss satisfies,

$$\sum_{\ell=1}^{t-1} \left(f_t^{(\tau, 2\tau]}(x_\ell, a_\ell) - f_\star(x_\ell, a_\ell) - \xi_\ell\right)^2 \mathbf{1}(\omega(x_\ell, a_\ell, \mathcal{G}'_\ell) \in (\tau, 2\tau]) \leq \sum_{\ell=1}^{t-1} \xi_\ell^2 \mathbf{1}(\omega(x_\ell, a_\ell, \mathcal{G}'_\ell) \in (\tau, 2\tau])$$

expanding the left hand side of the inequality above and rearranging terms yields,

$$\sum_{\ell=1}^{t-1} \left(f_t^{(\tau, 2\tau]}(x_\ell, a_\ell) - f_\star(x_\ell, a_\ell)\right)^2 \mathbf{1}(\omega(x_\ell, a_\ell, \mathcal{G}'_\ell) \in (\tau, 2\tau])$$

$$\leq 2\sum_{\ell=1}^{t-1} \left(f_t^{(\tau, 2\tau]}(x_\ell, a_\ell) - f_\star(x_\ell, a_\ell)\right) \cdot \xi_\ell \cdot \mathbf{1}(\omega(x_\ell, a_\ell, \mathcal{G}'_\ell) \in (\tau, 2\tau]) \tag{25}$$

To bound the right hand side of the inequality above we plug inequality 23 (setting $t' = t$) into equation 25 to conclude that when $\mathcal{B}_t \cap \widetilde{\mathcal{E}}'_t$ holds,

$$\sum_{\ell=1}^{t-1} \left(f_t^{(\tau, 2\tau]}(x_\ell, a_\ell) - f_\star(x_\ell, a_\ell)\right)^2 \mathbf{1}(\omega(x_\ell, a_\ell, \mathcal{G}'_\ell) \in (\tau, 2\tau]) \leq 2\sum_{\ell=1}^{t-1} \left(f_t^{(\tau, 2\tau]}(x_\ell, a_\ell) - f_\star(x_\ell, a_\ell)\right) \cdot \xi_\ell \cdot \mathbf{1}(\omega(x_\ell, a_\ell, \mathcal{G}'_\ell) \in (\tau, 2\tau])$$

$$\leq 16\tau \sqrt{\sum_{\ell=1}^{t-1} \sigma_\ell^2 \cdot \mathbf{1}\left(\omega(x_\ell, a_\ell, \mathcal{G}'_\ell) \in (\tau, 2\tau]\right) \cdot \ln \frac{12|\mathcal{F}|\ln 2t}{\tilde{\delta}_t}} +$$

$$24\tau B \ln \frac{12|\mathcal{F}|\ln 2t}{\tilde{\delta}_t}$$

We will call Corollary 4.3 setting $\delta' = \tilde{\delta}_t$ and the filtering indicator variables equal to $b_\ell^\tau = \mathbf{1}(\omega(x_\ell, a_\ell, \mathcal{G}'_\ell) \in (\tau, 2\tau])$. Let's call $\mathcal{C}_t$ denote the event that

$$\frac{2}{3} \cdot W_{t'}^{b_{t'}^\tau} - C \cdot B^2 \log(t'|\mathcal{F}|/\tilde{\delta}_t) \leq \overline{W}_{t'}^{b_{t'}^\tau} := \sum_{\ell=1}^{t'-1} \sigma_\ell^2 \cdot \mathbf{1}\left(\omega(x_\ell, a_\ell, \mathcal{G}'_\ell) \in (\tau, 2\tau]\right) \leq 2W_{t'}^{b_{t'}^\tau} + C \cdot B^2 \log(t'|\mathcal{F}|/\tilde{\delta}_t) \tag{26}$$

---

[5]This is where the definition of $\mathcal{G}_t$ as an intersection of all confidence sets becomes important. The intersection ensures that for any $\tau$ the minimizer of the filtered least squares is achieved at an $f_t^\tau$ for which the indicator $\mathbf{1}(f \in \mathcal{G}_\ell) = 1$ is true for all $\ell \leq t - 1$.

for all $t' \in \mathbb{N}$ (and in particular true for $t' = t$) so that $\mathbb{P}\left(\mathcal{C}_t\right) \geqslant 1 - \tilde{\delta}_t$. We conclude that for any $t \in \mathbb{N}$ when $\mathcal{B}_t \cap \mathcal{C}_t \cap \widetilde{\mathcal{E}}'_t$,

$$\sum_{\ell=1}^{t-1} \left(f_t^\tau(x_\ell, a_\ell) - f_\star(x_\ell, a_\ell)\right)^2 \mathbf{1}(\omega(x_\ell, a_\ell, \mathcal{G}'_\ell) \in (\tau, 2\tau]) \leqslant 16\tau \sqrt{\left(2W_t^{\mathbf{b}_t^\tau} + C \cdot B^2 \log(t|\mathcal{F}|/\tilde{\delta})\right) \cdot \ln \frac{12|\mathcal{F}| \ln 2t}{\tilde{\delta}_t}} +$$

$$24\tau B \ln \frac{12|\mathcal{F}| \ln 2t}{\tilde{\delta}_t}$$

$$= \mathcal{O}\left(\tau \sqrt{W_t^{\mathbf{b}_t^\tau} \log\left(t|\mathcal{F}|/\tilde{\delta}\right)} + \tau B \log\left(t|\mathcal{F}|/\tilde{\delta}_t\right)\right)$$

$$= C'\tau \sqrt{W_t^{\mathbf{b}_t^\tau} \log\left(t|\mathcal{F}|/\tilde{\delta}\right)} + C'\tau B \log\left(t|\mathcal{F}|/\tilde{\delta}_t\right)$$

for some universal constant $C' > 0$ (independent of $t$). Similarly, as a consequence of the LHS of equation 26, for all $t$ when $\mathcal{B}_t \cap \mathcal{C}_t \cap \widetilde{\mathcal{E}}'_t$ holds,

$$C'\tau \sqrt{W_t^{\mathbf{b}_t^\tau} \log\left(t|\mathcal{F}|/\tilde{\delta}\right)} + C'\tau B \log\left(t|\mathcal{F}|/\tilde{\delta}\right) \leqslant C''\tau \sqrt{\overline{W}_t^{\mathbf{b}_t^\tau} \log\left(t|\mathcal{F}|/\tilde{\delta}\right)} + C''\tau B \log\left(t|\mathcal{F}|/\tilde{\delta}\right)$$

for some universal constant $C'' > 0$ (independent of $t$). We finalize the result by noting that $\mathcal{B}_t \cap \mathcal{C}_t \cap \widetilde{\mathcal{E}}'_t \subseteq \mathcal{W}_t(\tau) \cap \widetilde{\mathcal{E}}'_t$ and that $\mathbb{P}(\mathcal{B}_t \cap \mathcal{C}_t \cap \widetilde{\mathcal{E}}'_t) \geqslant \mathbb{P}(\widetilde{\mathcal{E}}'_t) - 2\tilde{\delta}_t = \mathbb{P}(\widetilde{\mathcal{E}}'_t) - \frac{\tilde{\delta}}{2t^2}$ and therefore that $\mathcal{P}(\mathcal{W}_t(\tau) \cap \widetilde{\mathcal{E}}'_t) \geqslant \mathbb{P}(\mathcal{B}_t \cap \mathcal{C}_t \cap \widetilde{\mathcal{E}}'_t) \geqslant \mathbb{P}(\widetilde{\mathcal{E}}'_t) - 2\tilde{\delta}_t = \mathbb{P}(\widetilde{\mathcal{E}}'_t) - \frac{\tilde{\delta}}{2t^2}$. Finally, this implies $\mathbb{P}(\widetilde{\mathcal{E}}'_t \cap (\mathcal{W}_t(\tau))^c) \leqslant \frac{\tilde{\delta}}{2t^2}$. $\qquad \square$

**Lemma 4.8.** *Let $\delta \in (0, 1)$. When the confidence sets $\mathcal{G}'_t \subseteq \mathcal{F}$ are defined as in Algorithm 2, then $f_\star \in \mathcal{G}'_t$ so that $\max_{a \in \mathcal{A}} f_\star(x_t, a) \leqslant U_t(x_t, a_t)$ (optimism holds), and for all $i \in [q_t]$,*

$$\mathcal{G}'_t(\tau_i) \subseteq \left\{ f \in \mathcal{F} \text{ s.t. } \sum_{\ell=1}^{t-1} \left(f_t^{(\tau_i, 2\tau_i]}(x_\ell, a_\ell) - f(x_\ell, a_\ell)\right)^2 \mathbf{1}(\omega(x_\ell, a_\ell, \mathcal{G}_\ell) \in (\tau_i, 2\tau_i]) \leqslant \right.$$

$$\left. C''\tau_i \sqrt{\overline{W}_t^{\mathbf{b}_t^{\tau_i}} \log\left(2(i+1)^2 t|\mathcal{F}|/\delta\right)} + C''\tau_i B \log\left(2i^2 t|\mathcal{F}|/\delta\right) \right\}. \tag{7}$$

*with probability at least $1 - \delta$ for all $t \in \mathbb{N}$. Where $C'' > 0$ is the same universal constant as in Proposition 4.7.*

*Proof.* Applying Proposition 4.7 with $\widetilde{\mathcal{E}}'_t = \mathcal{E}_t$, $\tau = \tau_i$ and $\tilde{\delta} = \frac{\delta}{2i^2}$ we conclude that for any $i \in [q_t]$, the event $\mathcal{W}_t(\tau_i)$ satisfies $\mathbb{P}((\mathcal{W}_t(\tau_i))^c \cap \mathcal{E}'_t) \leqslant \frac{\delta}{4i^2t^2}$. A union bound over all $i \in [q_t]$, we conclude that events $\{\mathcal{W}_t(\tau_i)\}_{i \in [q_t]}$ satisfy,

$$\mathbb{P}\left(\left(\cap_{i \in [q_t]}\mathcal{W}_t(\tau_i)\right)^c \cap \mathcal{E}'_t\right) \leqslant \sum_{i \in [q_t]} \mathbb{P}\left((\mathcal{W}_t(\tau_i))^c \cap \mathcal{E}'_t\right) \leqslant \frac{\delta}{2t^2} \tag{27}$$

Notice that when $\mathcal{W}_t(\tau_i) \cap \mathcal{E}'_t$ holds, $f_\star \in \mathcal{G}'_t(\tau_i)$ for $i \in [q_t]$. And therefore when $\left(\cap_{i \in [q_t]}\mathcal{W}_t(\tau_i)\right) \cap \mathcal{E}'_t$ holds, $f_\star \in \mathcal{G}'_t$. Thus, $\left(\cap_{i \in [q_t]}\mathcal{W}_t(\tau_i)\right) \cap \mathcal{E}'_t \subseteq \mathcal{E}'_{t+1}$.

Define a sequence of events $\mathcal{V}_t$ as $\mathcal{V}_0 = \Omega$ (the whole sample space such that $\mathbb{P}(\mathcal{V}_0) = 1$), $\mathcal{V}_t = \cap_{\ell=1}^t \left(\cap_{i \in [q_t]}\mathcal{W}_\ell(\tau_i)\right)$. Notice that by definition $\mathcal{V}_1 \supseteq \mathcal{V}_2 \supseteq \cdots$.

Notice that $f_\star \in \mathcal{G}'_0 = \mathcal{F}$ so that $\mathbb{P}(\mathcal{E}'_1) = 1$. This combined with $\left(\cap_{i \in [q_t]}\mathcal{W}_t(\tau_i)\right) \cap \mathcal{E}'_t \subseteq \mathcal{E}'_{t+1}$ implies $\cap_{\ell=1}^t \left(\cap_{i \in [q_t]}\mathcal{W}_\ell(\tau_i)\right) \subseteq \mathcal{E}'_{t+1}$ so that $\mathcal{V}_t \subseteq \mathcal{E}_{t+1}$ for all $t$.

Applying Proposition 4.7 with $\widetilde{\mathcal{E}}'_t = \mathcal{V}_{t-1}$, $\tau = \tau_i$ and $\tilde{\delta} = \frac{\delta}{2i^2}$ we conclude that for any $i \in [q_t]$, the event $\mathcal{W}_t(\tau_i)$ satisfies $\mathbb{P}((\mathcal{W}_t(\tau_i))^c \cap \mathcal{V}_{t-1}) \leqslant \frac{\delta}{4i^2t^2}$. A union bound over all $i \in [q_t]$, we conclude that events $\{\mathcal{W}_t(\tau_i)\}_{i \in [q_t]}$ satisfy,

$$\mathbb{P}\left(\left(\cap_{i\in[q_t]}\mathcal{W}_t(\tau_i)\right)^c \cap \mathcal{V}_{t-1}\right) \leqslant \sum_{i\in[q_t]}\mathbb{P}\left((\mathcal{W}_t(\tau_i))^c \cap \mathcal{V}_{t-1}\right) \leqslant \frac{\delta}{2t^2} \tag{28}$$

Since $\mathcal{V}_t' = \left[\left(\cap_{i\in[q_t]}\mathcal{W}_t(\tau_i)\right) \cap \mathcal{V}_{t-1}\right] \cup \left[\left(\cap_{i\in[q_t]}\mathcal{W}_t(\tau_i)\right)^c \cap \mathcal{V}_{t-1}\right]$ we conclude that

$$\mathbb{P}\left(\mathcal{V}_t\right) \geqslant \mathbb{P}\left(\left(\cap_{i\in[q_t]}\mathcal{W}_t(\tau_i)\right) \cap \mathcal{V}_{t-1}\right) = \mathbb{P}(\mathcal{V}_{t-1}) - \mathbb{P}\left(\left(\cap_{i\in[q_t]}\mathcal{W}_t(\tau_i)\right)^c \cap \mathcal{V}_{t-1}\right) \geqslant \mathbb{P}(\mathcal{V}_{t-1}) - \frac{\delta}{2t^2}. \tag{29}$$

Since $\mathbb{P}(\mathcal{V}_0) = 1$, unrolling inequality 29 implies that for any $m \in \mathbb{N}$,

$$\mathbb{P}\left(\mathcal{V}_m\right) = \mathbb{P}\left(\cap_{m'\leqslant m}\mathcal{V}_{m'}\right) \geqslant 1 - \sum_{\ell=1}^{m}\frac{\delta}{2\ell^2}.$$

Thus, taking the limit we conclude that $\cap_{t=1}^{\infty}\mathcal{V}_t$ holds with probability at least $\lim_{t\to\infty}\left(1 - \sum_{\ell=1}^{t}\frac{\delta}{2\ell^2}\right) \geqslant 1 - \delta$.

Since $\cap_{t=1}^{\infty}\mathcal{E}_t' \supseteq \cap_{t=1}^{\infty}\mathcal{V}_t$, we also conclude that $\cap_{t=1}^{\infty}\mathcal{E}_t'$ holds with probability at least $1 - \delta$ (when $\cap_{t=1}^{\infty}\mathcal{V}_t$ holds).

Thus we conclude that for all $i \in [q_t]$,

$$C'\tau_i\sqrt{W_t^{\mathbf{b}_t^{\tau_i}}\log\left(2i^2t|\mathcal{F}|/\delta\right)} + C'\tau_iB\log\left(2i^2t|\mathcal{F}|/\delta\right)$$
$$\leqslant C''\tau_i\sqrt{\overline{W}_t^{\mathbf{b}_t^{\tau_i}}\log\left(2i^2t|\mathcal{F}|/\delta\right)} + C''\tau_iB\log\left(2i^2t|\mathcal{F}|/\delta\right). \tag{30}$$

and $f_\star \in \mathcal{G}_t'$ is satisfied for all $t \in \mathbb{N}$ simultaneously with probability at least $1 - \delta$. Optimism holds because when $f_\star \in \mathcal{G}_t'$, $f_\star(x_t, a) \leqslant \max_{f\in\mathcal{G}_t'} f(x_t, a) \leqslant U_t(x_t, a_t)$ for all $a \in \mathcal{A}$ and therefore $\max_{a\in\mathcal{A}} f_\star(x_t, a) \leqslant U_t(x_t, a_t)$. Moreover when inequality 30 is satisfied,

$$\mathcal{G}_t'(\tau_i) = \left\{f \in \mathcal{F} \text{ s.t. } \sum_{\ell=1}^{t-1}\left(f_t^{[\tau_i, 2\tau_i)}(x_\ell, a_\ell) - f(x_\ell, a_\ell)\right)^2 \mathbf{1}(\omega(x_\ell, a_\ell, \mathcal{G}_\ell) \in (\tau_i, 2\tau_i]) \leqslant\right.$$
$$\left. C'\tau_i\sqrt{W_t^{\mathbf{b}_t^{\tau_i}}\log\left(2i^2t|\mathcal{F}|/\delta\right)} + C'\tau_iB\log\left(2i^2t|\mathcal{F}|/\delta\right)\right\}$$
$$\subseteq \left\{f \in \mathcal{F} \text{ s.t. } \sum_{\ell=1}^{t-1}\left(f_t^{(\tau_i, 2\tau_i]}(x_\ell, a_\ell) - f(x_\ell, a_\ell)\right)^2 \mathbf{1}(\omega(x_\ell, a_\ell, \mathcal{G}_\ell) \in (\tau_i, 2\tau_i]) \leqslant\right.$$
$$\left. C''\tau_i\sqrt{\overline{W}_t^{\mathbf{b}_t^{\tau_i}}\log\left(2i^2t|\mathcal{F}|/\delta\right)} + C''\tau_iB\log\left(2i^2t|\mathcal{F}|/\delta\right)\right\}.$$

This finalizes the proof. $\square$

**Theorem 4.10.** *Let $T \in \mathbb{N}$, $\delta \in (0, 1)$ and $q = \log(T)$. The regret of Algorithm 2 satisfies,*

$$\text{Regret}(T)$$
$$\leqslant \mathcal{O}\left(d_{\text{eluder}}\left(\mathcal{F}, \frac{B}{T}\right)\sqrt{\left(\sum_{t=1}^{T}\sigma_t^2\right)\log(T)\log\left(T|\mathcal{F}|/\delta\right)} + Bd_{\text{eluder}}\left(\mathcal{F}, \frac{B}{T}\right)\log(T)\log(T|\mathcal{F}|/\delta)\right)$$

*simultaneously for all $T \in \mathbb{N}$ with probability at least $1 - \delta$.*

*Proof.* Let's condition on $\mathcal{E}'$. When this event holds, optimism ensures the pseudo-regret can be upper bounded by the sum of the widths,

$$\text{Regret}(T) \leqslant \sum_{t=1}^{T}\omega(x_t, a_t, \mathcal{G}_t').$$

This is the same argument as in the proof of Theorem 3.5.

Recall that $\tau_i = \frac{B}{2^i}$. In order to bound the RHS of this inequality, we use the fact that $\omega(x_t, a_t, \mathcal{F}) \leqslant B$ for all $t \in \mathbb{N}$,

$$
\begin{aligned}
\sum_{t=1}^{T} \omega(x_t, a_t, \mathcal{G}'_t) &\leqslant \frac{T \cdot B}{2^q} + \sum_{i=1}^{q} \left( \sum_{t=1}^{T} \mathbf{1} \left( \omega(x_t, a_t, \mathcal{G}'_t) \in \left( \frac{B}{2^i}, \frac{B}{2^{i-1}} \right] \right) \cdot \frac{B}{2^{i-1}} \right) \\
&= \frac{T \cdot B}{2^q} + \sum_{i=1}^{q} \left( \sum_{t=1}^{T} \mathbf{1} \left( \omega(x_t, a_t, \mathcal{G}'_t) \in (\tau_i, 2\tau_i] \right) \cdot 2\tau_i \right) \\
&= \frac{T \cdot B}{2^q} + 2 \sum_{i=1}^{q} \tau_i \cdot \left( \sum_{t=1}^{T} \mathbf{1} \left( \omega(x_t, a_t, \mathcal{G}'_t) \in (\tau_i, 2\tau_i] \right) \right)
\end{aligned}
$$

Lemma 4.9 implies that when $\mathcal{E}'$ holds,

$$
\begin{aligned}
\sum_{t=1}^{T} \omega(x_t, a_t, \mathcal{G}'_t) &\leqslant \frac{T \cdot B}{2^q} + 2 \sum_{i=1}^{q} \tau_i \cdot \left( \sum_{t=1}^{T} \mathbf{1} \left( \omega(x_t, a_t, \mathcal{G}'_t) \in (\tau_i, 2\tau_i] \right) \right) \\
&\leqslant \frac{T \cdot B}{2^q} + 2 \sum_{i=1}^{q} \tau_i \cdot \Big( \frac{\widetilde{C} \cdot d_{\mathrm{eluder}}(\mathcal{F}, \tau_i)}{\tau_i} \sqrt{\overline{W}_T^{\mathbf{b}_T^{\tau_i}} \log\left((i+1)T|\mathcal{F}|/\delta\right)} + \\
&\qquad \frac{\widetilde{C} \cdot B \cdot d_{\mathrm{eluder}}(\mathcal{F}, \tau_i)}{\tau_i} \log\left((i+1)T|\mathcal{F}|/\delta\right) + \widetilde{C} \cdot d_{\mathrm{eluder}}(\mathcal{F}, \tau_i) \Big) \\
&= \frac{T \cdot B}{2^q} + 2\widetilde{C} \sum_{i=1}^{q} \Big( d_{\mathrm{eluder}}(\mathcal{F}, \tau_i) \sqrt{\overline{W}_T^{\mathbf{b}_T^{\tau_i}} \log\left((i+1)T|\mathcal{F}|/\delta\right)} + \\
&\qquad B \cdot d_{\mathrm{eluder}}(\mathcal{F}, \tau_i) \log\left((i+1)T|\mathcal{F}|/\delta\right) + \tau_i d_{\mathrm{eluder}}(\mathcal{F}, \tau_i) \Big) \\
&\leqslant \frac{T \cdot B}{2^q} + 2\widetilde{C} d_{\mathrm{eluder}}\left(\mathcal{F}, \frac{B}{2^q}\right) \Big[ \sum_{i=1}^{q} \sqrt{\overline{W}_T^{\mathbf{b}_T^{\tau_i}} \log\left((q+1)T|\mathcal{F}|/\delta\right)} + \\
&\qquad B \log\left((q+1)T|\mathcal{F}|/\delta\right) + \tau_i \Big]
\end{aligned}
\tag{31}
$$

Notice that,

$$
\begin{aligned}
\sum_{i=1}^{q} \sqrt{\overline{W}_T^{\mathbf{b}_T^{\tau_i}} \log\left((q+1)T|\mathcal{F}|/\delta\right)} &= \sqrt{\log\left((q+1)T|\mathcal{F}|/\delta\right)} \left( \sum_{i=1}^{q} \sqrt{\overline{W}_T^{\mathbf{b}_T^{\tau_i}}} \right) \\
&\overset{(i)}{\leqslant} \sqrt{q \log\left((q+1)T|\mathcal{F}|/\delta\right)} \sqrt{\sum_{i=1}^{q} \overline{W}_T^{\mathbf{b}_T^{\tau_i}}} \\
&\leqslant \sqrt{\left( \sum_{t=1}^{T} \sigma_t^2 \right) q \log\left((q+1)T|\mathcal{F}|/\delta\right)}
\end{aligned}
\tag{32}
$$

where inequality $(i)$ holds because $\sum_{i=1}^{q} \sqrt{z_i} \leqslant \sqrt{q \sum_{i=1}^{q} z_i}$ for $z_1, \cdots, z_q \geqslant 0$. Plugging inequality 32 into 31 and using the fact that $q = \log(T)$ we conclude that

$$\sum_{t=1}^{T} \omega(x_t, a_t, \mathcal{G}_t') \leqslant B + 2\tilde{C} d_{\text{eluder}}\left(\mathcal{F}, \frac{B}{T}\right)\sqrt{\left(\sum_{t=1}^{T}\sigma_t^2\right)\log(T)\log\left((\log(T)+1)T|\mathcal{F}|/\delta\right)}+$$

$$2\tilde{C}B d_{\text{eluder}}\left(\mathcal{F}, \frac{B}{T}\right)\log(T)\log((\log(T)+1)T|\mathcal{F}|/\delta) + 2\tilde{C}B d_{\text{eluder}}\left(\mathcal{F}, \frac{B}{T}\right)$$

$$= \mathcal{O}\left(d_{\text{eluder}}\left(\mathcal{F}, \frac{B}{T}\right)\sqrt{\left(\sum_{t=1}^{T}\sigma_t^2\right)\log(T)\log\left(T|\mathcal{F}|/\delta\right)} + B d_{\text{eluder}}\left(\mathcal{F}, \frac{B}{T}\right)\log(T)\log(T|\mathcal{F}|/\delta)\right)$$

Using the fact that $\mathbb{P}(\mathcal{E}') \geqslant 1 - \delta$ finalizes the proof.

$\square$

## F    ELUDER LEMMAS

In this section we have compiled all Lemmas that deal with eluder dimension arguments.

**Lemma 4.9.** *If $\{\mathcal{G}_t'\}_{t=1}^{\infty}$ is the sequence of confidence sets produced by Algorithm 1, there exists a universal constant $\tilde{C} > 0$ such that when $\mathcal{E}'$ is satisfied,*

$$\sum_{t=1}^{T}\mathbf{1}(\omega(x_t, a_t, \mathcal{G}_t') \in (\tau_i, 2\tau_i]) \leqslant$$

$$\frac{\tilde{C} \cdot d_{\text{eluder}}(\mathcal{F}, \tau_i)}{\tau_i}\sqrt{\overline{W}_T^{\mathbf{b}\tau_i}\log\left(iT|\mathcal{F}|/\delta\right)} + \frac{\tilde{C} \cdot B \cdot d_{\text{eluder}}(\mathcal{F}, \tau_i)}{\tau_i}\log\left(iT|\mathcal{F}|/\delta\right) + \tilde{C} \cdot d_{\text{eluder}}(\mathcal{F}, \tau_i)$$

*for all $T \in \mathbb{N}$ and $i \in [q_T]$.*

*Proof.* For simplicity we'll use the notation $d = d_{\text{eluder}}(\mathcal{F}, \tau_i)$ .

Define $\mathcal{I}_T^{(\tau_i, 2\tau_i]} = \{\ell \leqslant T \text{ s.t. } \omega(x_\ell, a_\ell, \mathcal{G}_\ell) \in (\tau_i, 2\tau_i]\}$. And in order to refer to each of its component indices let's write $\mathcal{I}_T^{(\tau_i, 2\tau_i]} = \left\{\ell_1, \cdots, \ell_{|\mathcal{I}_T^{(\tau_i, 2\tau_i]}|}\right\}$ with $\ell_1 < \ell_2 \cdots < \ell_{|\mathcal{I}_T^{(\tau_i, 2\tau_i]}|}$.

Let $N \in \mathbb{N} \cup \{0\}$. We'll start by showing that if $|\mathcal{I}_T^{(\tau_i, 2\tau_i]}| > d(N+1)$ then there exists an index $m \in \left[|\mathcal{I}_T^{(\tau_i, 2\tau_i]}|\right]$ and a pair of functions $f_{\ell_m}^{(1)}, f_{\ell_m}^{(2)} \in \mathcal{G}_{\ell_m}$ such that $f_{\ell_m}^{(1)}(x_{\ell_m}, a_{\ell_m}) - f_{\ell_m}^{(2)}(x_{\ell_m}, a_{\ell_m}) \in (\tau_i, 2\tau_i]$ and $N+1$ non-empty disjoint subsets $\tilde{S}_1, \cdots, \tilde{S}_{N+1} \subseteq \{\ell_j\}_{j \leqslant m-1}$ such that,

$$\|f_{\ell_m}^{(1)} - f_{\ell_m}^{(2)}\|_{\tilde{S}_j}^2 := \sum_{\ell \in \tilde{S}_j}\left(f_{\ell_m}^{(1)}(x_\ell, a_\ell) - f_{\ell_m}^{(2)}(x_\ell, a_\ell)\right)^2 > \tau_i^2.$$

To prove this result, let's start building the sequence $\tilde{S}_1, \cdots, \tilde{S}_{N+1}$ by setting $\tilde{S}_j = \ell_j$ for all $j = 1, \cdots, N+1$. Let's look at $m = N+2, \cdots, |\mathcal{I}_T^{(\tau_i, 2\tau_i]}|$. Consider a pair of functions $f_{\ell_m}^{(1)}, f_{\ell_m}^{(2)} \in \mathcal{G}_{\ell_m}$ such that $f_{\ell_m}^{(1)}(x_m, a_m) - f_{\ell_m}^{(2)}(x_{\ell_m}, a_{\ell_m}) > \tau_i$. If $\|f_{\ell_m}^{(1)} - f_{\ell_m}^{(2)}\|_{\tilde{S}_j}^2 > \tau_i^2$ for all $j = 1, \cdots, N+1$ the result follows. Otherwise there exists at least one $j$ such that $\|f_{\ell_m}^{(1)} - f_{\ell_m}^{(2)}\|_{\tilde{S}_j}^2 \leqslant \tau_i^2$. Thus, we can add $\ell_m$ to $\tilde{S}_j$.

Finally, by construction, the $\tilde{S}_j$ sets satisfy the definition of eluder $\tau_i$-independence and therefore they must satisfy $|\tilde{S}_j| \leqslant d$ for all $j$.

This means the process of expanding the sets $\{\tilde{S}_j\}_{j=1}^{N+1}$ must 'fail' at most after all $\tilde{S}_j$ have $d$ elements. Thus $|\mathcal{I}_T^{(\tau_i, 2\tau_i]}| > d(N+1)$ guarantees this will occur.

As we have shown, if $|\mathcal{I}_T^{(\tau_i,2\tau_i)}| > d(N+1)$, there exists an index $m \in [|\mathcal{I}_T^{(\tau_i,2\tau_i)}|]$, disjoint subsets $\widetilde{S}_1, \cdots, \widetilde{S}_{N+1} \subseteq \{\ell_j\}_{j \leqslant m-1}$ and $f_{\ell_m}^{(1)}, f_{\ell_m}^{(2)} \in \mathcal{G}_{\ell_m}$ such that,

$$(N+1)\tau_i^2 < \sum_{j \in [N+1]} \|f_{\ell_m}^{(1)} - f_{\ell_m}^{(2)}\|_{\widetilde{S}_j}^2 \leqslant \|f_{\ell_m}^{(1)} - f_{\ell_m}^{(2)}\|_{\{\ell_j\}_{j=1}^{m-1}}^2. \tag{33}$$

On the other hand, since $f_{\ell_m}^{(1)}, f_{\ell_m}^{(2)} \in \mathcal{G}_{\ell_m}' \subseteq \mathcal{G}_{\ell_m}'(\tau_i)$, we have that if $\mathcal{E}'$ holds Lemma 4.8 implies,

$$\|f_{\ell_m}^{(j)} - f_{\ell_m}^{(\tau_i,2\tau_i)}\|_{\{\ell_j\}_{j=1}^{m-1}}^2 = \sum_{\ell=1}^{\ell_{m-1}} \left( f_{\ell_m}^{[\tau_i,2\tau_i)}(x_\ell, a_\ell) - f_{\ell_m}^{(j)}(x_\ell, a_\ell) \right)^2 \mathbf{1}(\omega(x_\ell, a_\ell, \mathcal{G}_\ell') \in [\tau_i, 2\tau_i))$$

$$\leqslant C'' \tau_i \sqrt{\overline{W}_{\ell_m}^{\mathbf{b}_{\ell_m}^{\tau_i}} \log\left(2i^2\ell_m|\mathcal{F}|/\delta\right)} + C'' \tau_i B \log\left(2i^2\ell_m|\mathcal{F}|/\delta\right)$$

$$\overset{(i)}{\leqslant} C'' \tau_i \sqrt{\overline{W}_T^{\mathbf{b}_T^{\tau_i}} \log\left(2i^2T|\mathcal{F}|/\delta\right)} + C'' \tau_i B \log\left(2i^2T|\mathcal{F}|/\delta\right)$$

for $j \in \{1,2\}$. Where inequality $(i)$ holds because $\ell_m \leqslant T$ and because $\overline{W}_t^{\mathbf{b}_t^{\tau_i}}$ is monotonic w.r.t. $t$ for all $i \in \{0\} \cup [q]$ (recall $\overline{W}_t^{\mathbf{b}_t^{\tau_i}} := \sum_{\ell=1}^{t-1} \sigma_\ell^2 \cdot \mathbf{1}(\omega(x_\ell, a_\ell, \mathcal{G}_\ell') \in [\tau_i, 2\tau_i)))$. Therefore when $\mathcal{E}'$ holds,

$$\|f_{\ell_m}^{(1)} - f_{\ell_m}^{(2)}\|_{\{\ell_j\}_{j=1}^{m-1}}^2 \leqslant 2\|f_{\ell_i}^{(1)} - f_\star\|_{\{\ell_j\}_{j=1}^{m-1}}^2 + 2\|f_\star - f_{\ell_i}^{(2)}\|_{\{\ell_j\}_{j=1}^{m-1}}^2$$

$$\leqslant 4 \cdot \left( C'' \tau_i \sqrt{\overline{W}_T^{\mathbf{b}_T^{\tau_i}} \log\left(2i^2T|\mathcal{F}|/\delta\right)} + C'' \tau_i B \log\left(2i^2T|\mathcal{F}|/\delta\right) \right) \tag{34}$$

For simplicity within the context of this proof let's use the notation

$$\widetilde{\beta}_T(\tau_i, \delta) := C'' \tau_i \sqrt{\overline{W}_T^{\mathbf{b}_T^{\tau_i}} \log\left(2i^2T|\mathcal{F}|/\delta\right)} + C'' \tau_i B \log\left(2i^2T|\mathcal{F}|/\delta\right).$$

Thus combining inequalities 33 and 34 we conclude that if $N \geqslant \max\left(\frac{4\widetilde{\beta}_T(\tau_i,\delta)}{\tau_i^2} - 1, 0\right)$ then $\tau_i^2(N+1) \geqslant 4\widetilde{\beta}_T(\tau_i, \delta)$ and therefore we would incur in a contradiction because

$$4\widetilde{\beta}_T(\tau_i, \delta) \leqslant (N+1)\tau^2 < \|f_{\ell_i}^{(1)} - f_{\ell_i}^{(2)}\|_{\{\ell_j\}_{j=1}^{m-1}}^2 \leqslant 4\widetilde{\beta}_T(\tau_i, \delta).$$

This implies

$$|\mathcal{I}_T^{(\tau_i,2\tau_i)}| \leqslant d\left(\max\left(\frac{4\widetilde{\beta}_T(\tau_i,\delta)}{\tau_i^2}, 0\right) + 1\right)$$

$$\leqslant d\left(\frac{4\widetilde{\beta}_T(\tau_i,\delta)}{\tau_i^2} + 1\right)$$

$$= \frac{4dC''}{\tau_i}\sqrt{\overline{W}_T^{\mathbf{b}_T^{\tau_i}} \log\left(2i^2T|\mathcal{F}|/\delta\right)} + \frac{4dC''B}{\tau_i}\log\left(2i^2T|\mathcal{F}|/\delta\right) + d$$

$$= \mathcal{O}\left(\frac{d_{\text{eluder}}(\mathcal{F}, \tau_i)}{\tau_i}\sqrt{\overline{W}_T^{\mathbf{b}_T^{\tau_i}} \log\left(iT|\mathcal{F}|/\delta\right)} + \frac{B \cdot d_{\text{eluder}}(\mathcal{F}, \tau_i)}{\tau_i}\log\left(iT|\mathcal{F}|/\delta\right) + d_{\text{eluder}}(\mathcal{F}, \tau_i)\right)$$

Since we have shown this result for an arbitrary $T \in \mathbb{N}$ and to prove the result we have only used that $T \in \mathbb{N}$ the result follows.

$\square$

**Lemma 3.3.** *If Algorithm 1 is run with input variance upper bound $\sigma > 0$, $\mathcal{E}$ is satisfied and $\{\mathcal{G}_t\}_{t=1}^\infty$ is the sequence of confidence sets produced by Algorithm 1 then for all $T \in \mathbb{N}$ and $\tau \geqslant \tau_{q_T}$,*

$$\sum_{t=1}^T \mathbf{1}(\omega(x_t, a_t, \mathcal{G}_t) > \tau) \leqslant 3 \cdot d_{\text{eluder}}(\mathcal{F}, \tau)\left(\frac{64B\log(T|\mathcal{F}|/\delta)}{\tau} + \frac{64\sigma^2\log(T|\mathcal{F}|/\delta)}{\tau^2} + 1\right)$$

*Proof.* For simplicity we'll use the notation $d = d_{\text{eluder}}(\mathcal{F}, \tau)$. We'll start by showing the following bound,

$$\sum_{t=1}^{T} \mathbf{1}(\omega(x_t, a_t, \mathcal{G}_t) \in (\tau, 2\tau]) \leq d \left( \frac{64CB \log(T|\mathcal{F}|/\delta)}{\tau} + \frac{64C\sigma^2 \log(T|\mathcal{F}|/\delta)}{\tau^2} + 1 \right)$$

Define $\mathcal{I}_T^{(\tau, 2\tau]} = \{\ell \leq T \text{ s.t. } \omega(x_\ell, a_\ell, \mathcal{G}_\ell) \in (\tau, 2\tau]\}$. And in order to refer to each of its component indices let's write $\mathcal{I}_T^{(\tau, 2\tau]} = \left\{ \ell_1, \cdots, \ell_{|\mathcal{I}_T^{(\tau, 2\tau]}|} \right\}$ with $\ell_1 < \ell_2 \cdots < \ell_{|\mathcal{I}_T^{(\tau, 2\tau]}|}$.

Let $N \in \mathbb{N} \cup \{0\}$. We'll start by showing that if $|\mathcal{I}_T^{(\tau, 2\tau]}| > d(N+1)$ then there exists an index $i \in \left[ |\mathcal{I}_T^{(\tau, 2\tau]}| \right]$ and a pair of functions $f_{\ell_i}^{(1)}, f_{\ell_i}^{(2)} \in \mathcal{G}_{\ell_i}$ such that $f_{\ell_i}^{(1)}(x_{\ell_i}, a_{\ell_i}) - f_{\ell_i}^{(2)}(x_{\ell_i}, a_{\ell_i}) \in (\tau, 2\tau]$ and $N+1$ non-empty disjoint subsets $\widetilde{S}_1, \cdots, \widetilde{S}_{N+1} \subseteq \{\ell_j\}_{j \leq i-1}$ such that,

$$\|f_{\ell_i}^{(1)} - f_{\ell_i}^{(2)}\|_{\widetilde{S}_j}^2 := \sum_{\ell \in \widetilde{S}_j} \left( f_{\ell_i}^{(1)}(x_\ell, a_\ell) - f_{\ell_i}^{(2)}(x_\ell, a_\ell) \right)^2 > \tau^2.$$

To prove this result, let's start building the sequence $\widetilde{S}_1, \cdots, \widetilde{S}_{N+1}$ by setting $\widetilde{S}_j = \ell_j$ for all $j = 1, \cdots, N+1$. Let's look at $m = N+2, \cdots, |\mathcal{I}_T^{(\tau, 2\tau]}|$. Consider a pair of functions $f_{\ell_m}^{(1)}, f_{\ell_m}^{(2)} \in \mathcal{G}_{\ell_m}$ such that $f_{\ell_m}^{(1)}(x_m, a_m) - f_{\ell_m}^{(2)}(x_{\ell_m}, a_{\ell_m}) > \tau$. If $\|f_{\ell_m}^{(1)} - f_{\ell_m}^{(2)}\|_{\widetilde{S}_j}^2 > \tau^2$ for all $j = 1, \cdots, N+1$ the result follows. Otherwise there exists at least one $j$ such that $\|f_{\ell_m}^{(1)} - f_{\ell_m}^{(2)}\|_{\widetilde{S}_j}^2 \leq \tau^2$. Thus, we can add $\ell_m$ to $\widetilde{S}_j$.

Finally, by construction, the $\widetilde{S}_j$ sets satisfy the definition of eluder $\tau$-independence and therefore they must satisfy $|\widetilde{S}_j| \leq d$ for all $j$.

This means the process of expanding the sets $\{\widetilde{S}_j\}_{j=1}^{N+1}$ must 'fail' at most after all $\widetilde{S}_j$ have $d$ elements. Thus when $|\mathcal{I}_T^{(\tau, 2\tau]}| > d(N+1)$ guarantees this will occur. Let $\widehat{i} = \arg\min\{j \text{ s.t. } \tau_j \geq 2\tau\}$ be the smallest index in the input thresholds for Algorithm 1 such that $\tau_{\widehat{i}} \geq 2\tau$. Notice that $4\tau \geq \tau_{\widehat{i}}$.

As we have shown, if $|\mathcal{I}_T^{(\tau, 2\tau]}| > d(N+1)$, there exists an index $i \in [|\mathcal{I}_T^{(\tau, 2\tau]}|]$ and $f_{\ell_i}^{(1)}, f_{\ell_i}^{(2)} \in \mathcal{G}_{\ell_i}$ such that,

$$(N+1)\tau^2 < \sum_{j \in [N+1]} \|f_{\ell_i}^{(1)} - f_{\ell_i}^{(2)}\|_{\widetilde{S}_j}^2 \leq \|f_{\ell_i}^{(1)} - f_{\ell_i}^{(2)}\|_{\mathcal{I}_T^{(\tau, 2\tau]} \cap \{\ell_j\}_{j=1}^{i-1}}^2. \tag{35}$$

On the other hand, since $f_{\ell_i}^{(1)}, f_{\ell_i}^{(2)} \in \mathcal{G}_{\ell_i} \subseteq \mathcal{G}_{\ell_i}(\tau_{\widehat{i}})$, we have that

$$\mathcal{I}_T^{(\tau_{\widehat{i}})} = \left\{ \ell \leq T \text{ s.t. } \omega(x_\ell, a_\ell, \mathcal{G}_\ell) \leq \tau_{\widehat{i}} \right\}$$

satisfies $\mathcal{I}_T^{(\tau, 2\tau]} \subseteq \mathcal{I}_T^{(\tau_{\widehat{i}})}$ and therefore,

$$\|f_{\ell_i}^{(1)} - f_{\ell_i}^{(2)}\|_{\mathcal{I}_T^{(\tau, 2\tau]} \cap \{\ell_j\}_{j=1}^{i-1}}^2 \leq \|f_{\ell_i}^{(1)} - f_{\ell_i}^{(2)}\|_{\mathcal{I}_T^{(\tau_{\widehat{i}})} \cap \{\ell_j\}_{j=1}^{i-1}}^2 \tag{36}$$

Proposition 3.1 implies

$$\begin{aligned}
\|f_{\ell_i}^{(1)} - f_{\ell_i}^{(2)}\|_{\mathcal{I}_T^{(\tau_{\widehat{i}})} \cap \{\ell_j\}_{j=1}^{i-1}}^2 &\leq 2\|f_{\ell_i}^{(1)} - f_t^{(\tau_{\widehat{i}})}\|_{\mathcal{I}_T^{(\tau_{\widehat{i}})} \cap \{\ell_j\}_{j=1}^{i-1}}^2 + 2\|f_t^{(\tau_{\widehat{i}})} - f_{\ell_i}^{(2)}\|_{\mathcal{I}_T^{(\tau_{\widehat{i}})} \cap \{\ell_j\}_{j=1}^{i-1}}^2 \\
&\leq 4\beta_{\ell_i}(\tau_{\widehat{i}}, \delta) \\
&\leq 4\beta_T(\tau_{\widehat{i}}, \delta) \\
&= 4 \cdot (4\min(\tau_{\widehat{i}} B, B^2) + 16\sigma^2) \log(T|\mathcal{F}|/\delta) \\
&\leq 4 \cdot (4\tau_{\widehat{i}} B + 16\sigma^2) \log(T|\mathcal{F}|/\delta) \\
&\leq 64 \cdot (\tau B + \sigma^2) \log(T|\mathcal{F}|/\delta)
\end{aligned}$$

when $\mathcal{E}$ holds.

Thus combining inequalities 35 and 36 we conclude that if $N \geqslant \max\left(\frac{4\beta_T(\tau_{\hat{i}}, \delta)}{\tau^2} - 1, 0\right)$ then $\tau^2(N+1) \geqslant 4\beta_T(\tau_{\hat{i}}, \delta)$ and therefore we would incur in a contradiction because

$$4\beta_T(\tau_{\hat{i}}, \delta) \leqslant (N+1)\tau^2 < \|f_{\ell_i}^{(1)} - f_{\ell_i}^{(2)}\|^2_{\mathcal{I}_T^{(\tau_{\hat{i}})} \cap \{\ell_j\}_{j=1}^{i-1}} \leqslant 4\beta_T(\tau_{\hat{i}}, \delta).$$

This implies

$$\begin{aligned}
|\mathcal{I}_T^{(\tau, 2\tau]}| &\leqslant d\left(\max\left(\frac{4\beta_T(\tau_{\hat{i}}, \delta)}{\tau^2}, 0\right) + 1\right) \\
&\leqslant d\left(\frac{4\beta_T(\tau_{\hat{i}}, \delta)}{\tau^2} + 1\right) \\
&\leqslant d\left(\frac{64B\log(T|\mathcal{F}|/\delta)}{\tau} + \frac{64\sigma^2\log(T|\mathcal{F}|/\delta)}{\tau^2} + 1\right)
\end{aligned} \tag{37}$$

Recall that $\tau_j = \tau_0 \cdot 2^{-j}$. Observe that $I_T^{(\tau, \tau_0]} \leqslant I_T^{(\tau, 2\tau]} + \sum_{j=1}^{\hat{i}+1} I_T^{(\tau_j, \tau_{j-1}]}$. We will apply the result in equation 37 to each of these quantities and focus on bounding $\sum_{j=1}^{\hat{i}+1} I_T^{(\tau_j, \tau_{j-1}]}$. Equation 37 along with the inequality $d_{\text{eluder}}(\mathcal{F}, \tau_j) \leqslant d = d_{\text{eluder}}(\mathcal{F}, \tau)$ for all $j \leqslant \hat{i} + 1$ implies we can focus on controlling $\sum_{j=1}^{\hat{i}+1} \frac{1}{\tau_j}$ and $\sum_{j=1}^{\hat{i}+1} \frac{1}{\tau_j^2}$. We proceed to bound these terms.

$$\sum_{j=1}^{\hat{i}+1} \frac{1}{\tau_j} = \frac{1}{\tau_0} \sum_{j=1}^{\hat{i}+1} 2^j = \frac{2}{\tau_0} \sum_{j=0}^{\hat{i}} 2^j = \frac{2}{\tau_0}(2^{\hat{i}+1} - 1) \leqslant \frac{2}{\tau_{\hat{i}+1}}$$

similarly

$$\sum_{j=1}^{\hat{i}+1} \frac{1}{\tau_j^2} = \frac{1}{\tau_0^2} \sum_{j=1}^{\hat{i}+1} 2^{2j} = \frac{4}{\tau_0^2} \sum_{j=0}^{\hat{i}} 2^{2j} = \frac{4}{\tau_0^2} \cdot \left(\frac{2^{2(\hat{i}+1)} - 1}{3}\right) \leqslant \frac{4}{3\tau_0^2} \cdot 2^{2(\hat{i}+1)} \leqslant \frac{2}{\tau_{\hat{i}+1}^2}$$

Finally, since $\tau \leqslant \tau_{\hat{i}+1}$, we have $\frac{2}{\tau_{\hat{i}+1}} \leqslant \frac{2}{\tau}$ and $\frac{2}{\tau_{\hat{i}+1}^2} \leqslant \frac{2}{\tau^2}$.

Combining these results we conclude that,

$$\left|\mathcal{I}_T^{(\tau, \tau_0]}\right| \leqslant 3 \cdot d\left(\frac{64B\log(T|\mathcal{F}|/\delta)}{\tau} + \frac{64\sigma^2\log(T|\mathcal{F}|/\delta)}{\tau^2} + 1\right)$$

the result follows

$\square$

**Lemma 3.4.** *If $\mathcal{E}$ holds, then for all $T \in \mathbb{N}$ the uncertainty widths of context-action pairs from Algorithm 1 satisfy,*

$$\sum_{t=1}^{T} \omega(x_t, a_t, \mathcal{G}_t) \leqslant \mathcal{O}\left(\sigma\sqrt{d_{\text{eluder}}(\mathcal{F}, B/T)\log(T|\mathcal{F}|/\delta)T} + Bd_{\text{eluder}}(\mathcal{F}, B/T)\log(T)\log(T|\mathcal{F}|/\delta)\right).$$

*Proof.* For simplicity we'll use the notation $d = d_{\text{eluder}}(\mathcal{F}, \frac{B}{T})$ and $\omega_t = \omega(x_t, a_t, \mathcal{G}_t)$. We will first order the sequence $\{w_t\}_{t=1}^T$ in descending order, as $w_{i_1}, \cdots, w_{i_T}$. We have,

$$\sum_{t=1}^{T} \omega_t = \sum_{t=1}^{T} \omega_{i_t} = \sum_{t=1}^{T} w_{i_t} \mathbf{1}\left(w_{i_t} > \frac{B}{T}\right) + \sum_{t=1}^{T} w_{i_t} \mathbf{1}\left(w_{i_t} \leqslant \frac{B}{T}\right) \leqslant B + \sum_{t=1}^{T} \omega_{i_t} \mathbf{1}\left(w_{i_t} > \frac{B}{T}\right)$$

Applying Lemma 3.3 by setting $\tau = \omega_{i_t} > \frac{B}{T}$ we have that,

$$
t \leqslant \sum_{\ell=1}^{T} \mathbf{1}(\omega(x_\ell, a_\ell, \mathcal{G}_\ell) > \omega_{i_t})
$$

$$
\leqslant 3 \cdot d_{\mathrm{eluder}}(\mathcal{F}, \omega_{i_t}) \left( \frac{64CB \log(T|\mathcal{F}|/\delta)}{\omega_{i_t}} + \frac{64C\sigma^2 \log(T|\mathcal{F}|/\delta)}{\omega_{i_t}^2} + 1 \right)
$$

$$
\leqslant 3 \cdot d_{\mathrm{eluder}}(\mathcal{F}, B/T) \left( \frac{64CB \log(T|\mathcal{F}|/\delta)}{\omega_{i_t}} + \frac{64C\sigma^2 \log(T|\mathcal{F}|/\delta)}{\omega_{i_t}^2} + 1 \right)
$$

$$
\overset{(i)}{\leqslant} 6 \cdot d_{\mathrm{eluder}}(\mathcal{F}, B/T) \left( \frac{64CB \log(T|\mathcal{F}|/\delta)}{\omega_{i_t}} + \frac{64C\sigma^2 \log(T|\mathcal{F}|/\delta)}{\omega_{i_t}^2} \right)
$$

where the removal of the $+1$ term in step $(i)$ follows because $\omega_{i_t} > B/T$. Therefore,

$$
t \leqslant 768C \cdot d_{\mathrm{eluder}}(\mathcal{F}, B/T) \log(T|\mathcal{F}|/\delta) \cdot \max \left( \frac{B}{\omega_{i_t}}, \frac{\sigma^2}{\omega_{i_t}^2} \right).
$$

This inequality can be used to produce a bound for $\omega_{i_t}$ when $\omega_{i_t} > B/T$,

$$
\omega_{i_t} \leqslant 768C \cdot d_{\mathrm{eluder}}(\mathcal{F}, B/T) \log(T|\mathcal{F}|/\delta) \cdot \frac{B}{t} + \sigma \sqrt{\frac{768C \cdot d_{\mathrm{eluder}}(\mathcal{F}, B/T) \log(T|\mathcal{F}|/\delta)}{t}}
$$

Since $\sum_{t=1}^{T} \frac{1}{t} \leqslant 2\log(T) + 1$ and $\frac{1}{\sqrt{t}} \leqslant 2\sqrt{T}$ we have that

$$
\sum_{t=1}^{T} \omega_{i_t} \mathbf{1}(w_{i_t} > \frac{B}{T}) \leqslant
$$

$$
1536CBd_{\mathrm{eluder}}(\mathcal{F}, B/T) \log(T) \log(T|\mathcal{F}|/\delta) + \sigma \sqrt{1536C \cdot d_{\mathrm{eluder}}(\mathcal{F}, B/T) \log(T|\mathcal{F}|/\delta)T}
$$

$\square$

