# OpenReview forum: "Second Order Bounds for Contextual Bandits with Function Approximation"
_ICLR.cc/2025/Conference — ICLR 2025 Poster_

### Official Review · Reviewer_1bL1 · 2024-10-19

**Soundness:** 3
**Presentation:** 1
**Contribution:** 3
**Rating:** 5
**Confidence:** 4

**Summary:**

This work studies the contextual bandit problem under general function approximation with finite eluder dimension. The author proposes a novel algorithm based on the uncertainty-filtered multi-scale least squares procedure and achieves a variance-aware regret guarantee for general function approximation without requiring knowledge of the conditional variance. The regret for the homogeneous case seems near-optimal when reduced to the linear function class, while the regret for inhomogeneous variance has an additional dependency on $\sqrt{d}$.

**Strengths:**

1. This work provides the first algorithm with a variance-aware regret guarantee for general function approximation without requiring knowledge of the conditional variance.

2. The second-order regret for the homogeneous case seems near-optimal when reduced to the linear function class.

3. The author proposes a novel uncertainty-filtered multi-scale least squares method, which could be of independent interest.

**Weaknesses:**

1. Some parts of the presentation are highly confusing, making it difficult to follow.

(1) In Line 264, there is no Propositional 3.2, and Appendix C discusses an existing algorithm with a Hoeffding-type confidence set.

(2) In Line 279, what is the definition of the event E?

(3) In Line 299, what is Algorithm 3?

2. Regarding the regret guarantee, there is a lack of comparison with previous results, making it hard to assess the contribution of the theoretical findings. To my knowledge, the regret for the homogeneous case seems near-optimal when reduced to the linear function class, while the regret for inhomogeneous variance has an additional dependency on $\sqrt{d}$ . However, more commentary on this point is needed. Additionally, it would be beneficial to compare the regret to that of previous algorithms for general function approximation.

3. The algorithm is highly inefficient. While calculating (threshold) confidence sets is a widely used technique in RL with general function approximation, this work further relies on calculating the intersection of the confidence sets and performing optimization over the intersected confidence set. This may be intractable when the confidence sets are not convex.

**Questions:**

1. Previous algorithms that achieved second-order regret with homogeneous variance typically relied on a weighted estimator, using either uncertainty weighting [1] or variance weighting [2]. It is interesting to explore how the proposed algorithm can achieve second-order regret without employing any weighting mechanisms, which could be of independent interest.

[1] Variance-dependent regret bounds for linear bandits and reinforcement learning: Adaptivity and computational efficiency.

[2] Nearly minimax optimal reinforcement learning for linear mixture markov decision processes

2. For the variance $\sigma_t$, should it be fixed, or can it be a random variable that depends on the selection of the action?

3. In Line 171, the claim regarding the first algorithm seems incorrect. As mentioned by the author in Line 99, second-order bounds for contextual bandits were already developed in Zhao et al. (2022).

4. For the contextual bandit problem, recently, a new work [3] achieved second-order regret for the dueling bandit case. It would be helpful if the paper provided some discussion on it.

[3] Variance-Aware Regret Bounds for Stochastic Contextual Dueling Bandits

---

> ### Author Response · Authors · 2024-11-22
> **Rebuttal for reviewer 1bL1**
>
> We are very happy to see the reviewer agrees our work is novel, tackles an interesting problem and does so convincingly. Regardless, we want to reiterate our contributions. In this work we introduce algorithms for regret minimization of variance aware contextual bandits with function approximation. We solve this problem for two different settings, first the scenario where the noise variance is uniform and equal to a value $\sigma^2$ during all rounds. In this case we show our algorithm (algorithm 1) achieves a regret rate scaling as $\sigma\sqrt{ d \log(F) T} + d\log(F)$ where $d$ is the eluder dimension and $\log(F)$ is the logarithm of the size of the reward function class. This result nails down the complexity of variance aware contextual bandits in the function approximation regime and generalizes existing results in that space that could only prove bounds for linear problems. At the time of submission ours was the first algorithm achieving this type of result. When the variances are unknown and changing at all time steps we achieve a regret bound of order $d \sqrt{ \sum_{t=1}^T \sigma_t^2 \log(F)} + d\log(F)$ (notice the $\sqrt{d}$ degradation with respect to the uniform variance result). This result also recovers and generalizes existing results that deal only with linear problems. The main difficulty in obtaining our results was to generalize the linear techniques to deal with general function approximation.
>
> We address the weaknesses section of the review:
>
> Thanks for finding the typo in Line 264. The label clearly refers to Lemma 3.2 that is right above this line. We changed the name from Proposition to Lemma this typo was introduced. We fail to see how this could have been highly confusing. We believe the reviewer has a typo in their review and they mean Appendix B. This appendix contains a fully self contained discussion on Optimistic Least Squares to make it easier for the reader to understand the related literature. We fail to understand why this is controversial, particularly since a) it is in the appendix, b) there is no claim in the main that this is our contribution. The event $E$ is defined in line 264 (before line 279). Algorithm 3 is the optimistic least squares algorithm the reviewer has pointed out above from Appendix B. It isn’t really needed in the discussion from the main section, so this reference can be removed. Thanks for pointing to this.
>  We are baffled by this “lack of comparison with previous results”. We describe several related works in this area in our introductory section. As the reviewer rightly points out, ours is the first work to deal with second order bounds in the function approximation regime and thus, there is no previous work that works on the exact same problem. That being said, we mention a plethora of related linear works from the literature that had a strong influence in our work along with their regret bounds. If the reviewer has a specific work in mind we should compare to we would be really happy to include it. In short, our fixed variance result matches the best second order bounds for linear bandits, and our changing variance results match the best results for linear bandits up to a factor $\sqrt{d}$ where $d$ equals the eluder dimension of the class. As for the comparison with second order algorithms for function approximation, there are no previous second order algorithms for general function approximation that do not assume the noise process can be fully learned, an assumption that is very strong in our opinion. Our fixed variance result matches the best second order bounds for function approximation even with this stronger assumption, and our changing variance results qualitatively match the best results under noise realizability up to a factor $\sqrt{d}$ where $d$ equals the eluder dimension of the class. Nonetheless, we argue these results are incomparable since, results with noise realizability depend on an eluder dimension that captures the complexity of the noise process, something that could be much much larger than the eluder dimension of the mean rewards function class that we depend on.
> We agree with this. It is an interesting question how to get around these intractability issues. That being said, we consider our contribution to be conceptual and theoretical and still believe that it can unlock a plethora of research questions along the way, including how to make these algorithms tractable.

---

> > ### Author Response · Authors · 2024-11-22
> > **Continuation of Rebuttal**
> >
> > Weighted estimator,
> >
> > This is a great question!  As the reviewer has noticed, one of the main issues and why generalizing linear results to the function approximation regime is not straightforward is that in linear problems it is possible to make use of weighted least squares objectives to derive variance aware results, while in the function approximation case, and due to the definition of eluder dimension, the weighted regression techniques of [Zhao et al 2023] are not viable. In the fixed variance case, we can overcome this problem by developing a sharp version of the eluder bound on the widths, (see Lemma 3.3 and its proof in the Appendix). This is one of our main contributions and what allows us to obtain a sharp bound that matches linear results even without using weighted regression. In the changing variances case, developing a sharp version of this lemma where the $1/\tau^2$ term depends on the changing variances is much more challenging, so we had instead to rely on Lemma 4.9 where instead we trade off the square dependence on $1/\tau^2$ for a $1/\tau$ that is instead multiplied with $\sqrt{\sum_{\ell}\sigma_\ell^2}$. We think there might exist a way to shave off the extra factor of $\sqrt{d}$ in our bounds. This is an exciting open problem. We really like the reviewer’s suggestion of adding comments detailing the technical challenges here.
> >
> >
> > Other questions.
> >
> > 2. In our work the variances can be adaptively selected by an adversary after the context, action pairs are selected.
> > 3. Line 171. Thanks a lot for catching this. We will change the wording of this to reflect this is the first work that achieves sharp second order bounds for function approximation under the assumption that the mean reward class is realizable.
> > 4. Thanks for pointing at this reference. We will certainly add this to the related work / introductory section.

---

> > > ### Author Response · Authors · 2024-11-29
> > > **Follow up**
> > >
> > > We want to reach out regarding our latest response. We hope it addresses the reviewer's concerns. In case it does, we would really appreciate it if the reviewer could consider increasing their score.

---

> > > > ### Author Response · Authors · 2024-12-02
> > > > **Follow up**
> > > >
> > > > Dear Reviewer,
> > > >
> > > > A kind reminder that the back and forth period between authors and reviewers is coming to an end. We hope our rebuttal addressed the reviewer's concerns. If so, we would appreciate it if the reviewer could consider improving their score.
> > > >
> > > > Thanks!
> > > >
> > > > The Authors

---

> > > > > ### Comment · Reviewer_1bL1 · 2024-12-03
> > > > >
> > > > > Thanks for the author's feedback and I will keep the score. Following are the details of the confusion:
> > > > >
> > > > > The sentence, "The proof of Proposition 3.2 can be found in Appendix C," on Line 264, is misleading. Proposition 3.2 focuses on the guarantees of a Bernstein-type algorithm, but Appendix C, titled "PROOFS OF SECTION B," instead discusses an existing algorithm with a Hoeffding-type confidence set (as referenced in Section B). This creates a significant discrepancy, as the appendix content does not appear to directly support Proposition 3.2.

---

> > > > > > ### Author Response · Authors · 2024-12-03
> > > > > > **Clarification**
> > > > > >
> > > > > > Dear Reviewer,
> > > > > >
> > > > > >
> > > > > > There is a typo in the link listed in the main.It should say Appendix D. Using the search function one can easily find the proof of Lemma 3.2 starts in Line 1119 of the Appendix. The table of contents in page 14 clearly specifies that Appendix D contains the proofs of section 3. If you go to that table of contents, you can see where all proofs are. The proofs of Appendix C  are from Section B, which contains a re-derivation of the bounds of the Optimistic Least Squares Algorithm from Russo et al, that only use Hoeffding bounds. If the reviewer had scrolled a little below, and looked at the table of contents, would have easily found the results they are looking for. We hope the reviewer is so kind as to acknowledge this message and that our rebuttal addressed the reviewer's concerns. If so, we would appreciate it if the reviewer could consider improving their score.
> > > > > >
> > > > > > Thanks!
> > > > > >
> > > > > > The authors.

---

### Official Review · Reviewer_PCBb · 2024-11-02

**Soundness:** 3
**Presentation:** 2
**Contribution:** 3
**Rating:** 8
**Confidence:** 3

**Summary:**

In this paper, the authors studied the variance-dependent regret bound for contextual bandits with function approximation. They focus on scenarios where the reward function has a bounded eluder dimension. The analysis begins by considering the case where the variance is known. The authors introduce an optimistic least squares approach with multiple layers, where each layer maintains a confidence set using only data points with correspondingly small uncertainty. Combined with a routine that estimates the cumulative variance, they demonstrate that the algorithm can be adapted to the case with unknown variance. This algorithm matches the performance of the linear bandit case with unknown variance. Finally, with a modified filtering method that selects data points where the variance falls within a certain range, they show that their algorithm can be generalized to scenarios where the unknown variance changes.

**Strengths:**

1. The paper studied a variance-dependent regret bound for contextual bandits, which I think is an interesting problem to study.

2. The paper is generally well-written, and the proof appears to be correct.

3. Their regret bound for contextual bandits with unchanged unknown variance matches the results for linear bandits. I think this is a valuable addition to the literature and may have further theoretical impacts.

**Weaknesses:**

1. Given the variance-dependent regret bound for linear bandits in the literature, it is not surprising that one can obtain a variance-dependent regret bound under general function approximation. Moreover, there is still a $\sqrt{d}$ factor gap in the general case where variance changes are allowed, compared to the linear setting. I would suggest that the authors highlight the obstacle to removing this gap in the main content.

2. The approach in this paper adapts [Zhao et al., 2023] to the general function approximation case. I would suggest that the authors highlight the technical challenges and the key difference in a specific subsection or paragraph.

3. I am not sure whether the regret bound under changed variance is a significant setting on its own; it appears somewhat artificial. By contrast, the linear bandit case holds practical implications, as demonstrated by [Zhao et al., 2023], where their algorithm is applied to linear mixture MDPs as an example. It would be great to provide motivating examples or potential applications for the changed variance setting.

Zhao, Heyang, et al. "Variance-dependent regret bounds for linear bandits and reinforcement learning: Adaptivity and computational efficiency." The Thirty Sixth Annual Conference on Learning Theory. PMLR, 2023.

**Questions:**

1. Do you think the current regret bound for changed variance is tight?

---

> ### Author Response · Authors · 2024-11-22
> **Rebuttal to reviewer PCBb**
>
> We are very happy to see the reviewer agrees our work is novel, tackles an interesting problem and does so convincingly. Regardless, we want to reiterate our contributions. In this work we introduce algorithms for regret minimization of variance aware contextual bandits with function approximation. We solve this problem for two different settings, first the scenario where the noise variance is uniform and equal to a value $\sigma^2$ during all rounds. In this case we show our algorithm (algorithm 1) achieves a regret rate scaling as $\sigma\sqrt{ d \log(F) T} + d\log(F)$ where $d$ is the eluder dimension and $\log(F)$ is the logarithm of the size of the reward function class. This result nails down the complexity of variance aware contextual bandits in the function approximation regime and generalizes existing results in that space that could only prove bounds for linear problems. At the time of submission ours was the first algorithm achieving this type of result. When the variances are unknown and changing at all time steps we achieve a regret bound of order $d \sqrt{ \sum_{t=1}^T \sigma_t^2 \log(F)} + d\log(F)$ (notice the $\sqrt{d}$ degradation with respect to the uniform variance result). This result also recovers and generalizes existing results that deal only with linear problems. The main difficulty in obtaining our results was to generalize the linear techniques to deal with general function approximation.
>
> We address the reviewer’s “weaknesses” comments now.
>
> [1. and 2.]  As the reviewer has noticed, one of the main issues and why this generalization is not straightforward is that in linear problems it is possible to make use of weighted least squares objectives to derive variance aware results, while in the function approximation case, and due to the definition of eluder dimension, the weighted regression techniques of [Zhao et al 2023] are not viable. In the fixed variance case, we can overcome this problem by developing a sharp version of the eluder bound on the widths, (see Lemma 3.3 and its proof in the Appendix). This is one of our main contributions and what allows us to obtain a sharp bound that matches linear results even without using weighted regression. In the changing variances case, developing a sharp version of this lemma where the $1/\tau^2$ term depends on the changing variances is much more challenging, so we had instead to rely on Lemma 4.9 where instead we trade off the square dependence on $1/\tau^2$ for a $1/\tau$ that is instead multiplied with $\sqrt{\sum_{\ell}\sigma_\ell^2}$. We think there might exist a way to shave off the extra factor of $\sqrt{d}$ in our bounds. This is an exciting open problem. We really like the reviewer’s suggestion of adding comments detailing the technical challenges here.
>
> [3.] Thanks a lot for the question. We would like to respectfully push back against the reviewer’s comment. The changing variance scenario is a precursor to the MDP case. In that case, the policy played is changing at all time-steps and therefore the variance of the policy played depends on it. We believe the techniques we have developed in this work can be further developed to derive variance dependent bounds for RL with function approximation.

---

> > ### Author Response · Authors · 2024-11-29
> > **Follow up**
> >
> > Dear Reviewer PCBb,
> >
> > Since the discussion time is coming to an end we wanted reach out regarding our latest response. We hope it addresses the reviewer's concerns. In case it does, we would really appreciate it if the reviewer could consider increasing their score.
> >
> > Thanks a lot!
> >
> > The Authors

---

> > > ### Comment · Reviewer_PCBb · 2024-11-30
> > >
> > > I am satisfied with the author's response, so I have decided to increase my score.

---

> > > > ### Author Response · Authors · 2024-12-01
> > > > **Thanks a lot**
> > > >
> > > > Dear Reviewer,
> > > >
> > > > Thank you so much for your feedback. We really appreciate it.
> > > >
> > > > Thanks!
> > > >
> > > > The Authors

---

### Official Review · Reviewer_Fdz3 · 2024-11-03

**Soundness:** 3
**Presentation:** 2
**Contribution:** 3
**Rating:** 6
**Confidence:** 3

**Summary:**

This paper introduces new regret bounds for contextual bandit problems with function approximation, assuming the realizability condition. The authors build on the framework of optimistic algorithms for contextual bandits, and they also propose new algorithms with improved regret bounds that adapt to the variance of the measurement noise under the eluder dimension.

**Strengths:**

- The authors present a study of contextual bandits with function approximation by developing algorithms with variance-dependent regret bounds, marking the first instance of such bounds in this setting.
- These results improve upon existing bounds by requiring only a realizability assumption on the mean reward function, avoiding stronger, often impractical assumptions on measurement noise.
- The paper proposes two algorithms with distinct performance guarantees. Algorithm 1 is designed for cases with uniform but unknown measurement noise variance across time steps. This algorithm achieves a sharp dependence on the complexity of the reward function class, measured by the eluder dimension. Algorithm 2 is tailored for scenarios with variable noise variances. Its regret bound scales with the square root of the cumulative noise variance but linearly with the eluder dimension, providing an efficient approach for non-uniform noise settings.

**Weaknesses:**

However, several aspects of the paper’s presentation and empirical support could be improved:
- The authors did not include empirical experiments or numerical results to validate the theoretical bounds. This leaves the practical effectiveness of the algorithms untested.
- The theoretical content is densely packed, with lemmas presented consecutively without sufficient exposition, making it challenging to follow the logical flow of arguments.

**Questions:**

- There is no direct comparison with prior work is provided, making it difficult to clearly assess the improvements or distinctions of these results relative to existing bounds in the literature.

---

> ### Author Response · Authors · 2024-11-18
> **Rebuttal Reviewer Fdz3**
>
> We are very happy to see the reviewer agrees our work is novel, tackles an interesting problem and does so convincingly. Regardless, we want to reiterate our contributions. In this work we introduce algorithms for regret minimization of variance aware contextual bandits with function approximation. We solve this problem for two different settings, first the scenario where the noise variance is uniform and equal to a value $\sigma^2$ during all rounds. In this case we show our algorithm (algorithm 1) achieves a regret rate scaling as $\sigma\sqrt{ d \log(F) T} + d\log(F)$ where $d$ is the eluder dimension and $\log(F)$ is the logarithm of the size of the reward function class. This result nails down the complexity of variance aware contextual bandits in the function approximation regime and generalizes existing results in that space that could only prove bounds for linear problems. At the time of submission ours was the first algorithm achieving this type of result. When the variances are unknown and changing at all time steps we achieve a regret bound of order $d \sqrt{ \sum_{t=1}^T \sigma_t^2 \log(F)} + d\log(F)$ (notice the $\sqrt{d}$ degradation with respect to the uniform variance result). This result also recovers and generalizes existing results that deal only with linear problems. The main difficulty in obtaining our results was to generalize the linear techniques to deal with general function approximation. One of the main issues there being that in linear problems it is possible to make use of weighted least squares objectives to derive variance aware results, while in the function approximation case, and due to the definition of eluder dimension, weighted regression is not a viable technique.
>
> Although we sympathize with the reviewer’s request for empirical evaluations, we want to emphasize that we consider the main contribution of our work to be conceptual / theoretical. In this case for example, the hard computational nature of algorithms for function approximation based on the principle of optimism is well known by the statistical learning community It would be a pity to reject a work because it is theory-intensive, particularly given that the reviewer seems to agree our results are novel, correct and meaningful. We are happy to rebut any valid criticism of our theoretical results based on their relationship to other relevant works in the field of statistical learning theory. If the reviewer has any such criticism we are more than happy to provide an answer. That being said, we will make sure the final version of our manuscript is friendlier to the reader than the current one. Any concrete actions the reviewer thinks we should take that do not compromise the quality and formality of our results will be highly appreciated. We also very much agree that it is an important question how to use these ideas to design second order algorithms that can be used empirically.
>
>
> We are confused by the reviewers comment of the lack of comparison with prior work. We describe several related works in this area in our introductory section (see for example lines 77 to 105). As the reviewer rightly points out, ours is the first work to deal with second order bounds in the function approximation regime and thus, there is no previous work that works on the exact same problem. That being said, we mention a plethora of related linear works from the literature that had a strong influence in our work. If the reviewer has a specific work in mind we should compare to we would be really happy to include it.
>
> To our knowledge, at the time of submission, the only work that dealt with the function approximation regime in second order bounds is [Wang 2024a,b]. We discuss in detail a comparison with that work in the introductory section. As the reviewer identified, we provide several improvements over their results, chief among them having no need for the measurement noise distribution to be learnable, something that we consider a very limiting assumption from [Wang 2024a,b].

---

> > ### Author Response · Authors · 2024-11-29
> > **Follow up**
> >
> > We want to reach out regarding our latest response. We hope it addresses the reviewer's concerns. In case it does, we would really appreciate it if the reviewer could consider increasing their score.

---

> > > ### Author Response · Authors · 2024-12-02
> > > **Follow up**
> > >
> > > Dear Reviewer,
> > >
> > > A kind reminder that the back and forth period between authors and reviewers is coming to an end. We hope our rebuttal addressed the reviewer's concerns. If so, we would appreciate it if the reviewer could consider improving their score.
> > >
> > > Thanks!
> > >
> > > The Authors

---

### Official Review · Reviewer_QJSj · 2024-11-12

**Soundness:** 2
**Presentation:** 2
**Contribution:** 2
**Rating:** 5
**Confidence:** 3

**Summary:**

The paper proposes variance-aware algorithms for contextual bandits with function approximation, with only mean reward realizability assumption for known and unknown variances of noise.

**Strengths:**

1. Solving for varying variance over time period is practical and may yield tighter regret bound in some cases.
2. The analysis of the variance and the truncated loss has novelty.

**Weaknesses:**

1. The organization of the paper is hard to follow. The theorems and colloraries are hard to follow since there is no intuitive explanation on the terms (e.g., $\mathcal{G}_t^\prime(\tau_i)$, $\tilde{\mathcal{E}_t^\prime}$ ). It is confusing whether the result mentions either $\sigma_t=\sigma$ case or not.  I suggest the authors move the $\sigma_t =\sigma$ case to appendix to clearly see the results.
2. Typos:

In line 163 $\mathcal{X} \times \mathcal{X} .. $ should be $\mathcal{X} \times \mathcal{A} .. $.

In Algorithm 1 line 5, $f_t^{\tau}$ should be $f_t^{\tau_i}

In eq. (5), $\widehat{\sigma}_t$ should be $\widehat{\sigma}_t^2$

**Questions:**

1. Could the authors give an overview for the main contribution of the paper?
2. How the estimation of $\sigma^2_t$ is possible, especially for the unknown variance?

---

> ### Author Response · Authors · 2024-11-18
> **Rebuttal to QJSj**
>
> In this work we introduce algorithms for regret minimization of variance aware contextual bandits with function approximation. We solve this problem for two different settings, first the scenario where the noise variance is uniform and equal to a value $\sigma^2$ during all rounds. In this case we show our algorithm (algorithm 1) achieves a regret rate scaling as $\sigma\sqrt{ d \log(F) T} + d\log(F)$ where $d$ is the eluder dimension and $\log(F)$ is the logarithm of the size of the reward function class. This result nails down the complexity of variance aware contextual bandits in the function approximation regime and generalizes existing results in that space that could only prove bounds for linear problems. At the time of submission ours was the first algorithm achieving this type of result. When the variances are unknown and changing at all time steps we achieve a regret bound of order $d \sqrt{ \sum_{t=1}^T \sigma_t^2 \log(F)} + d\log(F)$ (notice the $\sqrt{d}$ degradation with respect to the uniform variance result). This result also recovers and generalizes existing results that deal only with linear problems. The main difficulty in obtaining our results was to generalize the linear techniques to deal with general function approximation. One of the main issues there being that in linear problems it is possible to make use of weighted least squares objectives to derive variance aware results, while in the function approximation case, and due to the definition of eluder dimension, weighted regression is not a viable technique.
>
> One of our contributions is precisely the methods we develop for estimation of the variances up to constant accuracy in the case where the variances change at every time-step. As it is well explained in section 4.1 of the main paper, this can be done by considering . This is one of the contributions of our work.
>
> The terms mentioned by the reviewer are defined in the main paper, as a careful reading of our submission would indicate, for example $G_t’$ is defined in line 6 of algorithm 2. Tilde($E_t’$) is a self contained object from Proposition 4.7. Our manuscript also clearly states that the results of section  4.3 pertain to the setting where the variance is unknown but constant for all rounds. Similarly, the results of section 4.3 pertain to the setting where the variances are changing at each step. We disagree about moving the $\sigma_t= \sigma$ case to the appendix as this is one of our main results. Notice that this case has a regret bound with a better $d_{eluder}$ dependence than the changing variance scenario. We do appreciate the reviewer’s comments regarding organization and we will take these into account while making the final version of our manuscript.
>
> Thanks for catching these typos! We will change these in the final version.
>
> We hope the reviewer considers increasing their score if this response addressed the concerns.

---

> > ### Comment · Reviewer_QJSj · 2024-11-25
> >
> > Thank you for the response, but my concerns are not resolved.
> > When the theoretical results hold for arbitral $\sigma_t$, they should hold for $\sigma_t=\sigma$ case thus moving $\sigma_t=\sigma$ case to the Appendix does not reduces the contribution of the paper.
> >
> > I couldn't find the answer for the question 2 in the response.
> > While estimating $\sigma_t$ and including it into the confidence sets is the core part to obtain $O(\sqrt{\sum_{t=1}^{T} \sigma_t^2} ), when $\sigma_t$ is arbitral, we do not have enough observation to estimate $\sigma_t$.
> > We do not know whether the variance of the reward of the next decision period will increase or decrease.
> > How the estimation of $\sigma_t$ is possible, especially for the unknown variance?

---

> ### Author Response · Authors · 2024-11-25
> **Response to Reviewer QJSj**
>
> Dear Reviewer QJSj,
>
> Thanks a lot for your engagement. Section 4.1 in the paper describes in detail how to estimate the sum of the variances $\sum_{\ell=1}^t \sigma_\ell^2$ (up to multiplicative accuracy). See Lemma 4.2 and Corollary 4.3.
>
> The idea of this result is to:
>
> 1.	Estimate a regression model of the historical dataset (see lines 312 to 314). In this case we consider a filtered least squares estimator because we require to estimate the variances in each uncertainty bucket in Algorithm 2. We will continue the discussion using the simpler scenario when $b_\ell=1$ for all $\ell \in\mathbb{N}$. Call this estimator $f_t$.
> 2.	Use this regression model to produce an estimator of the **sum** of the variances $\sum_{\ell=1}^{t-1} \sigma_\ell^2$ by computing $\sum_{\ell=1}^{t-1} r_\ell – f_t(x_\ell, a_\ell))^2$. This is what we call $W_t$.
>
> The proof of Lemma 4.2 and Corollaries 4.3 uses the following logic. A martingale argument shows that for any fixed $f$, the conditional expectations $E[ ( r_\ell – f(x_\ell, a_\ell))^2 |  F_\ell ] = \sigma_\ell^2 + (f(x_\ell, a_\ell) – f_\star(x_\ell, a_\ell))^2 $ where in this response we use the notation $F_\ell$ as the sigma algebra of all events up until and including the selection of $x_\ell, a_\ell$. A martingale argument based on a Bernstein bound (see Proposition A.5 in Appendix A) shows that,
>
> $$ \Omega(  \sum_{\ell=1}^{t-1} \sigma_\ell^2 + (f(x_\ell, a_\ell) – f_\star(x_\ell, a_\ell))^2    - B^2 \log(t/\delta’)) \leq \sum_{\ell=1}^{t-1}   ( r_\ell – f(x_\ell, a_\ell))^2 \leq O(   \sum_{\ell=1}^{t-1}   \sigma_\ell^2 + (f(x_\ell, a_\ell) – f_\star(x_\ell, a_\ell))^2    +  B^2 \log(t/\delta’) )$$
>
> With probability at least $1-\delta’$ for all $t \in \mathbb{N}$. After a union bound over all functions in the class and setting $\delta’ = \delta/|F|$, we can plug in $f_t$ and conclude that,
>
> $$ \Omega(  \sum_{\ell=1}^{t-1} \sigma_\ell^2 + (f_t(x_\ell, a_\ell) – f_\star(x_\ell, a_\ell))^2    - B^2 \log(t|F|/\delta)) \leq \sum_{\ell=1}^{t-1}   ( r_\ell – f_t(x_\ell, a_\ell))^2 \leq O(   \sum_{\ell=1}^{t-1}   \sigma_\ell^2 + (f_t(x_\ell, a_\ell) – f_\star(x_\ell, a_\ell))^2    +  B^2 \log(t|F|/\delta) )$$
>
> Finally, we use that $\sum_{\ell=1}^{t-1} (f_t( x_\ell, a_\ell) – f_\star(x_\ell, a_\ell) ) = O( B^2 \log(t|F|/\delta)  ) $ (see Lemma 4.1 – apply an anytime version of this result) with probability $1-\delta$ for all $t \in \mathbb{N}$.  Substituting this result in the expression above and rearranging terms we conclude that,
>
>
> $$ \Omega(  \sum_{\ell=1}^{t-1} \sigma_\ell^2 - B^2 \log(t|F|/\delta’)) \leq \sum_{\ell=1}^{t-1}   ( r_\ell – f_t(x_\ell, a_\ell))^2 \leq O(   \sum_{\ell=1}^{t-1}   \sigma_\ell^2     +  B^2 \log(t|F|/\delta’) )$$
>
> This argument can be found in the proof of Lemma 4.2 and Corollary 4.3 in line 1219-1274 of Appendix. E.1
>
> This is the same as when we estimate the sums of the means of an adapted process. Using standard martingale arguments one can show it is possible to estimate the sum of the means of this process even if these change at all time-steps.
>
> **Moving fixed variance results to the appendix.** We want to push back against the reviewers suggestion. It is true that our changing variance results imply fixed variance results by setting all $\sigma_\ell = \sigma$. Nonetheless, our changing variance results have an extra factor of $\sqrt{d_{eluder}}$ in their regret in contrast with the fixed variance setting (See Theorem 4.6 vs Theorem 4.10). On the other hand, we have developed a sharp analysis and algorithm for the fixed unknown variance case (Algorithm 1 with unknown variance - Section 4.2) whose dimension dependence is (we believe) optimal since it matches the sharpest results for linear contextual problems (Theorem 4.6).
>
> We hope this addresses the reviewer's concerns. In case it does, we would really appreciate it if the reviewer could consider increasing their score.

---

> > ### Author Response · Authors · 2024-11-26
> > **Follow up**
> >
> > Dear Reviewer QJSj,
> >
> > We want to reach out regarding our latest response. We hope it addresses the reviewer's concerns. In case it does, we would really appreciate it if the reviewer could consider increasing their score.
> >
> > Thanks a lot!
> >
> > the authors

---

> > > ### Author Response · Authors · 2024-11-30
> > > **Follow up**
> > >
> > > Dear Reviewer
> > >
> > > We wanted to make sure this reached you before the rebuttal period ends.
> > >
> > > Thanks!

---

> > > > ### Author Response · Authors · 2024-12-02
> > > > **Follow up**
> > > >
> > > > Dear Reviewer,
> > > >
> > > > A kind reminder that the back and forth period between authors and reviewers is coming to an end. We hope our rebuttal addressed the reviewer's concerns. If so, we would appreciate it if the reviewer could consider improving their score.
> > > >
> > > > Thanks!
> > > >
> > > > The Authors

---

### Meta-Review · Area_Chair_UYZx · 2024-12-21

**Metareview:**

Summary:
This work addresses contextual bandits with function approximation, making a significant contribution by establishing a second-order regret bound for this setting for the first time.

Strengths:

- The studied setting is important and relevant to the field.
- The authors present a technically sound approach with a rigorous and correct proof.

Weaknesses:

- The derived second-order regret bound includes an additional dimension factor, which prevents it from matching the regret bounds achieved in linear bandit settings.

Decision:
Accept.

**Additional Comments On Reviewer Discussion:**

During the discussion, the authors addressed several typographical errors and provided additional clarification to highlight the significance of their work.

---

### Decision · Program_Chairs · 2025-01-22

Accept (Poster)